# The role of Edge-Driven Convection in the generation of volcanism-part 2: Interaction with Mantle Plumes, application to the Canary Islands

Antonio Manjón-Cabeza Córdoba[1,2] and Maxim D. Ballmer[3,1]

[1]ETH Zürich, Department of Earth Sciences, Institute of Geophysics, Sonnegstrasse 5, Zürich, Switzerland
[2]University of Oslo, Centre for Earth Evolution and Dynamics, Sem Sælands vei 2A, Oslo, Norway
[3]University College London, Deparment of Earth Sciences, 5 Gower Place, London, UK

**Correspondence:** Antonio Manjón-Cabeza Córdoba (antonio.manjon@csic.es)

**Abstract.** In the eastern Atlantic Ocean, several volcanic archipelagos are located close to the margin of the African continent. This configuration has inspired previous studies to suggest an important role of edge-driven convection (EDC) in the generation of intraplate magmatism. In a companion paper (Manjón-Cabeza Córdoba and Ballmer, 2021: The role of Edge-Driven Convection in the generation of intraplate volcanism - part 1: a 2D systematic study, doi:10.5194/se-12-613-2021), we showed
that EDC alone is insufficient to sustain magmatism of the magnitude required to match the volume of these islands. However, we also found that EDC readily develops near a step of lithospheric thickness, such as the oceanic-continental transition ("edge") along the western African cratonic margin. In this work, we carry out 3D numerical models of mantle flow and melting to explore the possible interactions between EDC and mantle plumes. We find that the stem of a plume that rises close to a lithospheric edge is significantly deflected ocean-ward (*i.e.*, away from the edge). The pancake of ponding hot material
at the base of the lithosphere is also deflected by the EDC convection cell (either away or towards the edge). The amount of magmatism and plume deflection depends on the initial geometric configuration, *i.e.*, the distance of the plume from the edge. Plume buoyancy flux and temperature also control the amount of magmatism, and influence the style and extent of plume-EDC interaction. Finally, comparison of model predictions with observations reveals that the Canary plume may be significantly affected and deflected by EDC, accounting for widespread and coeval volcanic activity. Our work shows that many of the
peculiar characteristics of eastern Atlantic volcanism are compatible with mantle-plume theory once the effects of EDC on plume flow are considered.

## 1  Introduction

Volcanism exerts a major control for material flux between the interior of the Earth and the surface/atmosphere system. Volcanic activity along mid-ocean ridges and subduction zones is readily explained by plate tectonics. However, in the absence of nearby
plate boundaries, plate tectonics cannot account for intraplate volcanism.

Several models have been proposed to explain the origin of such magmatism. The leading hypothesis is mantle plume theory, in which a deep columnar thermal anomaly rises from the core-mantle boundary to the base of the lithosphere in order to support

localized hotspot volcanism (Wilson, 1963; Morgan, 1971). Still, several predictions of plume theory are not fulfilled at many locations worldwide (*e.g.*, linear age progressions consistent with plate velocity; Courtillot et al., 2003) and other models have been put forward: Small Scale Convection (SSC; Richter, 1973; Parsons and McKenzie, 1978; Huang et al., 2003; Dumoulin et al., 2005; Ballmer et al., 2007), Shear-Driven Upwelling (SDU; Conrad et al., 2010) or Edge-Driven Convection (EDC; King and Anderson, 1995, 1998).

In the Eastern Atlantic, several volcanic archipelagos are located on the ocean floor near continental lithosphere. At these locations, many of the predictions of plume theory are not met. In the Canary Islands (where volcanism is as recent as the 2021 eruption of La Palma) volcano ages do not follow a consistent linear age-distance relationship, with coeval volcanism occurring across several hundreds of kilometers (Abdel-Monem et al., 1971, 1972; Thirlwall et al., 2000; Geldmacher et al., 2005), the plume swell is nearly absent (Sleep, 1990; King and Adam, 2014) (although see Huppert et al., 2020), the duration of volcanism at a single island is longer than expected in comparison with other chains (*e.g.* Carracedo, 1999). Besides, a cogenetic relation of these volcanoes with Alboran Domain volcanism has been suggested due to tectonism (Doblas et al., 2007); and with the north-west Africa cenozoic volcanism as part of the same upwelling (Duggen et al., 2009). These inconsistencies have led several authors to reject the plume model for these islands (*e.g.* Doblas et al., 2007; Martínez-Arevalo et al., 2013). Similar arguments against the plume model have been made for Cape Verde (King and Ritsema, 2000; Helffrich et al., 2010) or the Cameroon Volcanic Line (Fitton, 1980; Déruelle et al., 2007; Milelli et al., 2012), both of which have also been formed near the African continental margin.

Of the alternative models put forward to substitute mantle-plume theory, EDC is the only one that has been proposed for the three aforementioned volcanic regions (King and Anderson, 1998; King and Ritsema, 2000; Milelli et al., 2012). The EDC model postulates that a convection cell is generated due to the juxtaposition of two lithospheric sections of different age or structure: the related density difference is sufficient to generate a downwelling and an associated upwelling. In theory, the return upwelling flow would be enough to generate magma to sustain ocean island volcanism, provided that the overlying lid was sufficiently thin to facilitate decompression melting. EDC has been proposed for other regions of the globe as well, for example Vogt (1991) suggested that EDC could be related to the Bermuda Rise, although several studies (Shahnas and Pysklywec, 2004; Ramsay and Pysklywec, 2011) showed that EDC would produce a maximum upwelling (and a related increase in topography) closer to the margin than the Bermuda Rise; Davies and Rawlinson (2014) have proposed EDC as the mechanism for the south-eastern Australian volcanic province; and Afonso et al. (2016) have proposed EDC as an important mechanism for the Central-Western US volcanism.

Nonetheless, in a previous paper (Manjón-Cabeza Córdoba and Ballmer, 2021), we quantitatively tested the hypothesis of Edge-Driven Convection as an origin of oceanic intraplate volcanism near continental margins, and our results showed that, by itself, EDC can only support minor magmatism even under the most favorable conditions, and is clearly insufficient to generate long-lived island-building volcanism. While other studies have shown that a very steep oceanic-continental transition (Kim and So, 2020; Negredo et al., 2022), or additional geometrical complexities (Duvernay et al., 2021), could increase the amount of EDC-related melting calculated by the companion study, all of them coincide that magmatism is very restricted to account for volcanism at the Canary Islands. Furthermore, we speculated that due to the prevalence of EDC with Earth-like

mantle properties, most of EDC-related flow and melting should occur near mid-ocean ridges in young lithospheres, which was previously observed by geological (Ligi et al., 2011) and geodynamic (Buck, 1986; Boutilier and Keen, 1999; Sleep, 2007) studies alike, and not in old lithospheres (as is the case of the Canary Islands).

On the other hand, recent seismic-tomography studies provide evidence for deeply-rooted mantle plumes in the Eastern Atlantic by imaging continuous near-vertical low-velocity anomalies in the mantle (French and Romanowicz, 2015) or broad upwellings just below these archipelagos (Civiero et al., 2021). In addition, additional geophysical evidence points to the presence of thermal upwellings (plumes) at least from the base of the transition zone (Liu and Zhao, 2014; Saki et al., 2015).

In the light of the evidence gathered along these lines, we here explore the dynamics of mantle flow and melting related to plumes that rise near a continental margin (figure 1). We hypothesize that the interaction between plumes and EDC can explain (at least some of) the discrepancies between the predictions of plume theory and observations, as already suggested by Geldmacher et al. (2005). To study the interaction between plumes and EDC, we carry out three-dimensional (3D) numerical models of flow and melting near the transition between the oceanic and the continental lithosphere. Negredo et al. (2022) carried out a preliminary study of plumes and their potential interactions with EDC in 2D, finding that EDC could be responsible of plume migration in the Canaries. However, plumes are inherently 3D and including this third dimension allows us to include the effects of plate velocity. We explore the parameters that control plume flow (*e.g.*, plume buoyancy flux, plume excess temperature) and EDC (*e.g.*, mantle viscosity, distance of the plume from the continental margin).

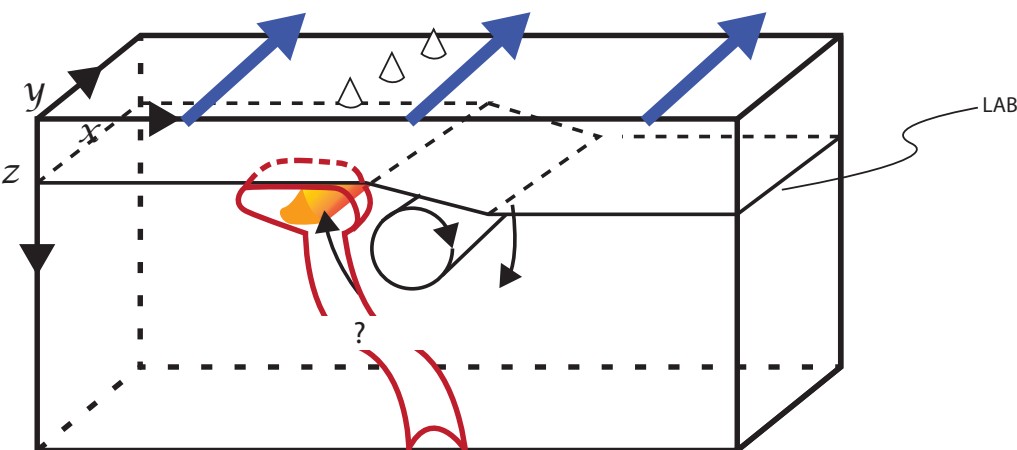

**Figure 1.** Schematic of a plume interacting with Edge-Driven Convection. In this work, we build on the models of Manjón-Cabeza Córdoba and Ballmer (2021) and add a plume in the form of a temperature anomaly at the bottom, and a plate velocity $v_{plate} = 2$ cm·yr$^{-1}$ consistent with the age-distance track to the north of the Canary Islands. An idealized Lithosphere-Asthenosphere Boundary (LAB) is labeled in the figure.

## 2  Methods

We run 3D-Cartesian numerical models using the same version of the finite-element code CITCOM (Moresi and Solomatov, 1995; Moresi and Gurnis, 1996; Zhong et al., 2000) as in our previous paper (Manjón-Cabeza Córdoba and Ballmer, 2021). The conservation equations of mass, momentum and energy are solved on the finite-element mesh according to the "extended Boussinesq approximation" (Christensen and Yuen, 1985) (although we impose the adiabat as a constant gradient with depth of 0.3 K/km); composition is tracked by passive Lagrangian particles (*i.e.*, tracers). 3D geometry of the model box is chosen due to the intrinsic 3D nature of the problem (see Figure 1) and the related complex flow patterns. To make our models comparable with the 2D cases in the companion paper (Manjón-Cabeza Córdoba and Ballmer, 2021), we use the same model-box depth $z_{box}$ and width $x_{box}$. The total extent of our computational domain is 2640×1980×660 km ($x_{box}$, $y_{box}$, and $z_{box}$, respectively). This domain is resolved by a grid of 384×288×96 elements with an uniform spacing. Resolution tests in the companion paper (Manjón-Cabeza Córdoba and Ballmer, 2021) confirmed that this is enough to accurately model EDC. Our plumes do not feature too low viscosities due to our low activation energies (Appendix, Figure A2), and are similarly resolved as regional or global models (*e.g.* Ballmer et al., 2013; Davies and Davies, 2009) for hotter (less viscous) plumes.

Free-slip is imposed at the side boundaries ($x = 2640$ km and $x = 0$ km); no slip is imposed at the bottom. To model Atlantic plate motion and achieve a steady-state for plume inflow and outflow, we impose a plate velocity at the top boundary layer parallel to the $y$-direction (the new dimension added, parallell to the edge) of $v_{plate} = 2$ cm·yr$^{-1}$; and a related Couette flow at the inflow boundary ($y = 0$ km) that is consistent with the viscosity profile. We acknowledge that the real absolute African plate motion could be oblique to the African margin near the Canaries today, but the volcanic track reflects a history of motion nearly parallel to the African Margin (Geldmacher et al., 2005); in any case, most frames of references depict a plate-movement parallel to the margin (Schellart et al., 2008; Martín et al., 2014). The corresponding outflow velocity boundary (at $y = 1980$ km) remains unconstrained to allow free exit of material, but we impose all flow perpendicular to the boundary (that is, no slip in the directions parallel to the boundary). We also open an unconstrained circular "hole" at the bottom of the box and $y$=660 km to allow free inflow at the plume location (Ballmer et al., 2011).

The top boundary is fixed at $T_{surf} = 0$ °C, while the bottom boundary is fixed at $T_{ref} = 1350$ °C (+198 °C are added corresponding to the adabatic gradient increase 0.3 K/km × 660 km); the $x$-normal boundaries are reflective. The models are internally heated as well ($H$=7.75×10$^{-12}$ W kg$^{-1}$). At the inflow boundary, the thermal distribution corresponds to the initial condition, which is identical to that of the 2D profile of the previous paper (Figure 2 in Manjón-Cabeza Córdoba and Ballmer, 2021), including a continental "edge" at $x = 1320$ km (fig. 2). In nearly all cases of this study, the initial thermal age of the juxtaposed continental and oceanic lithospheres are $\tau_c = 100$ Ma and $\tau_o = 40$ Ma respectively, except for when otherwise specified. This choice of $\tau_o$ results in an age of $\tau_{o,y=660} = 73$ Ma for the oceanic lithosphere right above the plume anomaly.

The transition between the two lithospheric thicknesses is linearly interpolated for both temperature and composition (Figure 2). The width of the transition is $w = 264$ km in all cases. In addition, we impose a circular plume thermal anomaly of radius $r_{plume}$ centered at $y = 660$ km (*i.e.*, sufficiently far away to avoid artifacts due to the proximity to the inflow boundary), and

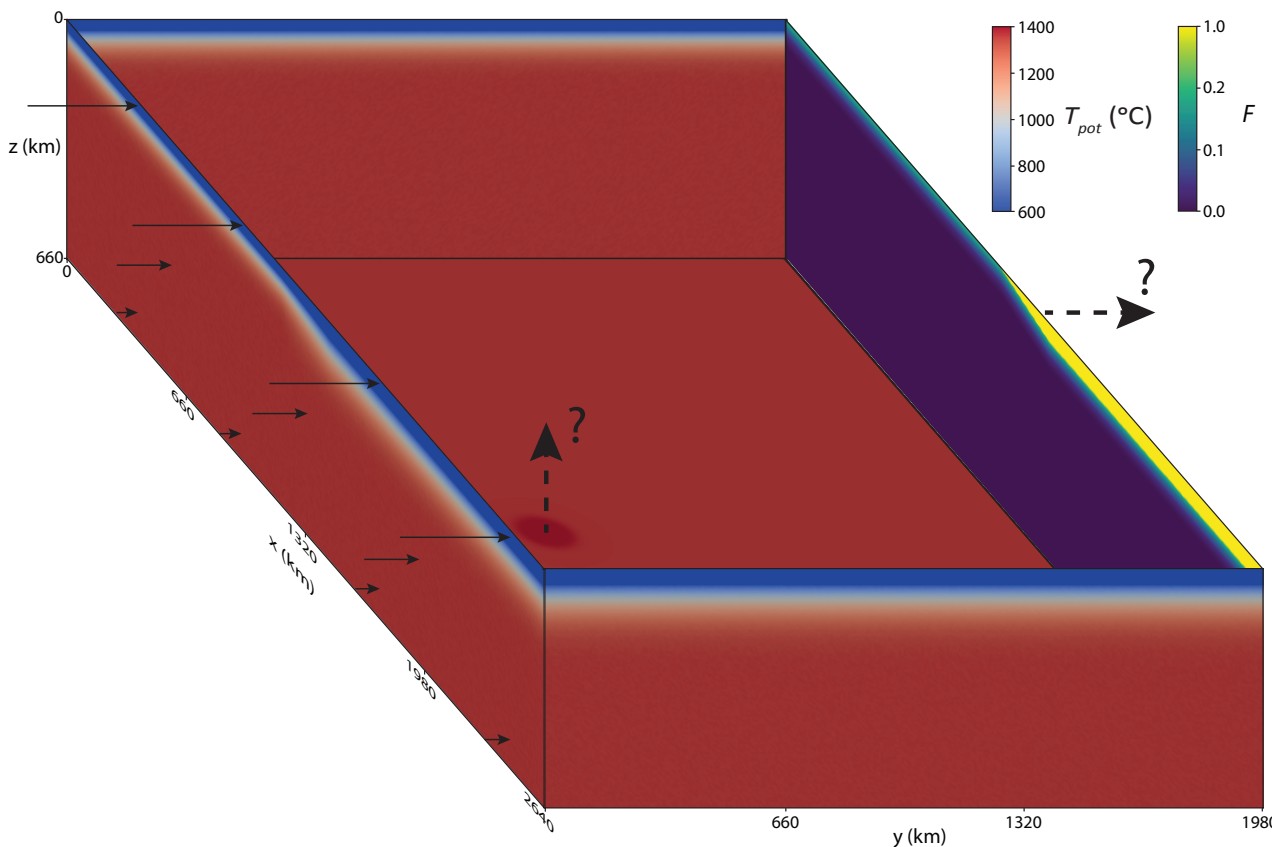

**Figure 2.** 2D sections depicting the initial thermal (potential temperature) and compositional (depletion for the "depleted component) profiles for the models in this work. Depletion (F) is defined as the amount of melt extracted from the mantle. Plate motion is imposed at the top boundary from left to right. The left side shows the initial thermal and inflow boundary conditions. The right side shows the initial compositional and inflow boundary conditions. The front and back sides show the thermal side boundary conditions. Solid and discontinuous arrows represent constrained (imposed) and unconstrained (open boundary) velocity boundaries respectively. For further details on the calculations of the initial profiles, see text.

variable distances from the edge $D_{plume}$ =1320 km - $x_{plume}$. The plume thermal anomaly at the bottom is described by the following condition:

$$\Delta T = \Delta T_{plume} \cdot e^{\frac{-r^2}{r_{plume}^2}} \tag{1}$$

110 where $\Delta T$ is the difference between the plume temperature and the background temperature, and *r* is the distance from $x_{plume}$. Plume buoyancy flux is kept nearly constant during the simulation by automatically adjusting $r_{plume}$ every 50 timesteps. For example, if the plume buoyancy flux $B(t)$ (measured at the bottom boundary) is different from the target value $B_{plume}$ for a

given model, $r_{plume}$ is adjusted by a factor of $\frac{B_{plume}}{B(t)}^{0.5}$. This approach keeps $B(t)$ practically constant through much of the simulation, but renders the ratio between the radius of the opening at the bottom of the model and $r_{plume}$ variable between cases. Nonetheless, we make sure that this ratio remains between 3.5 and 4 for all models in the statistical steady state. In this statistical steady state, $r_{plume}$ does not vary (see supplementary video).

Statistical steady state is evaluated by analyzing the vVrms (root mean squares of the vertical velocity component) and the melt flux (see below). Models are only evaluated when these properties do not change systematically over time (small chages are expected due to random thermal noise). The exception to this evaluation are the models with a $D_{plume} = 400$ km (see below, supplementary video), which featured periodic behavior. In those cases, evaluation occurs when as soon as the cycles are statistically symmetric. For examples of vVrms and melt-flux plots used to evaluate the statistical steady state, see Appendix, Figure A1.

The mantle source consists of a mechanical mixture of three different lithological components (depleted/dry peridotite, enriched/hydrous peridotite, pyroxenite), which make up 82 %, 15 %, and 3 % of the volume of the mantle (respectively). We assume that these lithologies are in thermal equilibrium but chemical disequilibrium due to their fine-scale nature (*i.e.*, smaller than the finite-element mesh). Each of these lithologies has a different density and is subject to a different melting law (Katz et al., 2003; Pertermann and Hirschmann, 2003; Manjón-Cabeza Córdoba and Ballmer, 2021; Ballmer et al., 2009). Initially, the lithosphere is depleted in all of the lithologies, and hence is buoyant and does not melt immediately. This lithological depletion is pre-calcculated from 2D models of flow and melting at a mid-ocean ridge (Manjón-Cabeza Córdoba and Ballmer, 2021), using the same method and parameters (*e.g.*potential temperature and melting laws) as the current study. Such approach restricts excess melting that may otherwise occurs due to arbitrary model choices (*e.g.*, an increased potential temperature of the models would be met with an increased melt depletion at mid-ocean ridges, therefore limiting further melting).

Progressive melting during the simulation affects the relevant densities due to melt retention and depletion of the residue. The driving forces (density anomalies) further depend on temperature and composition. As in the companion paper, the linearized density formulation is:

$$\rho = \rho_{ref} - \alpha \cdot \rho_{rhef} \cdot (T - T_{ref}) + F \cdot \Delta\rho_F + \phi \cdot \Delta\rho_\phi \tag{2}$$

this equation takes into account thermal expansivity ($\alpha$) as well as the density effects of melt fraction ($F$) and melt retention ($phi$). For the values of the characteristic constants of these density effects ($\Delta\rho_F$ and $\Delta\rho_\phi$) see table 1.

In turn, the resisting forces (viscosities) do not depend on melt retention or depletion. Our Newtonian viscosity formulation is temperature- and depth-dependent:

$$log\eta = log\eta_0 + \frac{E_a + P \cdot V_a}{R \cdot T} - \frac{E_a}{R \cdot T_{ref}} \tag{3}$$

with $E_a$ and $V_a$ the activation energy and volume (respectively). The chosen value for activation energy (table 1) is smaller than values obtained for fitting experimental data (Kohlstedt and Hansen, 2015) but is useful to obtain lithospheric thickenesses close to those obtained with non-Newtonian rheologies (Christensen, 1984; van Hunen et al., 2005).

**Table 1.** Relevant parameters for the models described in this paper. Values outside and inside of parentheses provide the reference value and the explored parameter space respectively.

| Notation | Parameter | Reference value (explored range) | Unit |
|---|---|---|---|
| $T_{ref}$ | Reference temperature | 1350 | °C |
| $D$ | Reference thickness | 660 | km |
| $\rho_{ref}$ | Reference density | 3300 | kg m$^{-3}$ |
| $\kappa$ | Thermal diffusivity | $1 \times 10^{-6}$ | m$^2$ s |
| $g$ | Gravity acceleration | 9.8 | m s$^2$ |
| $\alpha$ | Thermal expansivity | $3 \times 10^{-5}$ | K$^{-1}$ |
| $c_P$ | Heat capacity (constant pressure) | 1250 | J kg$^{-1}$ K$^{-1}$ |
| $\Delta\rho_F$ | Density variation with depletion | -100 | kg m$^{-3}$ |
| $\Delta\rho_\phi$ | Density variation with melt retention | -500 | kg m$^{-3}$ |
| $\eta_0$ | Reference viscosity | $8.29 \times 10^{18}$ ($5.53 \times 10^{18}$-$1.24 \times 10^{19}$) | Pa s |
| $E_a$ | Activation energy | 200 | kJ mol$^{-1}$ |
| $V_a$ | Activation volume | $5.00 \times 10^{-6}$ | m$^3$ mol$^{-1}$ |
| $\gamma_a$ | Adiabatic gradient | 0.3 | K km$^{-1}$ |
| $H$ | Internal heating | $7.75 \times 10^{-12}$ | W kg$^{-1}$ |
| $v_{plate}$ | Plate velocity | 2 | cm yr$^{-1}$ |
| $B_{plume}$ | Buoyancy flux | 100 (50-500) | kg s$^{-1}$ |
| $\Delta T_{plume}$ | Excess temperature of the plume | 150 (100-200) | °C |
| $D_{plume}$ | Distance of the plume thermal anomaly from the edge | 0 (0-400) | km |

## 3   Results

In the 2D models of the companion paper (Manjón-Cabeza Córdoba and Ballmer, 2021), we find that EDC starts right at the onset of the model evolution with a dominant downwelling below the continental side of the edge (or ocean-continent transition), and a return-flow upwelling below the oceanic side. The upwelling sustains erosion of the lithosphere, creating a "bump" or "dent" at its base. This "Dent" is characteristic of every model in this work and also present when no plume is imposed (Appendix, Figure A3). Ultimately, SSC also occurs at the base of the oceanic lithosphere far from the edge. We refer to SSC as a thermal-boundary layer instability that (in contrast to EDC) is not immediately triggered by the presence of a nearby edge, but rather typically occurs as soon as the boundary layer (nearly) reaches its critical thickness (Richter, 1973; Parsons and McKenzie, 1978).

In this study, the test cases without a mantle plume confirm that the results of Manjón-Cabeza Córdoba and Ballmer (2021) are robust and hold in our 3D geometry: EDC begins right after the material enters the model box, promoting a convection cell and related sub-lithospheric erosion above the upwelling on the oceanic side. SSC develops in our 3D models, appearing sooner (*i.e.*, closer to the inflow boundary) near the edge than far away for it, with convection cells typically aligned parallel to plate motion as "Richter rolls" (Richter, 1973; Richter and Parsons, 1975; Marquart, 2001; Huang et al., 2003). The development of Richter Rolls is stable even for our low $v_{plate}$ = 2 cm/yr due to our high Rayleigh number (Korenaga and Jordan, 2003). Note that no EDC melting is found neither in the case without a plume (Figure A3) nor in 2D cases with the same parameters as in this work (Manjón-Cabeza Córdoba and Ballmer, 2021).

Figure 3 shows the results of the reference case, which includes a plume with $\Delta T_{plume}$ = 150 °C, $B_{plume}$ = 100 kg·s$^{-1}$, and $D_{plume}$ = 0 km. Compared to other geodynamic studies of mantle plumes (Ribe and Christensen, 1994; Ballmer et al., 2011), the most evident characteristic of this model is the lateral deflection of the plume conduit. Instead of ascending vertically, the plume conduit is displaced towards the oceanic side with thinner lithosphere. This displacement suggests some interaction of plume flow with EDC-related flow.

The plume ponds at the base of the lithosphere as a pancake of hot material. Before melting depths, the temperature excess of the plume with respect to the initial potential temperature is nearly the same (within two degrees depending on the depth) as the thermal anomaly at the bottom, due to our approximation of the adiabat (note that at some depths, shear heating and internal heating may increase the temperature excess of the plume above $\Delta T_{plume}$, but this was found to be the exception compared to a slightly lower value). The hottest central part of the plume pancake is located at the minimum thickness of the oceanic lithosphere (*i.e.* at the aforementioned "dent" or "bump"). Without further analysis, however, it remains unclear whether the plume is conveyed to this minimum thickness created by EDC, or if the plume actively creates a dent, and EDC reorganizes accordingly. The plume pancake and melting zones are slightly asymmetric, but again: it remains unclear whether this asymmetry is due to the spreading of the pancake at the base of a lithosphere with variable thickness at the ocean-continent transition, or it is caused by EDC-related flow.

The plume acts to efficiently erode the imposed edge at the base of the lithosphere, displacing the thermal boundary layer. This erosion also creates a Plume Erosion Track (PET) that is observed in all our models (Ribe and Christensen, 1994). The PET can be defined thermally, as the region were heat flow is increased due to plume erosion (discontinuous line in Figure 3); or dynamically, as the erosion of the lithosphere that would contribute to dynamic topography (continuous line in Figure 3). In the reference case (Figure 3), the PET is mostly parallel to the direction of plate motion (independently of how is defined). The plume also displaces the main EDC-downwelling continent-wards: this effect starts around the plume pancake, but it continues downstream. In the reference case, it is difficult to clearly distinguish between the downwelling associated with the PET (*i.e.* plume curtain) and the main EDC downwelling (compare Figures A3 and 3).

To better quantify the lateral displacement of the plume stem and the pancake, we calculate a Plume Deflection Index (PDI) defined simply as the inverse of the slope ($\frac{\Delta x}{\Delta z}$) between two temperature maxima at two different depths. PDI$_{stem}$ is a proxy for the plume-stem displacement, calculated as the lateral distance between the plume stem (and related thermal maxima) at $z$ = 220 km and at $z$ = 660 km (divided by the difference between both these depths, *i.e*, 440 km); in addition, we also report a

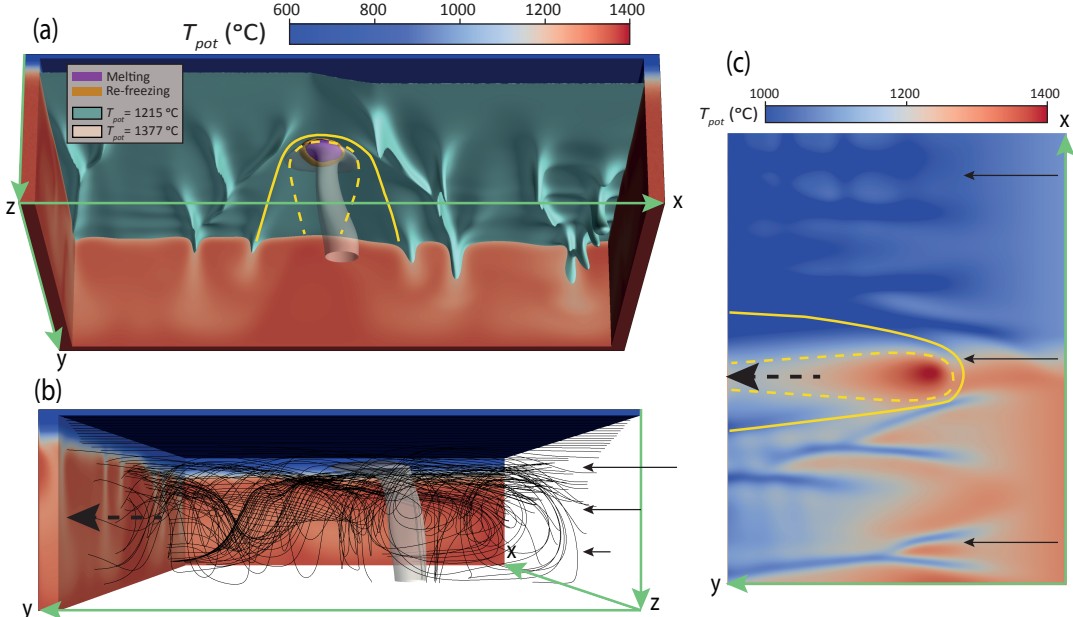

**Figure 3.** Steady-state temperature field and melting for the reference case with $\Delta T_{plume} = 150\,^{\circ}\mathrm{C}$, $B_{plume} = 100\ \mathrm{kg \cdot s^{-1}}$, $D_{plume} = 0\ \mathrm{km}$, and $\eta_0 = 8.29\cdot10^{18}$ Pa·s. (a) Front perspective of the reference case. Cross sections of potential temperature (at the margins of the model) are shown in red-to-blue colors. The light blue and white surfaces mark isotherms (as labeled), showing the base of the lithosphere and the plume, respectively. The purple isosurface outlines the region of active melting while the orange isosurface outlines the region of finite melt presence, including where active melt re-freezing occurs. Dashed and solid lines depict the dynamic and thermal PET (see text) respectively. (b) Side view of the model with stream lines. (c) top-view with cross section of the temperature field at z = 106 km. Black arrows depict plate motion. Note that the colorscale is different than in panels (a) and (b).

$\mathrm{PDI}_{pancake}$, which is calculated from the thermal maxima at depths of $z = 110$ km and $z = 220$ km, and otherwise analogously to $\mathrm{PDI}_{stem}$. In this work, we arbitrarily define positive values of PDI as distortions of upwelling flow "away from the edge" and negative values as "towards the edge". In the reference case, $\mathrm{PDI}_{stem} = 0.143$ and $\mathrm{PDI}_{pancake} = 0.109$. Both the stem and the pancake are deflected towards the oceanic side. These values correspond to absolute displacements of the plume towards the oceanic domain of 63 km from 660 km to 220 km depth, and another 12 km from 220 km to 110 km depth. In particular, the lateral displacement of the plume stem is significant. We will discuss the relevance of these values in comparison to other cases below.

We also investigate the compositional origin of mantle melts as a proxy for their geochemical signature. To do this, we evaluate the total melt volume flux $M$ (*i.e.*, melt produced in the mantle) and total volcanic volume flux $V$ (*i.e.*, melt extracted from the mantle), along with the melt flux and volcanic flux that is related to pyroxenite melting only: $M_{PX}$ and $V_{PX}$, respectively. These metrics provide a compositional index for mantle melting, $\frac{M_{PX}}{M}$, and melt extraction, $\frac{V_{PX}}{V}$. The latter

is the compositional origin of volcanism explicitly predicted by our models. Note, however, that this specific prediction of lithological origin of volcanism depends on the critical porosity explicitly assumed here (1 %), and on the style of melt extraction. For example, if pyroxenite-derived and peridotite-derived melts were already pooled in the mantle (instead of in a shallow magma chamber), and then were extracted together, or if all melts were efficiently extracted (*i.e.*, for fully fractional melting), $\frac{M_{PX}}{M}$ would provide a more appropriate geochemical proxy than $\frac{V_{PX}}{V}$. In other words, both $\frac{M_{PX}}{M}$ and $\frac{V_{PX}}{V}$ provide reasonable bounds for the compositional origin of predicted lavas.

In the reference case, $\frac{M_{PX}}{M}$ and $\frac{V_{PX}}{V}$ are 0.774 and 0.994, respectively. Such a dominance of pyroxenite-derived melting and volcanism is mostly explained by the relatively low plume excess temperatures and large relative seafloor ages modeled here (and relevant for Eastern Atlantic Volcanism; Müller et al., 2008). The related large lithospheric thicknesses restricts extensive peridotite melting, even though peridotite is the most abundant component in the plume source. Also note that pyroxenite melting starts at greater depths than peridotite melting and efficiently extracts latent heat, such that the ascent of peridotitic material is sub-adiabatic (less melting) and the ascent of pyroxenite material is super-adiabatic (more melting, Hirschmann and Stolper, 1996).

## 3.1 Effects of plume temperature

We conduct a series of cases with variable plume excess temperature $\Delta T_{plume}$ and constant buoyancy flux $B_{plume}$. We find that $\Delta T_{plume}$ has only minor effects on the overall flow patterns at a given $B_{plume}$. As $B_{plume}$ is kept constant, the radius of a hotter plume is implicitly smaller than that of a cooler plume. As a consequence of this implicit effect of $\Delta T_{plume}$ on plume radii, the plume pancake and the related PET tend to be wider for smaller $\Delta T_{plume}$. While the base of the lithosphere is eroded more efficiently for large $\Delta T_{plume}$, because a hotter plume sustains a lower-viscosity pancake, the differences are extremely small as far as $B_{plume}$ is kept constant (Appendix, Figure A4).

There is no indication that changing $\Delta T_{plume}$ while keeping $B_{plume}$ constant systematically changes the effect of EDC-related flow on plume ascent (or the effect of plumes on EDC). The lateral displacement of the plume by EDC is similar across all our models with different $\Delta T_{plume}$, as evidenced by the nearly flat trends of $PDI_{stem}$ and $PDI_{pancake}$ (Figure 4a). The only noticeable difference between the models is that the plume pancake is more asymmetric for the case with $\Delta T_{plume}$ = 100 °C than for greater $\Delta T_{plume}$. Note also that all PDIs in Figure 4a are positive, implying that the plume is consistently deflected away from the edge at all depths.

Melt fluxes (*i.e.*, volume fluxes of melts produced in the mantle) and volcanic fluxes (*i.e.*, volume fluxes of melts extracted from the mantle) systematically increase with $\Delta T_{plume}$ (fig. 4b). This result is intuitive, and consistent with previous work (Ribe and Christensen, 1994; Ballmer et al., 2011). In terms of the compositional origin of magmas, $\frac{M_{PX}}{M}$ and $\frac{V_{PX}}{V}$ decreases with the amount of melt produced, and therefore decreases with increasing $\Delta T_{plume}$ (fig. 4b). This result is expected as PX-derived melts are diluted by peridotite-derived melts for increasing degrees of melting.

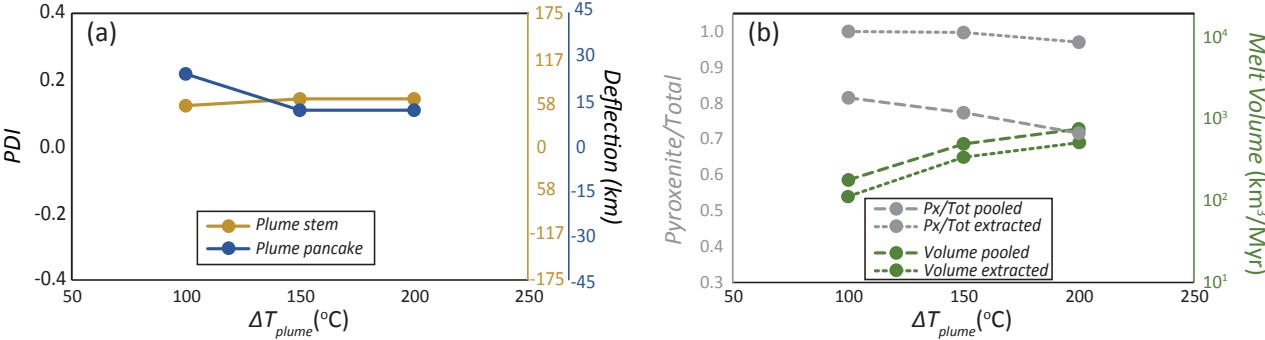

**Figure 4.** Diagrams showing the sensitivity of several output parameters as a function of $\Delta T_{plume}$. (a) Plume Distortion Index (PDI) for the different models (see text for explanation). Note that all values are positive (deflection away from the edge). (b) Melt volumes fluxes ($M$ and $V$) and melt compositional index ($\frac{M_{PX}}{M}$ and $\frac{V_{PX}}{V}$) for the different models. Predictions in terms of melt production ($M$, $\frac{M_{PX}}{M}$) are given as dashed lines; predictions in terms of melt extraction ($V$, $\frac{V_{PX}}{V}$) as dotted lines.

## 3.2 Effects of Plume Buoyancy Flux

We also explore the influence of $B_{plume}$ on model results. Figure 5 shows steady-state model predictions for cases with different $B_{plume}$, but otherwise the same parameters as in the reference case. Increasing $B_{plume}$ implicitly increases the radius of the plume. Thereby, the width and volume of the melting zone and of the plume pancake also increase, as does the area of PET. The PET remains mostly parallel to the plate velocity vector, as for the reference case.

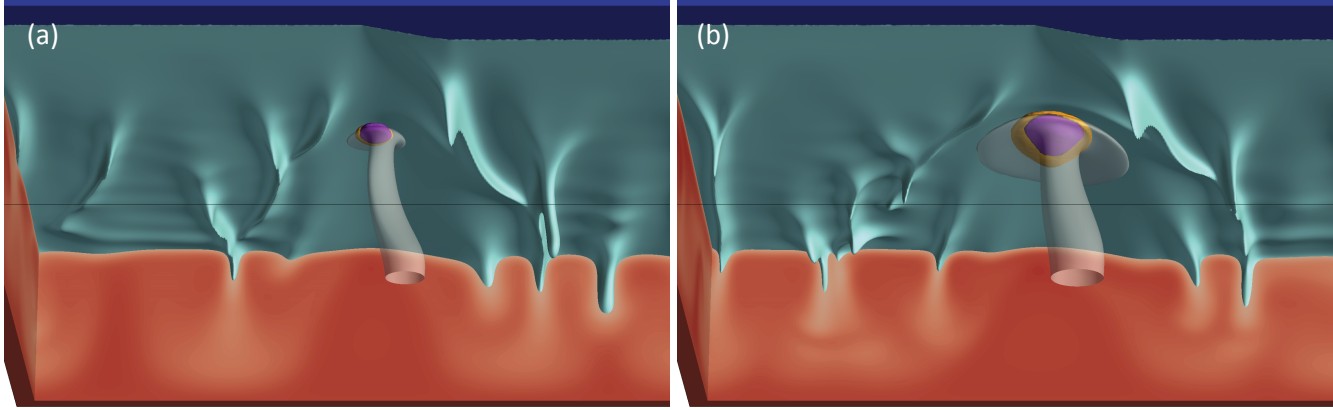

**Figure 5.** Steady-state snapshots of representative cases with different $B_{plume}$ but otherwise the same parameters as in the reference case (Figure 3). (a) $B_{plume}$ = 50 kg·s$^{-1}$; (b) $B_{plume}$ = 200 kg·s$^{-1}$. For reference to colors of surfaces and cross-sections, see Figure 3 caption and legend.

The lateral deflection of the plume stem is less evident for cases with higher than for cases with lower $B_{plume}$. Indeed, the high buoyancy-flux plume rises more straightly through the model box than the plume in the reference case. In fact, $PDI_{stem}$ tends to 0 as $B_{plume}$ increases (Figure 6a,c), providing evidence for a limitation of the ability of EDC (or of SSC in general) to affect the rise of plumes: efficient displacement is restricted to plumes with moderate-to-low buoyancy fluxes. Nevertheless,

the melting zone and the plume pancake display subtle asymmetry also in the case with the highest $B_{plume}$ modeled here. As for $PDI_{stem}$, $PDI_{pancake}$ also tends to decrease for increasing $B_{plume}$, but remains positive.

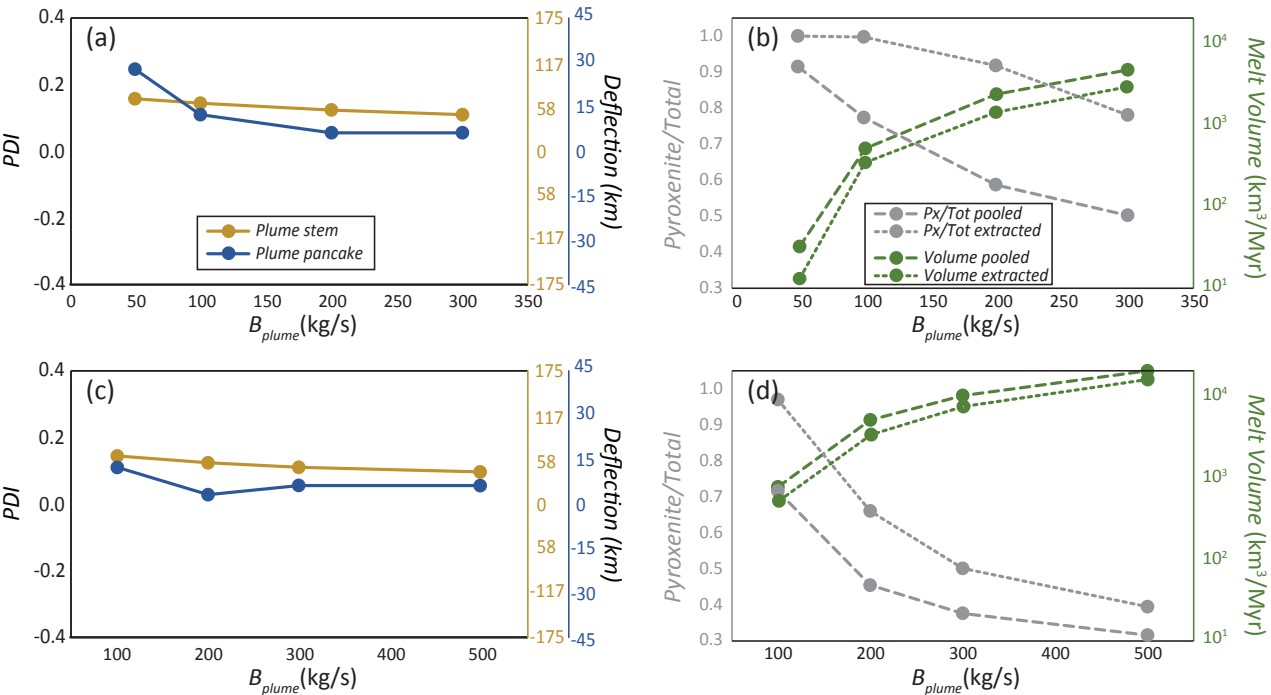

**Figure 6.** Diagrams showing the variation of key output parameters to changing $B_{plume}$. (a) $PDI_{stem}$ and $PDI_{pancake}$ for models with variable $B_{plume}$ and $\Delta T_{plume}$ = 150 °C. (b) $M$, $V$, $\frac{M_{PX}}{M}$, and $\frac{V_{PX}}{V}$ for models with variable $B_{plume}$ and $\Delta T_{plume}$ = 150 °C. (c) Same as (a) for models with $\Delta T_{plume}$ = 200 °C. (d) Same as (b) for models with $\Delta T_{plume}$ = 200 °C.

As far as the position of the main downwelling of EDC is concerned, increasing $B_{plume}$ increases the displacement of the main EDC downwelling towards the continent around the plume pancake. Once the plume pancake erodes the continent-ocean trainsitional lithosphere, the main EDC downwelling is not shifted towards the continent side further downstream, even if PET

is wider for cases with higher $B_{plume}$. In other words, while the plume controls the position of the main EDC downwelling close to the hotspot, the PET does not have an active effect on the position of the main EDC downwelling.

Due to the aforementioned radius increase as a function of $B_{plume}$, $M$ and $V$ both systematically increase with increasing $B_{plume}$. Regarding melt compositions, $\frac{M_{PX}}{M}$ and $\frac{V_{PX}}{V}$ display a shape that mirror melt volumes (Figure 6b,d), decreasing with increasing $B_{plume}$. Similar to the effects of plume excess temperature (see Figure 4), the trends of melt volumes and

compositions as a function of $B_{plume}$ mirror each other, because $\frac{M_{PX}}{M}$ and $\frac{V_{PX}}{V}$ decrease with increasing degrees of melting

of the dominant lithology, peridotite.The influence of $B_{plume}$ on magma compositions decreases at higher buoyancy fluxes,

probably because the extent of vertical sublithospheric erosion becomes nearly independent of $B_{plume}$ at some point. Note that

the convex upward shape of the dotted grey line in Figure 6b is due to the saturation of PX contributions at $\sim$100%. In Figure

6, the difference in composition between the produced melts and the extracted melts is greater for lower $\Delta T_{plume}$, which is not

evident in Figure 4. This is explained by the much higher productivity of pyroxenite melting (and hence: smaller sensitivity to

an extraction threshold) than for peridotite melting at high $B_{plume}$. Among all the parameters explored in this work, $B_{plume}$

shows the strongest effect on plume vigor and related melting.

### 3.3   Effects of distance of the Plume from the Edge

Next, we analyze the effects of the distance of the base of the plume from the edge, $D_{plume}$, on model results. The effects of

this parameter are a good indicator of plume-EDC interaction, because $D_{plume}$ changes the spatial relationship between the

plume and the edge, while leaving intrinsic plume parameters unchanged. Figure 7 shows 3D snapshots of mantle temperature

and melting as a function of $D_{plume}$ for two sets of $\Delta T_{plume}$ and $B_{plume}$ (in the top row, for a relatively weak plume with

parameters such as in the reference case: $\Delta T_{plume}$ = 150 K and $B_{plume}$ = 100 kg·s$^{-1}$; and the bottom row, for a moderately

strong plume with $\Delta T_{plume}$ = 200 K and $B_{plume}$ = 200 kg·s$^{-1}$). Interaction of the plume with the EDC convection cell and

topography at the base of the lithosphere causes systematic changes in the flow patterns and related melting characteristics. For

$D_{plume}$ = 200 km (fig 7a,c) the plume stem is deflected in a similar way as in the reference case (Figure 3), for which $D_{plume}$

= 0 km. For $D_{plume}$ = 400 km, the plume stem is instead generally less affected by the presence of the edge and related EDC.

Regarding the plume pancake, we find two significant changes in the behavior predicted by our cases with variable $D_{plume}$

with respect to the reference case (Figure 3). First, as the plume is shifted away from the edge (*i.e.*, for increasing $D_{plume}$), the

plume pancake is deflected towards the edge. This transition happens at a different $D_{plume}$ depending on plume properties ($\sim$25

km in Figure 8a, $\sim$125 km in 8c), but it happens nonetheless. After this rather sudden transition, the edge-ward deflection of

the pancake decreases progressively with increasing $D_{plume}$. At a distance of $D_{plume}$= 400 km, another notable phenomenon

occurs: vigorous SSC appears in the plume pancake with dominant transverse rolls (*i.e.*, perpendicular to the edge, Figure

7b,d). This peculiar geometry of SSC separates the plume-fed melting zone into two distinct melting zones (Figure 7b,d).

This separation is transient, however: as the SSC downwellings move with the plate, the two melting zones are separated and

merged periodically (see supplementary video). Dominant transverse rolls are a specific prediction for cases with $D_{plume}$ =

400 km. They neither occur for cases with any other $D_plume$, nor for test cases without a plume (see Appendix Figure A3).

This prediction highlights the subtle effects that EDC may have on plume-lithosphere interaction as a function of $D_{plume}$ and

EDC flux. In any case, SSC transverse rolls have also been found in studies of plumes without a nearby edge (Ballmer et al.,

280 2011), or studies of EDC without a plume present (Kaislaniemi and Van Hunen, 2014).

Figure 8 shows the effect of $D_{plume}$ on quantitative characteristics of plume ascent. Note that the significant changes shown

in Figure 8, both in terms of PDI and melt fluxes as a function of $D_{plume}$, are exclusively due to plume-EDC interaction

(intrinsic plume parameters remain unchanged in each row of Figure 8). PDI$_{stem}$ is generally positive, but highly variable. It

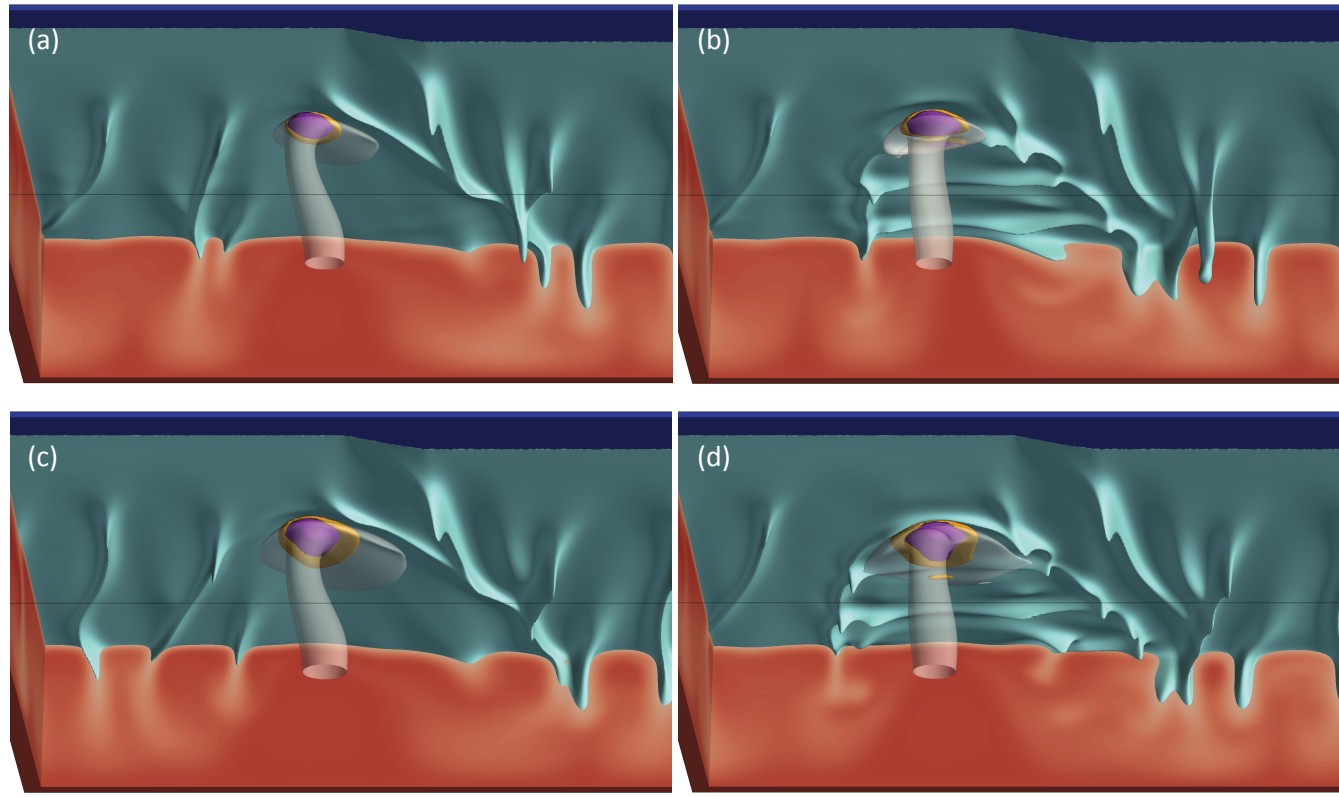

**Figure 7.** Steady-state snapshots of representative cases with variable $D_{plume}$. (a,c) Cases with $D_{plume}$ = 200 km. (b,d) Cases with $D_{plume}$ = 400 km. In the top row, models with a relatively weak plume with $\Delta T_{plume}$ = 150 °C and $B_{plume}$ = 100 kg·s$^{-1}$ are shown. In the bottom row, models with a relatively strong plume with $\Delta T_{plume}$ = 200 °C and $B_{plume}$ = 200 kg·s$^{-1}$ are shown. For reference to colors of surfaces and cross-sections, see fig. 3 caption and legend. The 3D perspective is the same as in Figure 3.

peaks at $D_{plume}$ = 50 km and $D_{plume}$ = 150 km for the relatively weak and strong plumes shown in the top and bottom rows
285 of Figure 8, respectively. For higher $D_{plume}$, PDI$_{stem}$ systematically decreases with $D_{plume}$. In turn, PDI$_{pancake}$ becomes
strongly negative for the $D_{plume}$ at which PDI$_{stem}$ peak, and progressively less negative for any higher $D_{plume}$. These results
emphasize the strong effects of plume-EDC interaction, and its diversity as a function of $D_{plume}$ (and for plumes with different
$\Delta T_{plume}$ and/or $B_{plume}$). The switch to dominantly transverse rolls in the pancake for plumes far from the edge (*i.e.* at $D_{plume}$
= 400 km) does not seem to strongly affect the deflection of the plume stem or shallow pancake.
290     As the plume pancake, PET changes with changing $D_{plume}$, becoming more asymmetric (with respect to the plate velocity
vector) and being deflected continent-wards whenever $PDI_{pancake}$ is negative. As a result, the continent-side limit of the PET
(right side in Figures 3, 7) always remains close to the main EDC-downwelling. The PET is also affected by SSC in the plume
pancake perpendicular to plate motion. Very likely, these predictions have implications for dynamic topography and swell

geometry. While the PET and the plume pancake are greatly affected by $D_{plume}$, this is not the case for the geometry of the
main downwelling of EDC. This downwelling is displaced by the plume in a similar way in all cases with variable $D_{plume}$.

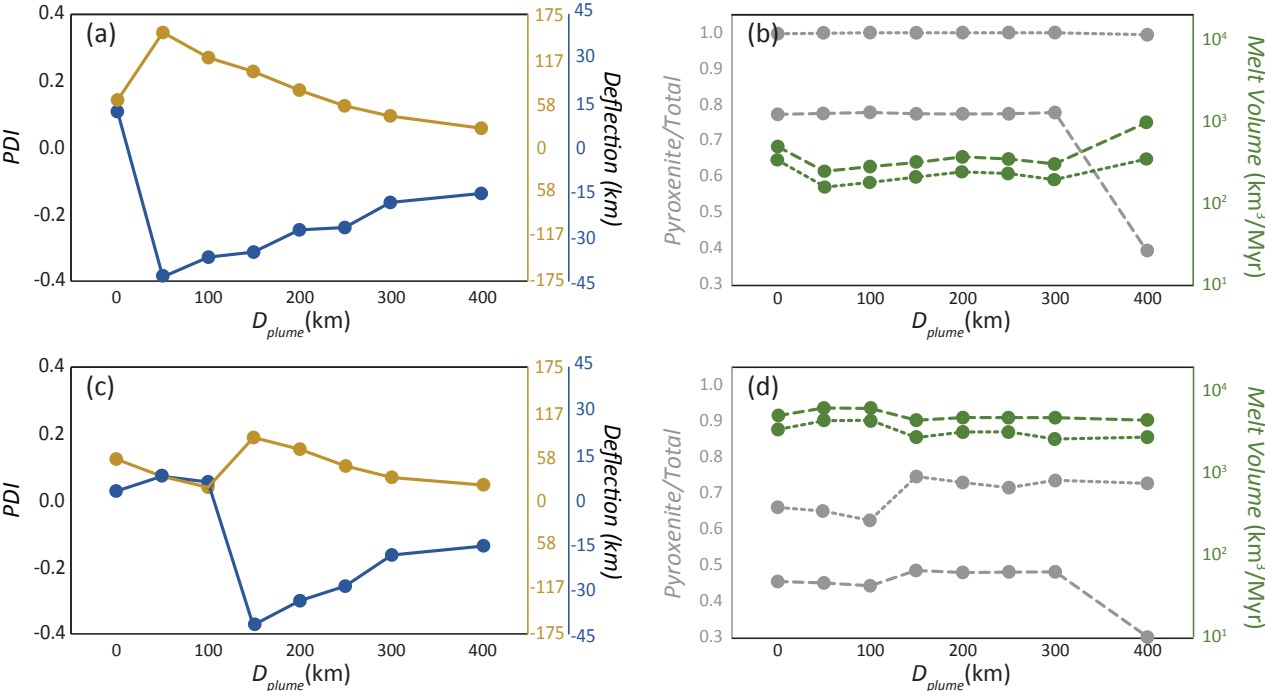

**Figure 8.** Diagrams showing the sensitivity of key output parameters to $D_{plume}$. (a,c) $PDI_{stem}$ (light brown) and $PDI_{pancake}$ (blue) in the
steady state for models with variable $D_{plume}$. (b,d) $M$, $V$, $\frac{M_{PX}}{M}$, and $\frac{V_{PX}}{V}$, for models with variable $D_{plume}$. In the top row, results for a
relatively weak plume with $\Delta T_{plume}$ = 150 °C and $B_{plume}$ = 100 kg·s$^{-1}$ are given. In the bottom row, results for with a relatively strong
plume with $\Delta T_{plume}$ = 200 °C and $B_{plume}$ = 200 kg·s$^{-1}$ are given. Legend as in Figures 4, 6.

Compared to those on PDI, the effects of $D_{plume}$ on melt fluxes and compositions are less severe. Figure 8b,d show the
trends of melting-related parameters as a function of $D_{plume}$ at the same scale than other figures (*e.g.*, Figure 6). The effects on
melt fluxes appear small, which is mostly due to the logarithmic scale of the figure; nonetheless several 'regimes' or different
behaviors can be distinguished on the basis of distance of the plume to the edge. Similar to the PDI figure, there is an initial
regime ($D_{plume}$ = 0 km for the cases with $B_{plume}$ = 100 kg·s$^{-1}$; and $D_{plume}$ = 0-100 km for the cases with $B_{plume}$ = 200
kg·s$^{-1}$) with lower PDI, and the melting volumes remain mainly flat (at least in Figure 8d). Then, at greater distances, EDC
interacts strongly with the plume, resulting in slightly lower melting volumes with a smooth peak around $D_{plume}$ = 200 km.
Finally, at $D_{plume}$ = 400 km, melting volumes increase substantially due to SSC, but volcanism remains practically the same
(suggesting that the main volume of melting still happens at the top of the plume conduit).

In general, plume deflection, as caused by the effects of EDC, tends to systematically decrease the amounts of hotspot
magmatism for a given plume vigor/temperature. The least negative (or most constructive) plume-EDC interaction occurs near

$D_{plume}$=0 km and $D_{plume}$=200 km. These locations roughly reflect the intrinsic pattern of the EDC-related and neighboring "triggered" SSC-related upwellings, as predicted by the companion paper, but not exactly so. The differences being likely due to the effect of the presence of a plume on EDC and SSC patterns. That the distance of the change of regime (from little to strong) of influence of EDC on the plume depends on plume vigor is also related to the effects of the plume (and plume pancake) on the wavelength of EDC. Thus, EDC appears to affect plume ascent and vice-versa.

### 3.4 Effects of mantle viscosity

Finally, we explore models with different reference viscosities. Figure 9 shows data for cases with variable viscosity, $D_{plume}$ = 200 km and $D_{plume}$ = 300 km, and otherwise the same parameters as in the reference case. Similar to the effects of plume temperature, the width of the plume stem is implicitly smaller with decreasing reference viscosity. At $D_{plume}$ = 200 km, one of the most striking characteristics of these cases is that the deflection of the plume stem is less severe for the high-viscosity and the low-viscosity case than for the intermediate-viscosity case, shown in Figure 7c (same $D_{plume}$ and same $B_{plume}$, but intermediate $\eta_0$). Indeed, PDI indexes (fig. 9a) display a maximum in terms of plume deflection for the intermediate viscosity value of the reference case $\eta_0$ = 8.29·$10^{18}$ Pa·s (Figure 3).

However, at $D_{plume}$ = 300 km the trends in Figure (9) depict a more systematic behavior with cases with lower viscosity featuring higher deflection. The differences between the cases at $D_{plume}$ = 200 and 300 km may be analogue to the differences between cases with different $B_{plume}$, in which the critical $D_{plume}$ where $PDI_{pancake}$ switches from positive to negative is increased with increasing buoyancy flux. However, note that in this case, the change is not from negative to positive $PDI_{pancake}$. Instead, we find that the greater spread of the plume pancake in cases with lower reference viscosity (Figure 10) makes the plume pancake to be affected by the topography of the edge, decreasing $PDI_{pancake}$ at $D_{plume}$ smaller than 300 km.

Since the vigor of EDC decreases with increasing $\eta_0$ (Sleep, 2007; Till et al., 2010; Davies and Rawlinson, 2014; Manjón-Cabeza Córdoba and Ballmer, 2021), EDC-plume interaction also becomes less important. This is evident at higher viscosities, where both $PDI_{stem}$ and $PDI_{pancake}$ decrease (Figure 9).

The position of the main EDC downwelling is weakly affected by changing viscosity. However, the relationship between the main EDC downwelling and the PET changes substantially. In cases with decreased viscosity (Figure 10), the PET expands, and the continent-side limit of the dynamic PET crosses the main downwelling of EDC, which is not observed in any of our other models (*i.e.*, changing any other property). This implies that in models with low viscosity the main EDC downwelling is situated inside the PET. We also find that the symmetry of the PET with respect to the vector of plate velocity is higher for the cases with low viscosity than for the cases with intermediate and high viscosity. In other words, the PET is more asymmetric (continent-ward) even if the plume pancake features small PDIs. These model predictions are explained by the very low viscosity of the pancake (in cases with low $\eta_0$), which promotes spreading independent of nearby features, such as the edge.

In turn, model predictions in terms of melting as a function of $\eta_0$ are as expected. Both $M$ and $V$ increase with decreasing $\eta_0$ and, along with this decrease, $\frac{M_{PX}}{M}$ and $\frac{V_{PX}}{V}$ decrease (Figure 9b). In addition to plume-related hotspot melting, melting away

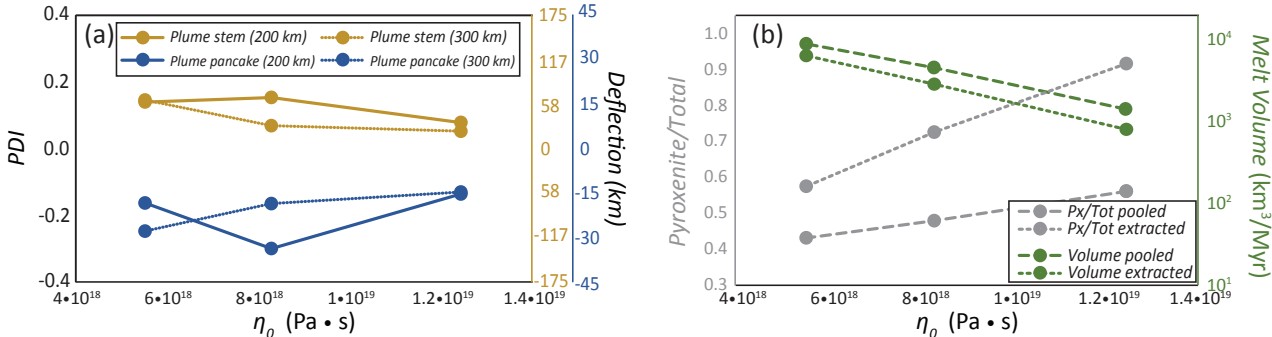

**Figure 9.** Diagrams showing the sensitivity of selected output parameters to $\eta_0$ in the steady state. Note that the PDI change is different for different $D_{plume}$

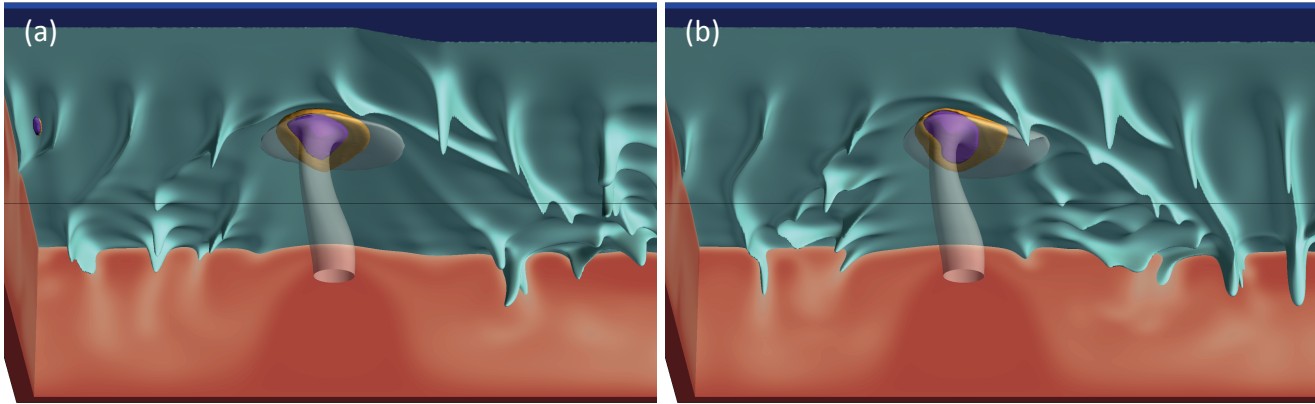

**Figure 10.** Steady-state model snapshot for the low viscosity cases ($\eta_0 = 5.53 \times 10^{18}$ Pa·s). (a) Case with $D_{plume}$ = 200 km and $B_{plume}$ = kg·s$^{-1}$. (b) Case with $D_{plume}$ = 300 km and $B_{plume}$ = kg·s$^{-1}$. Note that the deflection of the pancake at 300 km is greater than that of 200 km (Figure 9) as opposed to the case for intermediate and high viscosity.

from the hotspot (*i.e.* directly due to EDC or SSC) appears for some cases with low $\eta_0$ . This melting is minor and consistent with the low $\eta_0$ cases of the companion paper (Manjón-Cabeza Córdoba and Ballmer, 2021).

## 4 Discussion

We ran a wide range of 3D numerical models to systematically explore the interaction between EDC and mantle plumes. One of our main results is that the plume geometry, its interaction with the lithosphere, and the extent of related melting depends

on the distance of the plume from the edge, being altered by EDC. Despite these important effects, the buoyancy flux of the plume still remains the main influence on the characteristics of plume-lithosphere interaction and hotspot magmatism.

We quantify the deflection of plumes by two characteristic parameters: the deflection of the plume stem, and the deflection of the shallow plume conduit and the plume pancake. The plume stem is systematically deflected away from the edge. This may provide an explanation as to why hotspot tracks in the Atlantic preferentially occur near and sub-parallel to the continental margin, but rarely across it (an exception to this is the Cameroon Volcanic Line; Fitton, 1980; Déruelle et al., 2007). On the other hand, plume pancake deflection commonly (but not always) occurs towards the edge. This prediction may explain why some hotspot tracks (such as the Canaries) do not strictly align with plate velocity, and volcanism is widespread with more activity far from the continental margin than near to it (*e.g.*, La Palma vs. Gran Canaria).

The nature of the deflection of the plume pancake (and as well of the plume stem) systematically changes in our models with the distance of the plume to the edge. However, this deflection is generally predicted to decrease with increasing $B_{plume}$ (fig. 6a,c) relative to the EDC flux. Test cases with a greater step of lithospheric thickness at the continental margin confirm this prediction (Appendix, Figure A5). Such a configuration implies greater downwelling fluxes of EDC (see Manjón-Cabeza Córdoba and Ballmer, 2021), and leads to greater deflections of the plume and more asymmetric plume-lithosphere interaction than the reference case. In terms of PDI, the absolute value of $PDI_{stem}$ is 0.12, as opposed to 0.10 in the case with regular edge and similar parameters (fig. 7c); and $PDI_{pancake}$ is -0.07, as opposed to -0.05. A higher value for both indices suggests that the plume-EDC interaction depends on the ratio between the downward flux of materials due to EDC and the upward flux of materials due to plume activity.

The dependence of plume-EDC interaction to these two fluxes is consistent with previous work such as Ballmer et al. (2011), who found that, even for high plume fluxes, sublithospheric convective instabilities can have an effect on the surface expression of mantle plumes. In turn, our results challenge the opinion that strong external fluxes will overprint or even ignore EDC (King and Anderson, 1998; Till et al., 2010; Kaislaniemi and Van Hunen, 2014), a statement that may only hold true for high buoyancy flux plumes.

We find that weak-to-intermediate plumes can be strongly deflected by EDC with significant effects on plume-related magmatism (Figures 7, 8). Plumes can be laterally deflected by >100 km (PDI > 0.2) as a function of plume-EDC geometrical configuration. While the specific $D_{plumes}$ for which maximum plume deflection occurs depends on plume flux and astenospheric viscosity, these extents of deflection are of the same order than the apparent displacements of the location of the main EDC downwelling.

There is a clear difference in the position of the main EDC downwelling between cases with and without plume (see Figure A1, and Manjón-Cabeza Córdoba and Ballmer, 2021). The downwelling is systematically pushed continent-wards near the hotspot and also further downstream. Nonetheless, this displacement is only influenced little by changing plume properties or the rheology of the mantle. Of the parameters studied, only $B_{plume}$ efficiently affects the position of the main EDC downwelling, (see also Burov et al., 2007). More surprisingly, the main EDC cell is not strongly bound to the plume flow downstream of the hostpot (and the PET), as evidenced by our low $\eta_0$ models. On the contrary, we find a high coupling between the plume pancake and stem, and the EDC cell near the hotspot. Overall, this evidence points towards a greater effect

of EDC on plumes than vice-versa. However, this affirmation is likely dependent of parameter choices choices (such as plume buoyancy flux), as well as initial conditions (such as edge geometry).

    The presence of the plume can also affect the vigor of EDC. In particular, the vertical component of the main EDC down-welling is faster in the cases with negative $PDI_{pancake}$ than in cases with positive $PDI_{pancake}$. For example, the reference case shows a vertical velocity component of up to $\sim$1.3 cm·yr$^{-1}$ near the plume pancake, while an equivalent case of $D_{plume}$
= 200 km features a vertical velocity component of $\sim$1.5 cm·yr$^{(-1)}$. This highlights again the ffects of constructive vs. destructive plume-EDC interaction as a function of geometry (or plume-edge distance).

    Our models stand in contrast to other studies that focused on very high buoyancy flux plumes and plume arrival (plume heads). For example, Burov et al. (2007) and François et al. (2018) observed a deflection of the plume head ocean-wards, not unlike our models with $D_{plume}$ = 0 km. In such a scenario, the plume completely disrupts the EDC cell (favoring craton
removal), and plume flow is conditioned by the topography of the lithosphere. For very strong plumes and plume heads, it is likely that plume flow dominates over EDC. Concerning the influence of the plume head, our models are not suited to evaluate its influence because they are focused on a steady-state plume setting (Ribe and Christensen, 1999). Moreover, various studies have shown that different lithospheric strength models will interact with different kinds of plume heads in radically different ways everything else being equal (Gerya et al., 2015; Koptev et al., 2021). It is unlikely, however, that the plume head of the
Canaries still bears influence on the Archipelago since the hotspot track is >60 Ma old.

    In contrast to the EDC-only cases in the first part of our work (Manjón-Cabeza Córdoba and Ballmer, 2021), volcanic (or melt) volume fluxes are significantly higher, displaying strong variations as a function of plume parameters and moderate variations as a function of $D_{plume}$. Therefore, a subset of our models can account for the volumes of Eastern Atlantic hotspots. However, both $B_{plume}$ and $\Delta T_{plume}$ also affect the geochemistry of the melts. In our models, the composition of melting due
to plume+EDC is generally less enriched than for melting due to EDC-only (Manjón-Cabeza Córdoba and Ballmer, 2021). Any increase in volcanism due to higher $B_{plume}$ or $\Delta T_{plume}$ is associated with a decrease in enrichment (*i.e.*, the fraction of melting products from enriched lithologies such as pyroxenite, Figures 4b, 6b,d), with only minor effects on geochemical proxies as a function of $D_{plume}$ (fig. 8).

    In the companion paper (Manjón-Cabeza Córdoba and Ballmer, 2021), we clearly showed that EDC alone is insufficient
to generate the Canary Islands magmatism, and that the contribution from a mantle plume (or equivalent source) is required. The Canary Islands feature two islands in shield-building stage: for El Hierro, Carracedo et al. (1998) estimates a minimum volcanic flux of $4\times10^{-1}$ km$^3$ kyr$^{-1}$; for La Palma, the inferred volcanic rates are around 1 km$^3$ kyr$^{-1}$ (Day et al., 1999). For the whole archipelago, independent estimates place extrusion volumes at 1-10 km$^3$ kyr$-1$, depending on whether the 18$^{th}$ century Timanfaya eruption(s) are considered an anomalous event (Longpré and Felpeto, 2021). These numbers can easily
be doubled when considering underplating and plutonism beneath the islands (Klügel et al., 2005). Overall, these values are higher than any published estimate of EDC-generated volume, including work that considered additional geometric complexity for the oceanic-continent transition (Duvernay et al., 2021; Negredo et al., 2022).

    Our models predict that relatively weak plumes with parameters similar to that of the reference case are sufficient to generate these amounts of magmatism. However, it remains difficult to pinpoint plume parameters, *e.g.*, as plume temperature

and buoyancy flux trade off with each other (Figures 4b, 6b,d). This result implies that the Canary plume must be of either medium/low flux (*i.e.*, $B_{plume} < 200$ kg s$^{-1}$), or medium/low temperature (*i.e.*, $\Delta T_{plume} < 200$ °C), but not both, as the melt volumes would be too low then (*i.e.* $< 0.1$ km$^3$ kyr$^{-1}$). Note that $M$ and $V$ in the relevant model cases are of the same order of magnitude, suggesting that our results are robust despite model limitations and simplifications in terms of modeling mantle melting and extraction. Our predictions for $\frac{M_{PX}}{M}$ and $\frac{V_{PX}}{V}$ further constrain the properties of the plume: occurrence of

shield-like magmas in the Canary Islands (Abdel-Monem et al., 1971, 1972; Carracedo et al., 1998) strengthen the suggestion that $B_{plume}$ and $\Delta T_{plume}$ cannot (both) be lower than for our reference case, as this would generate melts that are too enriched, *i.e.* with PX contributions much higher that inferred from petrological work (Day et al., 2009; Day and Hilton, 2011). More likely, one of these parameters (or both) must be slightly higher than in the reference case.

The distance of the Canary hotspot from the African passive margin is ~250-300 km. Considering the PDI values predicted

for relatively weak plumes (Figure 8b), we estimate that the Canary plume at 660 km depth is centered ~50-100 km closer to the African margin than the hotspot (which is located near El Hierro). Likewise, we estimate that the plume is at 410 km depth is centered ~30-70 km closer to the margin. This prediction is consistent with the receiver functions study of Saki et al. (2015): the location of the shallowest 410 km discontinuity is shifted from the hotspot at El Hierro eastward towards Lanzarote. In adition, the recent tomographic study by Civiero et al. (2021) shows an arcuate upper mantle plume beneath the Canaries not unlike the plumes from our study. Attending to this evidence, we estimate that the cases with $150 \leq D_{plume} \leq 250$ km

best match the configuration of the Canary plume. This finding implies that plume-EDC interaction (*e.g.*, as quantified by PDI values) is significant for the Canary hotspot. The plume stem is pushed to the west (away from the edge) by about 80-110 km and the plume pancake is pulled back to the east (towards the edge) by about 25-35 km. If the volcanic flux at the Canaries is significantly higher than estimated by Carracedo et al. (1998), *e.g.* due to un-accounted magmatic crustal underplating, we

reach the same conclusions, predicting very similar PDI values (Figure 8c).

In addition, we find that several key characteristics of the Canary Islands are matched by our models. The Canaries present active volcanism far from the inferred deflection point of the plume stem near El Hierro (in fact, all islands are currently active with the exception of La Gomera; Abdel-Monem et al., 1971, 1972; Carracedo, 1999; Geldmacher et al., 2005). Several of our cases predict deflection of the plume pancake and the melting zones toward the continental margin (Figure 5), including the

cases with $150 \leq D_{plume} \leq 250$, which would explain the shape of the whole archipelago and the geographic distribution of volcanism. Even with a plate velocity that would produce a volcanic track parallel to the ocean-continent transition, given the right distances to the edge, the plume pancake may not necessarily be parallel to the plate movement.

While our models are able to explain coeval volcanism across the islands, they do not reproduce the general westward progression of the main shield stage over time (Geldmacher et al., 2005). Indeed, the Canary hotspot may have moved in

the last few million years westward with respect to the African margin (Wang et al., 2018). Accordingly, the distance of the hotspot relative to the African plate may have changed, rendering plume-EDC and plume-lithosphere interaction a transient phenomenon, which cannot be explicitly adressed by our steady-state model setting. However, Figures 7 and 8 provide an indication of how the geometry of the plume and plume-lithosphere interaction may have changed during such a movement. As the plume moved away from the margin, the effects of changes in plume-EDC interaction may have extended and deflected

the pancake and, therefore, extended the area of volcanism from a single track to a wide zone, consistent with Geldmacher et al. (2005).

Recently, Negredo et al. (2022) provided a mechanism by which plume migration due to EDC can also occur. While the deflection of the plume pancake continent-ward is not predicted by their models, it is possible that a combination of (theirs and ours) phenomena can explain the history of the Canaries. Our model predicts that, eventually, further movement of the

plume away from the edge may decrease the extent of plume deflection. Alternatively, if the vigor of the plume (or of EDC) has recently changed, plume-EDC interaction and plume displacement would have also changed, and the recent movement of the hotspot relative to the African margin would be potentially unrelated to any movement of the deep plume stem or even the plate movement.

The application of our models to other hotspots in the Eastern Atlantic is less obvious. Lodhia et al. (2018) and King and

Ritsema (2000) have suggested a link between the Cape Verde plume and the downwelling at the African Margin near Cape Verde. From our models, however, a significant effect of EDC on plume ascent over such long distances (over 1000 km) is not justified. It is true that higher mantle viscosities may result in larger EDC cells, but it will also result in lower EDC-related fluxes as already shown in Figure 5a of Manjón-Cabeza Córdoba and Ballmer (2021). Regardless, the models presented here include a plate velocity that is not fully consistent with the Cape Verde 'near-zero' plate velocity. In fact, Patriat and Labails (2006)

detected a "bulge" or "bum" along the continental-oceanic transition between the Canary Islands and Cape Verde. Whether this "bulge" is related to an EDC upwelling is difficult to determine, but a topography high is expected in the area of maximum sub-lithospheric erosion above an EDC upwelling (Manjón-Cabeza Córdoba and Ballmer, 2021). Such a relationship is also consistent with lithospheric models that detect EDC-related erosion at the bottom of the lithosphere beneath the Canaries (see for instance Figure 7, model c2 of Fullea et al., 2015), which would imply that the main EDC upwelling happens very close

to the edge. Such a proximity is also suggested by our 3D models here. The aforementioned "bulge" is consistent with the location of the eastern islands of the Canary archipelago, but does not seem to be related directly to the Cape Verde hotspot.

The Cameroon volcanic line is very close to the continental margin and even crosses it, such that an influence of EDC is expected. From our models, it can be inferred that at least part of the volcanic 'track' with widespread volcanism perpendicular to the ocean-continent transition is consistent with plume-EDC interaction. However, the geometry of the edge near the

Cameroon volcanic line is considerably more complicated than in our models, and Duvernay et al. (2021) have shown that the patterns of volcanism due to EDC can change considerably with complex 3D configurations. To better understand this volcanic field, more specific work taking into consideration the shape of the ocean-continent transition in this region is required.

Regarding future work, our models can be expanded to address limitations. While we use a decreased activation energy to mimic dislocation creep, we expect the explicit effects of a composite rheology including non-Newtonian viscosity laws will

quantitatively modify our results. In particular, shear-thinning rheologies are expected to localize EDC flow and, potentially, plume ascent. In addition, the continental lithosphere in our models is not rheologically stabilized. Differences between models with and without rheological stabilization (*e.g.*, Kaislaniemi and Van Hunen, 2014; Manjón-Cabeza Córdoba and Ballmer, 2021) suggest that stabilization may decrease EDC downwelling flux, and protect the continental lithosphere against erosion.

As a consequence, the migration of the EDC main downwelling may decrease for a given lithospheric age; and $PDI$ absolute values decrease as well for a given $\tau_c$.

Our current models also rely on a simplified compositional approximation. In our work, we considered purely thermal plumes with a composition equal to the 'background' mantle. In turn, this is inconsistent with $CO_2$-rich volcanism in the Canary Islands (Allegre et al., 1971; Taracsák et al., 2019). However, explicitly modeling $CO_2$-related melting in geodynamic models is not possible to date. Moreover, the fraction of pyroxenite in the source is unlikely to be constant along the plume path (Day et al., 2010). Future models are needed to explore the transient changes along the Canary hotspot track.

Moreover, as said above, in this work we did not explore in depth geometrical considerations of the craton and the continental-ocean transition. In previous works, several authors noted the importance of this geometrical aspects on conditioning the mantle flow around the edge (Till et al., 2010; Kim and So, 2020; Manjón-Cabeza Córdoba and Ballmer, 2021; Duvernay et al., 2021). In this article, we focused on plume characteristics instead, but we acknowledge that the quantitative inferences made could change when changing craton characteristics, although preliminary tests confirm that results will be qualitatively consistent (Appendix, Figure A5).

One of the main characteristics of the Canary Islands hotspot is the near absence of a hotspot swell. While the PET provides a proxy for the potential location of the plume swell, analysis of dynamic topography using geodynamic models may answer whether deflection of the plume and the pancake by EDC can blur the dynamic-topography signal of the plume (Huppert et al., 2020). Another area of potential further work is to better constrain the geochemical fingerprint of our model magmas. In our models, $\Delta T_{plume}$ and $B_{plume}$ have very similar effects on the compositional proxies of volcanism used here ($\frac{M_{PX}}{M}$ and $\frac{V_{PX}}{V}$). However, as soon as several geochemical systems are considered (*e.g.*, major and trace elements, isotopes), the effects of $\Delta T_{plume}$ should have a distinct effect on the geochemistry of magmas from increasing $B_{plume}$. Unfortunately, additional assumptions in terms of starting composition of PX and peridotite (or their potential interactions Ballmer et al., 2013; Jones et al., 2017) are required to explicitly predict trace-element and isotopic signatures (Bianco et al., 2008). To date, no practical melting parameterization is available to realistically predict major element compositions from geodynamic models. Future work will focus on a new melting parameterization that can help to discriminate between parameters in this setting and other geodynamic models.

## 5 Conclusions

We studied the effects of Edge-Driven Convection (EDC) on low-to-intermediate buoyancy flux plumes. The following points summarize the main findings of this study:

- Low and intermediate buoyancy flux plumes interact with shallow mantle flow related to sub-lithospheric convective instability, which cause the plume to be deflected with important effects on the volume flux (and composition) of hotspot melting.

– The interaction of the plume with Edge-Driven Convection highly depends on the distance of the plume to the ocean-continent transition, but the distance for which EDC has the strongest influence varies with physical properties of the mantle and plumes. For example, weaker plumes (lower buoyancy flux, lower temperature) are most affected closer to the edge (*i.e.*, continental margin) than more vigorous plumes (higher buoyacy flux, higher temperature).

– The ratio of the buoyancy flux of the plume with respect to the flux of material from EDC is one of the most important
factors to control plume-EDC interaction at a given plume-edge distance, including deflection of the plume stem away from the continental edge, and of the pancake towards the edge.

– In the Canary Islands, a plume of low buoyancy flux and high temperature or, alternatively, a plume with moderate buoyancy flux and low temperature may be rising at 200 km from the continental margin, being deflected and creating the complex age progression and widespread volcanism.

*Code availability.* CITCOM CU (Moresi and Solomatov, 1995; Moresi and Gurnis, 1996; Zhong et al., 2000) is an open source code available at https://geodynamics.org/cig/software/citcomcu/. The modified version of CITCOM CU with the modifications described in the text is available at https://doi.org/10.5281/zenodo.4293656

**Appendix A:  Additional figures**

*Author contributions.* AMCC performed the numerical experiments and post-processed, analyzed and plotted the data. Both authors devised
the study, interpreted results and wrote the paper.

*Competing interests.* We decleare that we do not have any competing interest.

*Acknowledgements.* The authors want to thank reviewers Russel Pysklywec, Ana Negredo and Rhodri Davies for their thorough and useful comments. Esteban Gazel, Taras Gerya, Oli Shorttle and Paul Tackley provided insightful suggestions on an earlier version of the article. AMCC was funded by the Schweizerischer Nationalfonds zur Förderung der Wissenschaftlichen Forschung (grant no. 2-77026-16).

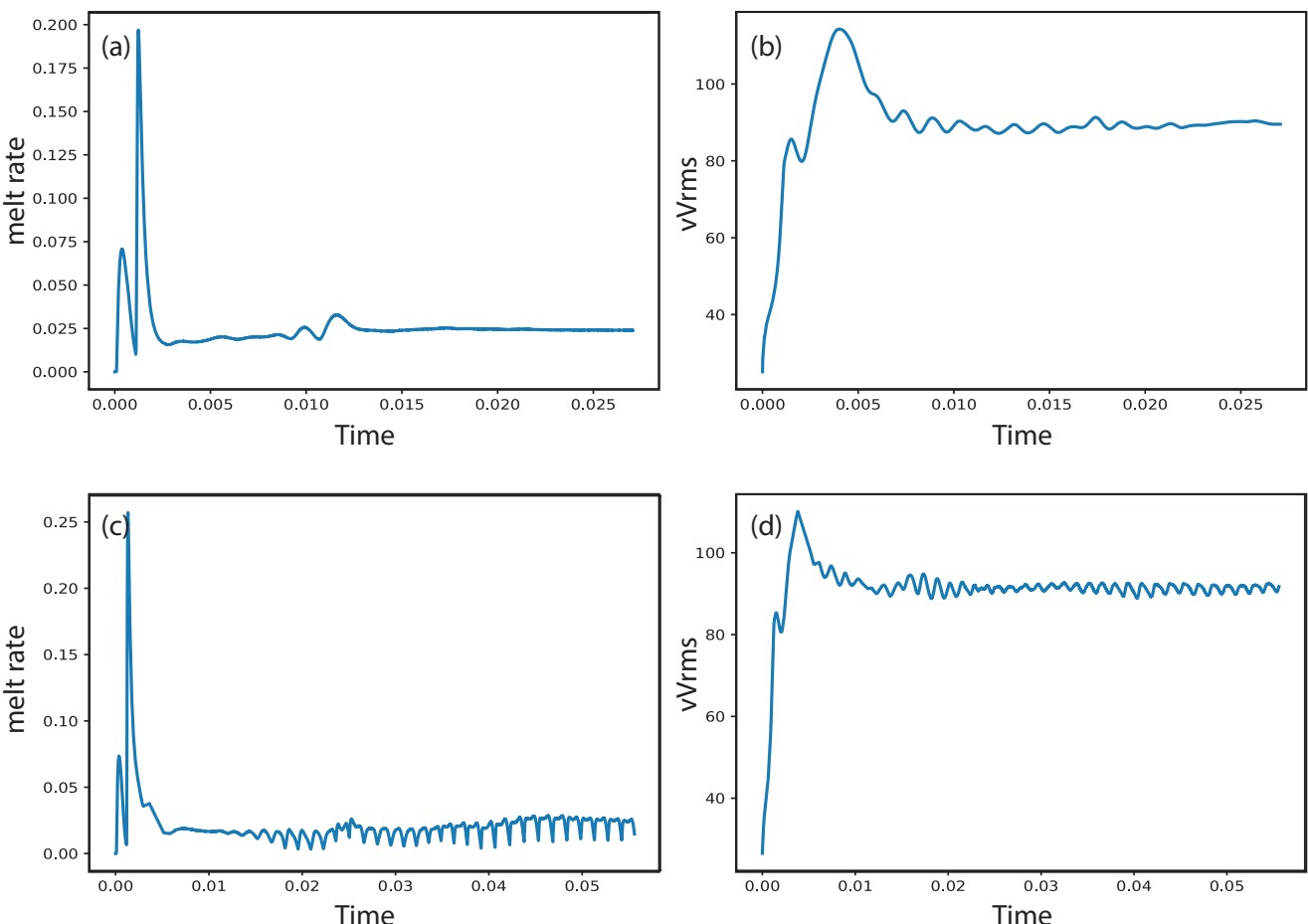

**Figure A1.** Example of trends used to evaluate steady state in our models. Shown are the reference case (panels a and b, see also Figure 3), and the case of $D_{plume}$ = 400 km, $\Delta T_{plume}$ = 150 °C and $B_{plume}$ = 100 kg·s$^{-1}$ (panels c and d, see also Figure 7b and supplementary video). Panels (a) and (c) show the total melting rate (integrated across the whole model) and panels (b) and (d) show the root mean square of the vertical velocity (vVrms). All axes are in non-dimensional scale.

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

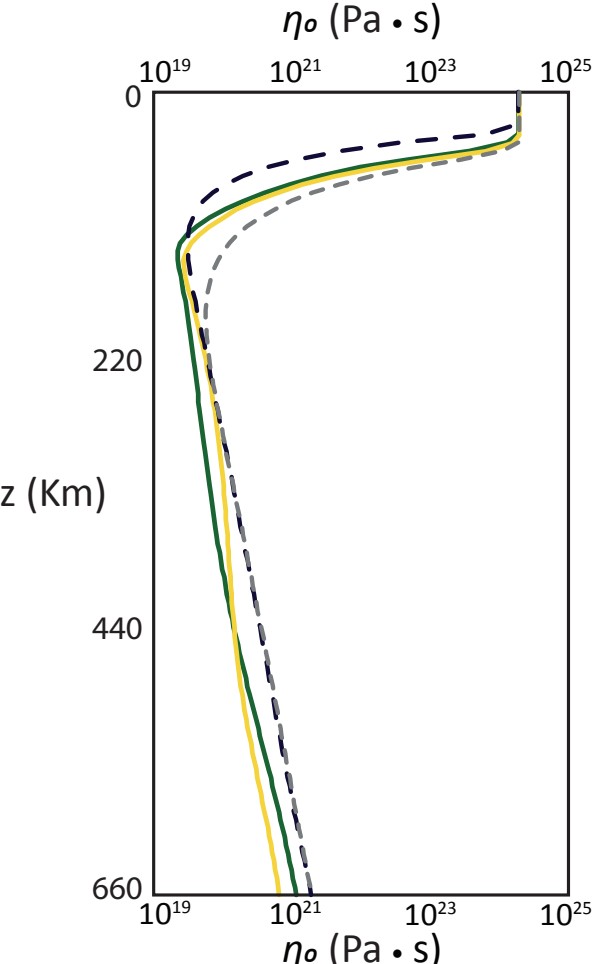

**Figure A2.** Viscosity profiles representative of the reference case (Figure 3). In dashed lines, viscosity profiles for the inflow boundary conditions: in black, oceanic side; in grey, continental side (see also Manjón-Cabeza Córdoba and Ballmer, 2021). In solid lines, two representative viscosity profiles of the plume at $x = 1254$ km (green line) and $x = 1287$. Note that due to the plume deflection, a single vertical profile representing the whole plume is not possible (see also Figure A4).

Afonso, J. C., Rawlinson, N., Yang, Y., Schutt, D. L., Jones, A. G., Fullea, J., and Griffin, W. L.: 3-D multiobservable probabilistic inversion for the compositional and thermal structure of the lithosphere and upper mantle: III. Thermochemical tomography in the Western-Central U.S., Journal of Geophysical Research: Solid Earth, 121, 7337–7370, https://doi.org/10.1002/2016JB013049, 2016.

Allegre, C. J., Pineau, F., Bernat, M., and Javoy, M.: Evidence for the Occurrence of Carbonatites on the Cape Verde and Canary Islands, Nature, 233, 103–104, 1971.

Ballmer, M. D., van Hunen, J., Ito, G., Tackley, P. J., and Bianco, T. A.: Non-hotspot volcano chains originating from small-scale sublithospheric convection, Geophysical Research Letters, 34, https://doi.org/10.1029/2007GL031636, 2007.

Ballmer, M. D., van Hunen, J., Ito, G., Bianco, T. A., and Tackley, P. J.: Intraplate volcanism with complex age-distance patterns: A case for
small-scale sublithospheric convection, Geochemistry, Geophysics, Geosystems, 10, https://doi.org/10.1029/2009GC002386, 2009.

Ballmer, M. D., Ito, G., Van Hunen, J., and Tackley, P. J.: Spatial and temporal variability in Hawaiian hotspot volcanism induced by small-scale convection, Nature Geoscience, 4, 457–460, https://doi.org/10.1038/ngeo1187, 2011.

Ballmer, M. D., Ito, G., Wolfe, C. J., and Solomon, S. C.: Double layering of a thermochemical plume in the upper mantle beneath Hawaii, Earth and Planetary Science Letters, 376, 155–164, https://doi.org/10.1016/j.epsl.2013.06.022, 2013.

Bianco, T. A., Ito, G., van Hunen, J., Ballmer, M. D., and Mahoney, J. J.: Geochemical variation at the Hawaiian hot spot caused by upper mantle dynamics and melting of a heterogeneous plume, Geochemistry, Geophysics, Geosystems, 9, Q11 003, https://doi.org/10.1029/2008GC002111, 2008.

Boutilier, R. R. and Keen, C. E.: Small-scale convection and divergent plate boundaries, Journal of Geophysical Research: Solid Earth, 104, 7389–7403, https://doi.org/10.1029/1998JB900076, 1999.

Buck, R. W.: Small-scale convection induced by passive rifting: the cause for uplift of rift shoulders, Earth and Planetary Science Letters, 77, 362–372, https://doi.org/10.1016/0012-821X(86)90146-9, 1986.

Burov, E., Guillou-Frottier, L., D'Acremont, E., Le Pourhiet, L., and Cloetingh, S.: Plume head-lithosphere interactions near intra-continental plate boundaries, Tectonophysics, 434, 15–38, https://doi.org/10.1016/j.tecto.2007.01.002, 2007.

Carracedo, J. C.: Growth, structure, instability and collapse of Canarian volcanoes and comparisons with Hawaiian volcanoes, Journal of
Volcanology and Geothermal Research, 94, 1–19, https://doi.org/10.1016/S0377-0273(99)00095-5, 1999.

Carracedo, J. C., Day, S., Guillou, H., Rodríguez Badiola, E., Canas, J. A., and Pérez Torrado, F. J.: Hotspot volcanism close to a passive continental margin: the Canary Islands, Geological Magazine, 135, 591–604, https://www.cambridge.org/core/journals/geological-magazine/article/hotspot-volcanism-close-to-a-passive-continental-margin-the-canary-islands/26D6FD3C9CA2DAC201366C987DBAFC31, 1998.

Christensen, U. R.: Convection with pressure- and temperature-dependent non-Newtonian rheology, Geophysical Journal of the Royal Astronomical Society, 77, 343–384, https://doi.org/10.1111/j.1365-246X.1984.tb01939.x, 1984.

Christensen, U. R. and Yuen, D. A.: Layered convection induced by phase transitions, Journal of Geophysical Research, 90, 10 291, https://doi.org/10.1029/JB090iB12p10291, 1985.

Civiero, C., Custódio, S., Neres, M., Schlaphorst, D., Mata, J., and Silveira, G.: The Role of the Seismically Slow Central-
East Atlantic Anomaly in the Genesis of the Canary and Madeira Volcanic Provinces, Geophysical Research Letters, 48, 1–15, https://doi.org/10.1029/2021GL092874, 2021.

Conrad, C. P., Wu, B., Smith, E. I., Bianco, T. A., and Tibbetts, A.: Shear-driven upwelling induced by lateral viscosity variations and asthenospheric shear: A mechanism for intraplate volcanism, Physics of the Earth and Planetary Interiors, 178, 162–175, https://doi.org/10.1016/J.PEPI.2009.10.001, 2010.

Courtillot, V., Davaille, A., Besse, J., and Stock, J.: Three distinct types of hotspots in the Earth's mantle, Earth and Planetary Science Letters, 205, 295–308, https://doi.org/10.1016/S0012-821X(02)01048-8, 2003.

Davies, D. R. and Davies, J. H.: Thermally-driven mantle plumes reconcile multiple hot-spot observations, Earth and Planetary Science Letters, 278, 50–54, https://doi.org/10.1016/j.epsl.2008.11.027, 2009.

Davies, D. R. and Rawlinson, N.: On the origin of recent intraplate volcanism in Australia, Geology, 42, 1031–1034, https://doi.org/10.1130/G36093.1, 2014.

Day, J. M. and Hilton, D. R.: Origin of 3He/4He ratios in HIMU-type basalts constrained from Canary Island lavas, Earth and Planetary Science Letters, 305, 226–234, https://doi.org/10.1016/j.epsl.2011.03.006, 2011.

Day, J. M., Pearson, D. G., Macpherson, C. G., Lowry, D., and Carracedo, J. C.: Pyroxenite-rich mantle formed by recycled oceanic lithosphere: Oxygen-osmium isotope evidence from Canary Island lavas, Geology, 37, 555–558, https://doi.org/10.1130/G25613A.1, 2009.

Day, J. M., Pearson, D. G., Macpherson, C. G., Lowry, D., and Carracedo, J. C.: Evidence for distinct proportions of subducted oceanic crust and lithosphere in HIMU-type mantle beneath El Hierro and La Palma, Canary Islands, Geochimica et Cosmochimica Acta, 74, 6565–6589, https://doi.org/10.1016/j.gca.2010.08.021, 2010.

Day, S. J., Carracedo, J. C., Guillou, H., and Gravestock, P.: Recent structural evolution of the Cumbre Vieja volcano, La Palma, Canary Islands: Volcanic rift zone reconfiguration as a precursor to volcano flank instability?, Journal of Volcanology and Geothermal Research, 94, 135–167, https://doi.org/10.1016/S0377-0273(99)00101-8, 1999.

Déruelle, B., Ngounouno, I., and Demaiffe, D.: The 'Cameroon Hot Line' (CHL): A unique example of active alkaline intraplate structure in both oceanic and continental lithospheres, Comptes Rendus - Geoscience, 339, 589–600, https://doi.org/10.1016/j.crte.2007.07.007, 2007.

Doblas, M., Lo´pez-Ruiz, J., and Cebria´, J.-M.: Cenozoic evolution of the Alboran Domain: A review of the tectonomagmatic models, in: Cenozoic Volcanism in the Mediterranean Area, vol. 418, pp. 303–320, Geological Society of America, https://doi.org/10.1130/2007.2418(15), 2007.

Duggen, S., Hoernle, K. A., Hauff, F., Klügel, A., Bouabdellah, M., and Thirlwall, M. F.: Flow of Canary mantle plume material through a subcontinental lithospheric corridor beneath Africa to the Mediterranean, Geology, 37, 283–286, https://doi.org/10.1130/G25426A.1, 2009.

Dumoulin, C., Doin, M. P., Arcay, D., and Fleitout, L.: Onset of small-scale instabilities at the base of the lithosphere: Scaling laws and role of pre-existing lithospheric structures, Geophysical Journal International, 160, 345–357, https://doi.org/10.1111/j.1365-246X.2004.02475.x, 2005.

Duvernay, T., Davies, D. R., Mathews, C. R., Gibson, A. H., and Kramer, S. C.: Linking Intra-Plate Volcanism to Lithospheric Structure and Asthenospheric Flow, Geochemistry, Geophysics, Geosystems, i, e2021GC009 953, https://doi.org/10.1029/2021GC009953, 2021.

Fitton, J. G.: The Benue Trough and Cameroon Line - A migrating rift system in West Africa, Earth and Planetary Science Letters, 51, 132–138, 1980.

François, T., Koptev, A., Cloetingh, S., Burov, E., and Gerya, T.: Plume-lithosphere interactions in rifted margin tectonic settings: Inferences from thermo-mechanical modelling, Tectonophysics, 746, 138–154, https://doi.org/10.1016/j.tecto.2017.11.027, 2018.

French, S. W. and Romanowicz, B.: Broad plumes rooted at the base of the Earth's mantle beneath major hotspots, Nature, 525, 95–99, https://doi.org/10.1038/nature14876, 2015.

Fullea, J., Camacho, A. G., Negredo, A. M., and Fernández, J.: The Canary Islands hot spot: New insights from 3D coupled geophysical-petrological modelling of the lithosphere and uppermost mantle, Earth and Planetary Science Letters, https://doi.org/10.1016/j.epsl.2014.10.038, 2015.

Geldmacher, J., Hoernle, K., Bogaard, P. V., Duggen, S., and Werner, R.: New $^{40}/Ar^{39}Ar$ age and geochemical data from seamounts in the Canary and Madeira volcanic provinces: Support for the mantle plume hypothesis, Earth and Planetary Science Letters, 237, 85–101, https://doi.org/10.1016/j.epsl.2005.04.037, 2005.

Gerya, T. V., Stern, R. J., Baes, M., Sobolev, S. V., and Whattam, S. A.: Plate tectonics on the Earth triggered by plume-induced subduction initiation, Nature, 527, 221–225, https://doi.org/10.1038/nature15752, 2015.

Helffrich, G., Faria, B., Fonseca, J. F., Lodge, A., and Kaneshima, S.: Transition zone structure under a stationary hot spot: Cape Verde, Earth and Planetary Science Letters, 289, 156–161, https://doi.org/10.1016/j.epsl.2009.11.001, 2010.

Hirschmann, M. M. and Stolper, E. M.: A possible role for garnet pyroxenite in the origin of the "garnet signature" in MORB, Contributions to Mineralogy and Petrology, 124, 185–208, https://doi.org/10.1007/s004100050184, 1996.

Huang, J., Zhong, S. J., and van Hunen, J.: Controls on sublithospheric small-scale convection, Journal of Geophysical Research, 108, https://doi.org/10.1029/2003JB002456, 2003.

Huppert, K. L., Perron, J. T., and Royden, L. H.: Hotspot swells and the lifespan of volcanic ocean islands, Science Advances, 6, eaaw6906, https://doi.org/10.1126/sciadv.aaw6906, 2020.

Jones, T. D., Davies, D. R., Campbell, I. H., Iaffaldano, G., Yaxley, G., Kramer, S. C., and Wilson, C. R.: The concurrent emergence and causes of double volcanic hotspot tracks on the Pacific plate, Nature, 545, 472–476, https://doi.org/10.1038/nature22054, 2017.

Kaislaniemi, L. and Van Hunen, J.: Dynamics of lithospheric thinning and mantle melting by edge-driven convection: Application to Moroccan Atlas mountains, Geochemistry, Geophysics, Geosystems, 15, 3175–3189, https://doi.org/10.1002/2014GC005414, 2014.

Katz, R. F., Spiegelman, M., and Langmuir, C. H.: A new parameterization of hydrous mantle melting, Geochemistry, Geophysics, Geosystems, 4, https://doi.org/10.1029/2002GC000433, 2003.

Kim, D.-H. and So, B.-D.: Effects of rheology and mantle temperature structure on edge-driven convection: Implications for partial melting and dynamic topography, Physics of the Earth and Planetary Interiors, 303, 106 487, https://doi.org/10.1016/j.pepi.2020.106487, 2020.

King, S. D. and Adam, C.: Hotspot swells revisited, Physics of the Earth and Planetary Interiors, 235, 66–83, https://doi.org/10.1016/j.pepi.2014.07.006, 2014.

King, S. D. and Anderson, D. L.: An alternative mechanism of flood basalt formation, Earth and Planetary Science Letters, 136, 269–279, https://doi.org/10.1016/0012-821X(95)00205-Q, 1995.

King, S. D. and Anderson, D. L.: Edge-driven convection, Earth and Planetary Science Letters, 160, 289–296, https://doi.org/10.1016/S0012-821X(98)00089-2, 1998.

King, S. D. and Ritsema, J.: African Hot Spot Volcanism: Small-Scale Convection in the Upper Mantle Beneath Cratons, Science, 290, 1137–1139, http://science.sciencemag.org/, 2000.

Klügel, A., Hansteen, T. H., and Galipp, K.: Magma storage and underplating beneath Cumbre Vieja volcano, La Palma (Canary Islands), Earth and Planetary Science Letters, 236, 211–226, https://doi.org/10.1016/j.epsl.2005.04.006, 2005.

Kohlstedt, D. L. and Hansen, L. N.: Constitutive Equations, Rheological Behavior, and Viscosity of Rocks, in: Treatise on Geophysics: Second Edition, vol. 2, pp. 441–472, Elsevier B.V., https://doi.org/10.1016/B978-0-444-53802-4.00042-7, 2015.

Koptev, A., Cloetingh, S., and Ehlers, T. A.: Longevity of small-scale ('baby') plumes and their role in lithospheric break-up, Geophysical Journal International, 227, 439–471, https://doi.org/10.1093/gji/ggab223, 2021.

Korenaga, J. and Jordan, T. H.: Linear stability analysis of Richter rolls, Geophysical Research Letters, 30, https://doi.org/10.1029/2003GL018337, 2003.

Ligi, M., Bonatti, E., Tontini, F. C., Cipriani, A., Cocchi, L., Schettino, A., Bortoluzzi, G., Ferrante, V., Khalil, S., Mitchell, N. C., and Rasul, N.: Initial burst of oceanic crust accretion in the Red Sea due to edge-driven mantle convection, Geology, 39, 1019–1022, https://doi.org/10.1130/G32243.1, 2011.

Liu, X. and Zhao, D.: Seismic evidence for a mantle plume beneath the Cape Verde hotspot, International Geology Review, 56, 1213–1225, https://doi.org/10.1080/00206814.2014.930720, 2014.

Lodhia, B. H., Roberts, G. G., Fraser, A. J., Fishwick, S., Goes, S., and Jarvis, J.: Continental margin subsidence from shallow mantle convection: Example from West Africa, Earth and Planetary Science Letters, 481, 350–361, https://doi.org/10.1016/j.epsl.2017.10.024, 2018.

Longpré, M.-A. and Felpeto, A.: Historical volcanism in the Canary Islands; part 1: A review of precursory and eruptive activity, eruption parameter estimates, and implications for hazard assessment, Journal of Volcanology and Geothermal Research, 419, 107 363,
https://doi.org/10.1016/j.jvolgeores.2021.107363, 2021.

Manjón-Cabeza Córdoba, A. and Ballmer, M. D.: The role of edge-driven convection in the generation of volcanism - Part 1: A 2D systematic study, Solid Earth, 12, 613–632, https://doi.org/10.5194/se-12-613-2021, 2021.

Marquart, G.: On the geometry of mantle flow beneath drifting lithospheric plates, Geophysical Journal International, 144, 356–372, https://doi.org/10.1046/j.0956-540X.2000.01325.x, 2001.

Martín, A., Sevilla, M., and Zurutuza, J.: Crustal deformation study in the Canary Archipelago by the analysis of GPS observations, Journal of Applied Geodesy, 8, 129–140, https://doi.org/10.1515/jag-2014-0002, 2014.

Martínez-Arevalo, C., Mancilla, F. d. L., Helffrich, G., and García, A.: Seismic evidence of a regional sublithospheric low velocity layer beneath the Canary Islands, Tectonophysics, 608, 586–599, https://doi.org/10.1016/j.tecto.2013.08.021, 2013.

Milelli, L., Fourel, L., and Jaupart, C.: A lithospheric instability origin for the Cameroon Volcanic Line, Earth and Planetary Science Letters,
335-336, 80–87, https://doi.org/10.1016/j.epsl.2012.04.028, 2012.

Moresi, L. N. and Gurnis, M.: Constraints on the lateral strength of slabs from three-dimensional dynamic flow models, Earth and Planetary Science Letters, 138, 15–28, https://doi.org/10.1016/0012-821x(95)00221-w, 1996.

Moresi, L. N. and Solomatov, V. S.: Numerical investigation of 2D convection with extremely large viscosity variations, Physics of Fluids, 7, 2154–2162, https://doi.org/10.1063/1.868465, 1995.

Morgan, W. J.: Convection Plumes in the Lower Mantle, Nature, 230, 42–43, https://doi.org/10.1038/230042a0, 1971.

Müller, R. D., Sdrolias, M., Gaina, C., and Roest, W. R.: Age, spreading rates, and spreading asymmetry of the world's ocean crust, Geochemistry, Geophysics, Geosystems, 9, Q04 006, https://doi.org/10.1029/2007GC001743, 2008.

Negredo, A. M., van Hunen, J., Rodríguez-González, J., and Fullea, J.: On the origin of the Canary Islands: Insights from mantle convection modelling, Earth and Planetary Science Letters, 584, 117 506, https://doi.org/10.1016/j.epsl.2022.117506, 2022.

Parsons, B. and McKenzie, D.: Mantle convection and the thermal structure of the plates, Journal of Geophysical Research, 83, 4485–4496, https://doi.org/10.1029/jb083ib09p04485, 1978.

Patriat, M. and Labails, C.: Linking the Canary and Cape-Verde Hot-Spots, Northwest Africa, Marine Geophysical Researches, 27, 201–215, https://doi.org/10.1007/s11001-006-9000-7, 2006.

Pertermann, M. and Hirschmann, M. M.: Anhydrous Partial Melting Experiments on MORB-like Eclogite: Phase Relations,
Phase Compositions and Mineral-Melt Partitioning of Major Elements at 2-3 GPa, Journal of Petrology, 44, 2173–2201, https://doi.org/10.1093/petrology/egg074, 2003.

Ramsay, T. and Pysklywec, R.: Anomalous bathymetry, 3D edge driven convection, and dynamic topography at the western Atlantic passive margin, Journal of Geodynamics, 52, 45–56, https://doi.org/10.1016/j.jog.2010.11.008, 2011.

Ribe, N. M. and Christensen, U.: The dynamical origin of Hawaiian volcanism, Earth and Planetary Science Letters, 171, 517–531, https://doi.org/10.1016/S0012-821X(99)00179-X, 1999.

Ribe, N. M. and Christensen, U. R.: Three-dimensional modeling of plume-lithosphere interaction, Journal of Geophysical Research, 99, 669–682, https://doi.org/10.1029/93JB02386, 1994.

Richter, F. M.: Convection and the large-scale circulation of the mantle, Journal of Geophysical Research, 78, 8735–8745, https://doi.org/10.1029/jb078i035p08735, 1973.

Richter, F. M. and Parsons, B.: On the interaction of two scales of convection in the mantle, Journal of Geophysical Research, 80, 2529–2541, https://doi.org/10.1029/jb080i017p02529, 1975.

Saki, M., Thomas, C., Nippress, S. E., and Lessing, S.: Topography of upper mantle seismic discontinuities beneath the North Atlantic: The Azores, Canary and Cape Verde plumes, Earth and Planetary Science Letters, 409, 193–202, https://doi.org/10.1016/J.EPSL.2014.10.052, 2015.

Schellart, W., Stegman, D., and Freeman, J.: Global trench migration velocities and slab migration induced upper mantle volume fluxes: Constraints to find an Earth reference frame based on minimizing viscous dissipation, Earth-Science Reviews, 88, 118–144, https://doi.org/10.1016/j.earscirev.2008.01.005, 2008.

Shahnas, M. H. and Pysklywec, R. N.: Anomalous topography in the western Atlantic caused by edge-driven convection, Geophysical Research Letters, 31, L18 611, https://doi.org/10.1029/2004GL020882, 2004.

Sleep, N. H.: Hotspots and mantle plumes: Some phenomenology, Journal of Geophysical Research, 95, 6715–6736, https://doi.org/10.1029/JB095iB05p06715, 1990.

Sleep, N. H.: Edge-modulated stagnant-lid convection and volcanic passive margins, Geochemistry, Geophysics, Geosystems, 8, n/a–n/a, https://doi.org/10.1029/2007GC001672, 2007.

Taracsák, Z., Hartley, M., Burgess, R., Edmonds, M., Iddon, F., and Longpré, M.-A.: High fluxes of deep volatiles from ocean island volcanoes: Insights from El Hierro, Canary Islands, Geochimica et Cosmochimica Acta, 258, 19–36, https://doi.org/10.1016/J.GCA.2019.05.020, 2019.

Thirlwall, M. F., Singer, B. S., and Marriner, G. F.: [39]Ar [40]Ar ages and geochemistry of the basaltic shield stage of Tenerife, Canary Islands, Spain, Journal of Volcanology and Geothermal Research, 103, 247–297, www.elsevier.nl/locate/jvolgeores, 2000.

Till, C. B., Elkins-Tanton, L. T., and Fischer, K. M.: A mechanism for low-extent melts at the lithosphere-asthenosphere boundary, Geochemistry, Geophysics, Geosystems, https://doi.org/10.1029/2010GC003234, 2010.

van Hunen, J., Zhong, S., Shapiro, N. M., and Ritzwoller, M. H.: New evidence for dislocation creep from 3-D geodynamic modeling of the Pacific upper mantle structure, Earth and Planetary Science Letters, 238, 146–155, https://doi.org/10.1016/J.EPSL.2005.07.006, 2005.

Vogt, P. R.: Bermuda and Appalachian-Labrador rises: Common non-hotspot processes?, Geology, 19, 41, https://doi.org/10.1130/0091-7613(1991)019<0041:BAALRC>2.3.CO;2, 1991.

Wang, S., Yu, H., Zhang, Q., and Zhao, Y.: Absolute plate motions relative to deep mantle plumes, Earth and Planetary Science Letters, 490, 88–99, https://doi.org/10.1016/j.epsl.2018.03.021, 2018.

Wilson, J. T.: A Possible Origin of the Hawaiian Islands, Canadian Journal of Physics, 41, 863–870, https://doi.org/10.1139/p63-094, 1963.

Zhong, S. J., Zuber, M. T., Moresi, L. N., and Gurnis, M.: Role of temperature-dependent viscosity and surface plates in spherical shell models of mantle convection, Journal of Geophysical Research: Solid Earth, 105, 11 063–11 082, https://doi.org/10.1029/2000JB900003, 2000.

730

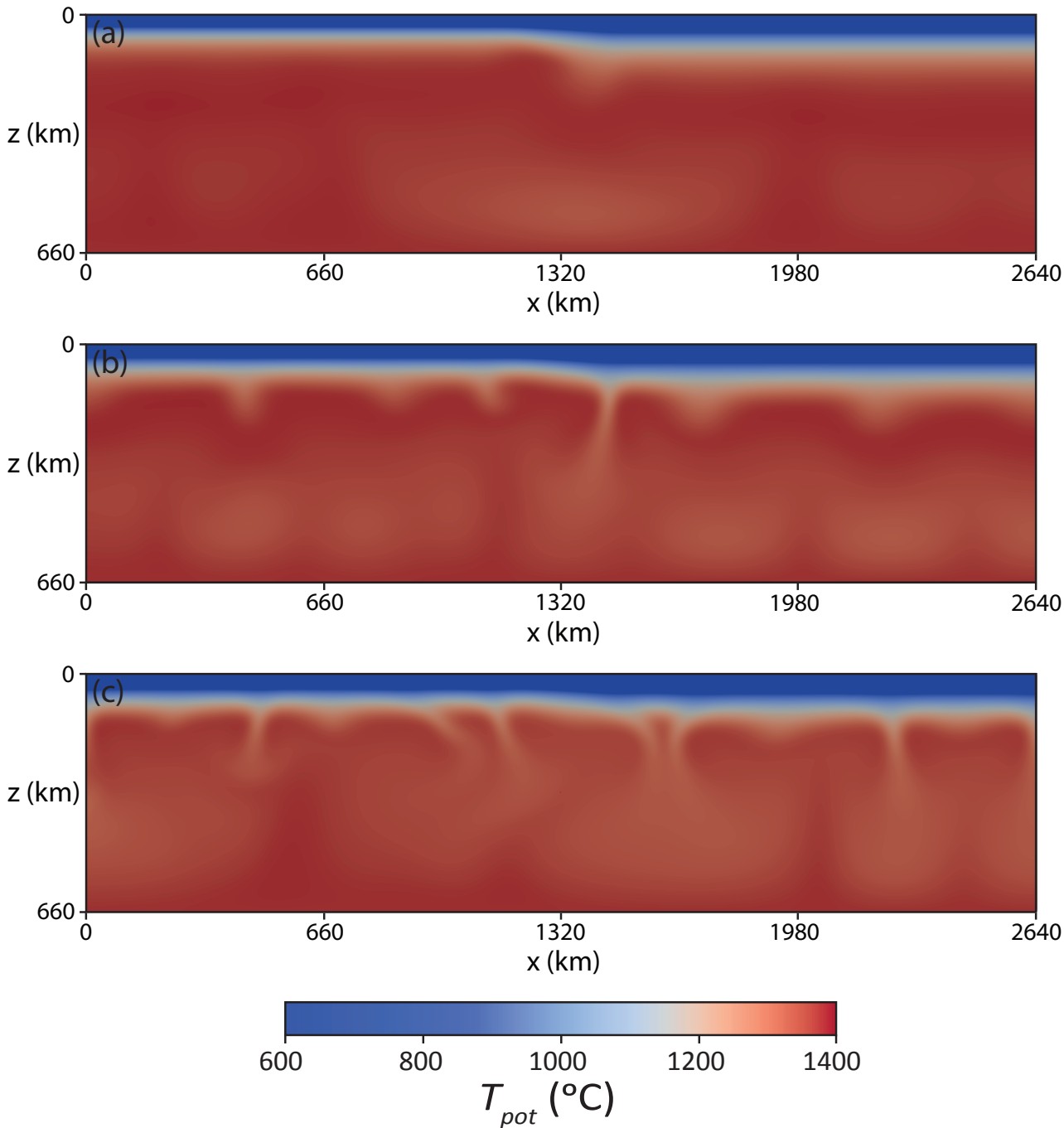

**Figure A3.** Vertical cross-sections of a case without a plume perpendicular to plate velocities. (a) y = 220 km; (b) y = 660 km; (c) y = 1320 km. Compared with Figure 3 of Manjón-Cabeza Córdoba and Ballmer (2021), changes in time of EDC are well represented in depth, while plate velocities do not disrupt the general planforms of EDC.

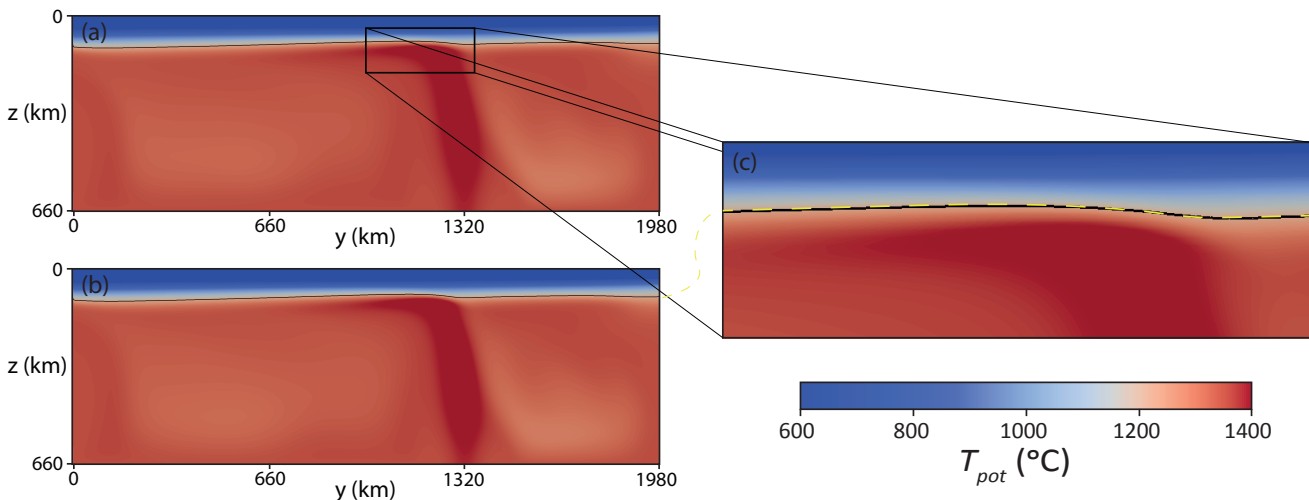

**Figure A4.** Vertical cross-sections parallel to the edge (perpendicular to the x-direction) and $x$ =1254 km. The isotherm of $T_{pot}$ = 1215 °C is drawn as a black line. (a) Reference case. (b) Case with $\Delta T$ = 200 °C and other values as in the reference case. (c) Closeup of the reference case with the isotherm of the case depicted in panel (b) (dashed yellow line). Note that due to plume deflection, the $x$ coordinte for maximum lithospheric erosion does not correspond with the location of the thermal anomaly at the bottom.

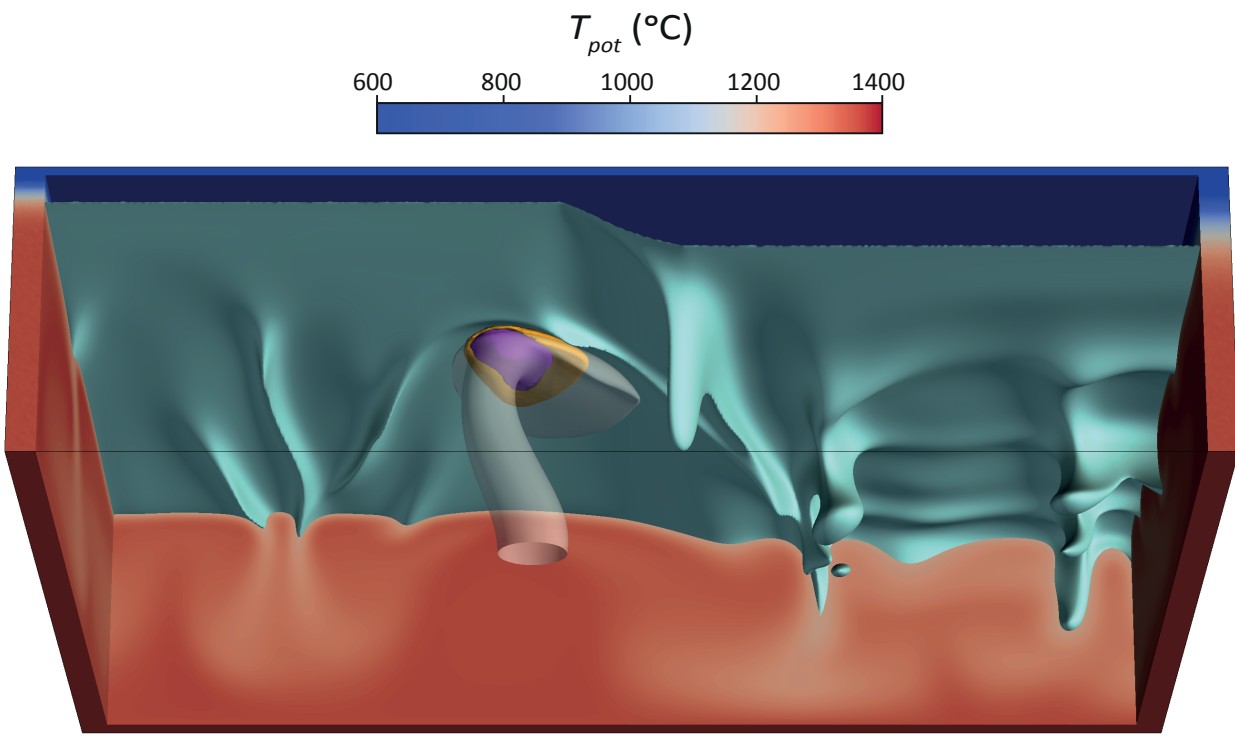

**Figure A5.** Snapshot of test case with an increased continental lithospheric thickness ($\tau_c$ = 200 Ma), $D_{plume}$ = 200 km, and otherwise properties as in the reference case. See text for details.