# Peer review of "The role of Edge-Driven Convection in the generation of volcanism-part 2: Interaction with Mantle Plumes, application to the Canary Islands"

_EGUsphere, 2022_

## Referee Comment (RC3)

**Title:** The role of edge-driven convection in the generation of volcanism – ii – interactions between edge-driven convection and thermal plumes with application to the Eastern Atlantic.
**Journal:** EGU Sphere
**Authors:** Cordoba & Ballmer
**Reviewer:** Rhodri Davies – The Australian National University

In this manuscript, authors use a 3-D geodynamical modelling approach to investigate potential interactions between mantle plumes and edge-driven convection. Based upon their modelling results, they subsequently link the dynamics and synthetic melt predictions to the intra-plate volcanic record of the Eastern Atlantic Ocean (particularly that associated with the Canary hotspot). The topic will clearly be of interest to a broad audience and the journal's readership.

Despite this, I feel that the paper suffers from a few major shortcomings, in addition to a lack of clarity in places, which should preclude publication in its current form. Although I outline several issues below, my suggestions amount to a major revision: I feel that addressing these points, and adding further clarity and support around major conclusions, will yield an important contribution towards better linking the volcanic record at Earth's surface to dynamical processes within its interior, particularly in the vicinity of step-changes in lithospheric thickness.

Hopefully these points are useful and will allow the authors to improve their study. Thank you for the opportunity to review this work, and my apologies for being a little slow.

Rhodri

**Main comments**

1. The interaction between edge-driven convective (EDC) cells and mantle plumes occurs both ways, with plumes likely modifying edge-driven cells and cells potentially influencing plumes. Although the authors quantify how plumes (both the conduit and pancake) are deflected during plume ascent in the vicinity of lithospheric steps, there was very little (if any) quantification about how edge driven cells behave prior to, during, and after, plume interaction. In other words, the study focuses on one aspect of the interaction between plumes and EDC, but does very little to shed light on other aspects. My expectation would be that the dynamics and melting expression of the cell adjacent to the lithospheric step changes quite dramatically upon interaction with a mantle plume, and this would have important manifestations in the geological record. However, the paper did not analyse this which, to me, is a major shortcoming: how can you examine the interaction between edge-driven convection and mantle plumes without quantitatively demonstrating how edge-driven convection is affected? Given that this paper builds squarely on the authors previous work (where 2-D edge-driven cells were examined in isolation), it is very important to quantify how results differ to those of that previous study: only by doing so can a reader really understand the role of a plume in this scenarios simulated in the paper. I'm left wondering: how, exactly, do plumes modify edge-driven cells? How does this interaction change with time? How is this manifest through melting and what are the potential implications of this for volcanic composition and volume? I'd strongly recommend that the authors compute some diagnostics that show, more definitively, how these two important melt-generation processes interact: this would really add value to the paper, as not many studies have examined such interactions.

2. I am not convinced by one of the paper's main conclusions, specifically that the ascent of plumes is modified by EDC. I am not doubting the authors results that a plume is deflected during its ascent, generally from beneath the continent towards, and away from, lithospheric steps. My uncertainty comes from what is causing this deflection. Unless I'm missing something (which is entirely possible!), given how the models examined have been set up, there will be a pressure gradient driving flow from beneath the continent (thick lithosphere) towards the oceanic realm (thinner lithosphere), which will be sufficient, in many cases, to deflect plumes in that direction. This is very different from a small-scale, shallow instability at the lithospheric step (i.e. edge-driven convection) inducing this deflection. With this in mind, I am left wondering what is causing plumes to deflect during their ascent in the models shown? Is it the larger-scale pressure gradient, or the shallow flow regime adjacent to the lithospheric step – i.e. EDC? I feel that the authors need to pull these potential mechanisms apart, to provide more support to their conclusions that shallow edge-driven convection is sufficient to deflect a plume. At what stage of its ascent does a plume start to deflect? It seems from the plume stem diagnostics shown and the 3-D snapshots provided that this happens at depth, which (at least to my understanding) fits better with the pressure gradient driving the deflection. The results highlighted on line 289-291 (for thicker continents) are also consistent with the pressure gradient being a key factor. One potential avenue that authors could use to pull these contributions apart would be to run an 'instantaneous flow model' where the thermal and compositional fields in their reference model remain fixed, and flow velocities are computed in response. Is flow driven towards the oceanic realm in this scenario? If so, what are these velocities relative to the ascent velocities of the plume? If these velocities are negligible (as I said, I could very well be wrong) and it turns out that shallow EDC is the main driver of plume deflection, I feel that a careful explanation of why this is the case would really add weight to the paper.

3. Given the manuscript title, I was expecting more background to the volcanic record of the Eastern Atlantic, as, ultimately, this is what the models were set up to understand. What is it about these volcanic provinces that is inconsistent with the mantle plume hypothesis and why? I feel that the manuscript falls short in this regard. If the authors really want to focus on the Eastern Atlantic, more background to regional volcanism should be provided, providing more context for a non-specialist reader. Saying that, the results of this paper are potentially also applicable to other intra-plate volcanic regions such as South America and Australia where interactions between plumes and lithospheric steps have been postulated (e.g. Davies et al. 2015, Rawlinson et al. 2017) - so the paper could potentially be expanded to include such regions, with less of an emphasis on the Eastern Atlantic. Obviously this is the authors decision - but both will be of interest.

4. In its current form, I would not be able to go and reproduce the results in this paper: the models are generally too briefly described. Yes, the authors refer back to their previous study, but I'm not a fan of having to dig out another paper to find some key model information. At the very least, the authors should provide more of a summary of how each component of their models are set-up in this paper (with only the in depth information restricted to the previous paper): I found some of this key information lacking (discussed further below).

5. The limitations of the models and how these may impact results need to be discussed. As with all models, there will be shortcomings and we have to make assumptions, but these should be highlighted to a reader. They can also be used to identify important avenues for future research. The authors are more qualified than I am to identify these limitations, but some aspects that I would recommend covering are: (i) models are 3-D which is great – however, the step geometry is essentially 2-D, extending across the entire length of the 3-D domain. The model therefore misses some 3-D complexity that likely exists on Earth and this should be acknowledged; (ii) melting model – I like the model used and it has some nice features, such as multi-component melting. However, please spell out its limitations for a non-expert (for example, do you consider reactions between pyroxenite melts and adjacent mantle? These are likely to be important.).

6. Results should really be better placed in the context of existing literature. There are a number of studies that have examined edge-driven convection and shear-driven upwelling. As you point out, fewer studies have examined the interaction between these processes and mantle plumes. Most of the key studies are cited, although not really discussed, whereas others are not cited or discussed. For example, the study of Duvernay et al. (2021), which you cite, whilst generally agreeing with the 2-D results of your previous study, can, in places predict melt fractions that seem compatible with some of the Eastern Atlantic volcanics quoted in your paper: part of the differences being due to 3-D complexity in lithospheric geometries incorporated in their models. This should be pointed out, so that a reader better understands the uncertainties around the modelling side. I think reviewing some of this literature and showing how your study builds on, complements and improves on earlier work, is important. These are a number of new, important and potentially very exciting findings in your study: for a reader to appreciate these, they need to be placed in the context of existing literature. The studies that that spring to my mind are (Demidjuk et al. 2007, Farrington et al. 2010, Davies & Rawlinson 2014, Afonso et al. 2016, Rawlinson et al. 2017), although I note that other reviewers have suggested some more (some of which I was not familiar with and will be reading myself!).

**Minor points**

1. Line 24 – it is stated that 'several predictions of plume theory are not fulfilled at many locations worldwide'. What aspects, specifically? Spell them out. I note that a number of studies demonstrate that thermo-chemical plumes can have a complex surface manifestation (e.g. Farnetani & Samuel 2005, Dannberg & Sobolev 2015) (in addition to some of Maxim's own work) whilst plumes simulated in a spherical geometry at realistic Rayleigh number can explain many of the complexities traditionally deemed inconsistent with mantle plume theory (e.g. Davies & Davies 2009). There are obviously other aspects of the volcanic record that seem inconsistent with mantle plumes, even when these complexities are taken in to account, and I agree that they are, but spell them out for a non-expert, so that they, and others in the community, better understand the motivation for the important work that you're doing (allowing them to better see the novelty in your paper).

2. Line 42 – 'in theory, the return upwelling flow would be enough to generate magma to sustain ocean island volcanism'... *provided that the overlying lid was sufficiently thin to facilitate decompression melting.* I think the additional qualifier is important, particularly for a non-expert.

3. Line 46 – 'very' is superfluous here. The Duvernay et al. (2021) study shows that EDC (and SDU) can account for many of Earth's lower volume (and potentially shorter lived) volcanic provinces – saying that magmatism is 'very' restricted could therefore give a false impression. It is markedly less than the magmatism induced by an upwelling plume, admittedly (as demonstrated in the more recent paper that is currently under review at G3: Duvernay et al. (2022)), but melting nonetheless remains significant.

4. Line 56: whilst it is true that not many studies have examined plumes interacting with EDC, some studies, by for example Koptev, Burov, Gerya, have carefully examined plume lithosphere interaction: it would be fair to

cite these here I think because the dynamical interactions that these studies highlight should be important for controlling magmatism in these settings.

5. Line 62 – remove comma.

6. Methods: as noted in main comments, several details of the modelling approach are lacking. This sections needs to be written more fully. Some key points for me (there are likely others):

   - Be specific that you are using the EBA approximation.

   - Is your mesh spacing uniform in the vertical dimension? Have you run resolution tests to confirm that these plume models are fully-resolved? I note that you mentioned this in your original paper, but these models are more complex and likely demand higher resolution, so wanted to confirm.

   - Line 75 – you specify a Couette profile at the inflow boundary that is consistent with the viscosity profile – spell out how you do this (from personal experience, it's not particularly straightforward, and requires explanation – unless again I'm missing the obvious!).

   - You have 'free-inflow' and an 'unconstrained' outflow boundary – are these fully unconstrained or do they essentially prescribe a hydrostatic pressure? Again, it's important to spell this out as they will drive very different flow regimes.

   - Line 89 – linearly interpolated transition. What is linearly interpolated along the transition? Age? Temperature? Depth of LAB? There will be subtle (but important!) differences between each.

   - Lower boundary condition – I find this highly unusual and it requires justification – you maintain an (almost) constant buoyancy flux with an open boundary condition by changing the radius over time. Why? Why not inject material at a constant buoyancy flux which will naturally be handled through the outflow boundary condition? There will clearly be a motivation behind your choice – but again, this needs to be explained – essentially you are switching between a zero normal-flow and an inflow boundary condition by changing $r$, which is unusual in finite element modelling.

   - Provide your viscosity relationship and a figure showing viscosity as a function of depth both inside and outside of the plume. Without this relationship, the key material property in your simulations is hard to visualise - and Section 3.4 is more challenging to interpret as a result.

7. The paper examines plumes with an excess temperature of 100-200K. I assume this is the excess temperature at the base of the model? Could the authors comment on how these temperatures change with depth and, specifically, what they are in the melt region for each case? In an EBA model, plume excess temperatures change with depth, so it'd be nice to have this information for comparison with other studies.

8. Line 118 – it is stated that conclusions from 2-D study hold in 3-D. This is true in this simplified geometry and it is indeed nice: but you are essentially assuming a 2-D step, so it is not overly surprising. As demonstrated in Davies & Rawlinson (2014), Duvernay et al. (2021), complex 3-D lithospheric geometries can lead to coalescing edge-driven cells, and secondary instabilities, which are further complicated by shear-driven upwelling and background mantle flow. These complexities can have important impacts on the flow regime and associated melting in the vicinity of lithospheric steps. This should probably be highlighted somewhere.

9. Line 127: 'this displacement suggests some interaction of plume flow with EDC-related flow' – see main comment 2 above. Likewise line 246 – 'plume deflection, caused by the effects of EDC'. I think you need to more clearly demonstrate cause and effect here.

10. Line 265: 'Since the vigour of EDC decreases with increasing viscosity' – also fair to cite Davies & Rawlinson (2014) here, in addition to Duvernay et al. (2021), both of which examined this sensitivity (amongst others).

11. Section 3.4 – effects of mantle viscosity: could you add a comparable image to Figure 7 showing the plumes in these cases? It may help a reader try to understand the puzzling results highlighted on lines 260-264.

12. Discussion – line 279 – end of paragraph 1: in this study, the buoyancy flux of the plume is one of the most important components controlling plume lithosphere interaction, but I think it's presumptuous to state that it is the *main influence on hotpost magmatism.* The models examined in this study are idealized. On Earth, the LAB is far more complex, and several studies argue that lithospheric structure is a key control on how plumes and EDC induce magmatism, particularly beneath continents such as Australia and Africa, which host large changes in lithospheric thickness over small length-scales. In addition, work by Burov, Gerya, Koptev (etc. . . ) demonstrates that the rheology of the crust and lithosphere will likely play a huge role on how plumes (and EDC) induce volcanism in these regions. With this in mind, I would suggest re-framing that statement - and acknowledging the other important factors not considered in your study.

13. Line 297 – final sentence of paragraph – I'm not sure I follow what is meant by this sentence sorry.

14. Line 306 – I find this statement interesting. The results of Duvernay et al. (2021) suggest that EDC could be sufficient to generate magmatic fluxes such as those observed in the Canary islands. Part of the reason that Duvernay et al. (2021) got higher melting rates was the addition of 3-D complexity, as noted above. I would therefore suggest toning down the statement that your previous paper 'clearly showed that EDC alone is insufficient to generate such magmatism'. The differences between melting rates in your study and Duvernay et al. (2021) probably need to be carefully examined (and I am not suggesting doing so as part of this paper) – but at this stage, I think your statement is too strong.

**References**

Afonso, J. C., Rawlinson, N., Yang, Y., Schutt, D. L., Jones, A. G., Fullea, J. & Griffin, W. L. (2016), '3-d multiobservable probabilistic inversion for the compositional and thermal structure of the lithosphere and upper mantle: Iii. thermochemical tomography in the western-central us', *Journal of Geophysical Research: Solid Earth* **121**(10), 7337–7370.

Dannberg, J. & Sobolev, S. V. (2015), 'Low-buoyancy thermochemical plumes resolve controversy of classical mantle plume concept', *Nature communications* **6**(1), 1–9.

Davies, D. R. & Davies, J. H. (2009), 'Thermally–driven mantle plumes reconcile multiple hotspot observations', *Earth Planet. Sci. Lett.* **278**, 50–54.

Davies, D. R. & Rawlinson, N. (2014), 'On the origin of recent intra-plate volcanism in Australia', *Geology* **42**, 1031–1034.

Davies, D. R., Rawlinson, N., Iaffaldano, G. & Campbell, I. H. (2015), 'Lithospheric controls on magma composition along Earth's longest continental hotspot track', *Nature* **525**, 511–514.

Demidjuk, Z., Turner, S., Sandiford, M., George, R., Foden, J. & Etheridge, M. (2007), 'U-series isotope and geodynamic constraints on mantle melting processes beneath the Newer Volcanic Province in South Australia', *Earth Planet. Sci. Lett.* **261**, 517–533.

Duvernay, T., Davies, D. R., Mathews, C., Gibson, A. H. & Kramer, S. C. (2022), 'Continental magmatism: The surface manifestation of dynamic interactions between cratonic lithosphere, mantle plumes and edge-driven convection'.

Duvernay, T., Davies, D. R., Mathews, C. R., Gibson, A. H. & Kramer, S. C. (2021), 'Linking intraplate volcanism to lithospheric structure and asthenospheric flow', *Geochem. Geophys. Geosys.* **22**(8), e2021GC009953.

Farnetani, C. G. & Samuel, H. (2005), 'Beyond the thermal plume paradigm', *Geophys. Res. Lett.* **32**.

Farrington, R. J., Stegman, D. R., Moresi, L. N., Sandiford, M. & May, D. A. (2010), 'Interactions of 3D mantle flow and continental lithosphere near passive margins', *Tectonophys.* **483**, 20–28.

Rawlinson, N., Davies, D. R. & Pilia, S. (2017), 'The mechanisms underpinning Cenozoic intraplate volcanism in eastern Australia: Insights from seismic tomography and geodynamic modeling', *Geophys. Res. Lett.* **44**, 9681–9690.

---

## Author Comment (AC2)

**Reply to comments by Ana M. Negredo on "The role of Edge-Driven Convection in the generation of volcanism – part 2: Interactions between Edge-Driven Convection and thermal plumes, application to the Eastern Atlantic"**

*Antonio Manjón-Cabeza Córdoba and Maxim D. Ballmer*

We thank referee Ana M. Negredo for her thorough review and her scientific input. Overall, we agree with the appreciations of the referee. Please, find the reply to your main comments below.

A more complete response, with the specific changes made and the relevant lines will be provided upon submission of the revised manuscript.

**Main comments:**

**1.** I think that the explanation of the model setup does not allow understanding fundamental aspects of the modelling as for example the implementation of the plume thermal anomaly and the initial phases of the model evolution. The authors should clearly explain how the 'statistical steady-state' (line 98) is achieved. For example, do the authors activate first EDC and later on force plume upwelling? Or both processes are activated simultaneously instead? How is plume upwelling forced? Is equation 1 a bottom boundary condition that forces the development of a plume? I guess this is the case, otherwise, how can r_plume change every 50 timesteps without artificially perturbing the thermal distribution? I find puzzling that this update with time of r_plume does not perturb the steady-state flow and thermal fields. Similarly, which is the radius of the opening at the bottom of the model mentioned in line 97?
Also the approximations assumed (extended Boussinesq approximation, I guess, as in their former EDC study) are not mentioned. Overall, please clearly state which are the initial and boundary conditions, explain how plume upwelling is forced and the initial evolution previous to the statistical steady-state. Below there are additional specific comments related to model setting definition.

In general, all reviewers agree that the methods section should be better explained. Therefore, we rewrite it considering the suggestions by the referee. We explain better the boundary conditions and (also to address a question by reviewer 3) the development of the plume. In addition, a brief explanation of the statistical steady state is included. We emphasize that the radius of the plume thermal anomaly and of the opening at the bottom are fixed at a specific value once the statistical steady state is reached

**2.** I realize that illustrating the dynamics of 3D models can be very challenging, but I still consider that the quality of this fully-coupled 3D modeling is somehow obscured by the figures shown in the manuscript. Note that results are illustrated in only two type of figures, the style of figures 3, 5, 7 and the style of figures 4, 6, 8, 9. For example the interplay between EDC and plume upwelling could be better illustrated in vertical crosssections showing the temperature, melting and velocity fields, at least for the reference case. That would also be useful to distinguish between SSC and EDC. There are a number of statements (I list them in the comments below) that are not illustrated at all in any figure. For example, the authors mention that figures shown (Figures 3, 5, 7) refer to thermal anomalies, and that the melting areas may be

deflected even more than the thermal anomaly' (lines 334-335), but this is not illustrated nor quantified by any means, which makes it difficult the comparison with the Canaries. Similarly, the important sentence: 'plume pancake may not necessarily be parallel to the plate movement' (lines 336-337), which is crucial for the comparison with the Canaries, could be shown for example on a horizontal section at the surface. I suggest adding additional figures showing for example the horizontal geometry of melting anomalies.

This was also a comment by referee 3. We have therefore re-done several figures to better illustrate both, the model setting and the model results.

**3.** Regarding the comparison with the Canary Archipelago, the authors state that several models predict 'deflection of the plume pancake and the melting zones toward the continental margin, which would explain the shape of the whole archipelago and the geographic distribution of volcanism'. However, this deflection is of only 25-35 km, so I don't see in which sense the interaction between plume and EDC is required to explain the E-W extension of the archipelago. In this sense, perhaps a control test with a flat lithosphere would be helpful to see how the geometry of the plume pancake is affected by the mentioned interaction. This is important to support the last conclusion, which states that for the Canary Islands a plume may be rising at 200 km from the continental margin, being deflected and creating the complex age progression and widespread volcanism. I agree that the lateral deflection may explain the widespread volcanism (although the plume pancake is only deflected 25-35 km), but the age progression is not reproduced provided that this modelled deflection is towards de edge, while the Canarian volcanism becomes younger away from the edge.

We apologize for the misunderstanding. As the referee suggest, we did not want to imply that the deflection of EDC can, by itself, reproduce the temporal evolution of the Canaries, but the current state. As suggested by the referee, plume migration may be needed. In fact, a previous version of the paper included a section about potential plume migration. We have rephrased the current text to be more specific and expanded the discussion regarding this issue. Note however that the absolute values of the deflection may change with different mantle properties or vigor of EDC ( to show this effect, we add a figure to the supplement). We are now also more specific regarding deflections.

**Minor comments.**

Lines 33-34. In the statement 'Besides, a cogenetic relation of these volcanoes with other volcanic fields has been suggested on the basis of geochemistry (Doblas et al., 2007; Duggen et al., 2009)', please, be more specific, which volcanic fields, which relation?

We add examples as per the reviewer suggestion.

Line 46. Please, mention here the similar results found in the recent 2D study by Negredo et al. (2022; EPSL doi: https://doi.org/10.1016/j.epsl.2022.117506) which is

closely related to the present work and was probably published after submission of the manuscript by Manjón-Cabeza Córdoba and Ballmer.

While we were aware of this work, the referee is correct, we couldn't cite it because it was not published (we expected its publication during review). We gladly cite it now.

Line 83-84 please clarify in which sense the models are 'bottom heated': by means of a temperature increase or a heat flow increase…?

We are more specific now.

Line 84 better say 'thermal distribution' (it is a surface) instead of 'thermal profile'

We corrected the statement.

Line 88. Why is the plume located at y=660 km rather than being centered in the box, at y=1980/2=990 km. This would make sense to avoid artefacts related to the different distance to the y-normal boundaries.

We had to decide a place that was far away enough from the inflow boundary (to avoid the aforementioned artifacts), but where we could still control EDC flow (*i.e.* close enough from the inflow boundary) and allowed the pancake and PET to fully develp. We set for y=660 km, but ran a test with the plume at the middle of the model confirming that results were qualitatively the same.

Line 93. Is this temperature increase a bottom boundary condition?

That is correct. We now describe the plume boundary conditions better.

Lines 135-136. this sentence 'Plume Erosion Track (PET) that is observed in all models Ribe and Christensen (1994)' seems ambiguous to me. Do the authors mean all models in this study? All models on plume dynamics?

While the PET should appear in all models of plume dynamics including plate velocity, we now corrected the sentence to a more conservative 'in all our models'.

Lines 136-137. The authors say 'In the reference case (fig. 3), the PET is mostly parallel to the direction of plate motion', but I cannot see this at all, mainly because the figure shows a snapshot in a steady-state situation, so it is difficult to see the development of a track.

Because the models are in steady state, the PET does not change (except for the models with 'cyclic' behavior). We now try to explain better what the plume track is and hope that it is clearer, also with the new figures.

Line 170. The sentence 'Nonetheless, the base of the lithosphere is eroded more efficiently for large DeltaTplume,' is not illustrated in any figure.

To better illustrate this, we add a supplementary figure where the lithospheric thickness of different models are compared.

Lines 220-221. The statement '…another notable phenomenon occurs: vigorous SSC occurs in the plume pancake with dominant transverse rolls (i.e., perpendicular to the

edge..' as well as the description of two melting anomalies that separate and merge periodically are not illustrated in any figure.

We agree with the referee that it is difficult to see this for the untrained reader based on snapshots (as we do not explicitly visualize the velocity field), particularly in terms of the periodic behavior. We add an animation of one of these cases in the supplementary material, which should provide a good intuition of the flow dynamics as well as periodic behavior.

Line 273-274. I don't understand this sentence 'We also find that the symmetry of the PET is higher for the cases with lower viscosity than for the case with intermediate and with high viscosity'. Symmetry with respect to what? Can the authors add any figure to illustrate this?

We meant symmetric with respect to an axis through the hotspot and parallel to the plate velocity (or parallel to the edge, same direction). We add a clarification and hope that the new figures illustrate this phenomenon better.

Lines 283-286. The authors affirm 'On the other hand, plume pancake deflection commonly (but not always) occurs towards the edge. This prediction may explain why some hotspot tracks (such as the Canaries) do not strictly align with plate velocity, and volcanism is widespread with more activity far from the continental margin than near to it (e.g., La Palma vs. Gran Canaria)'. I agree, but this would not be consistent with volcanic islands age decreasing away from the edge in the Canaries. Can the authors explain this?

We agree completely with the reviewer and apologize for not being clearer in the text. Obviously, some plume migration (or some other anomalous phenomenon) is required for the Canaries. We now added some lines in the discussion to specify it better in the text.

Lines 295-298. Here the authors compare model results with previous work about the interaction between mantle plume and lithospheric instabilities. In this context, a comparison with the recent 2D transient modelling by Negredo et al., (EPSL, 2022) is pertinent. The results obtained from both studies are consistent and complementary, although the sense of migration of the plume 'pancake' is opposed, perhaps because of the different timesteps of the simulations chosen for interpretation purposes.

We agree and this links with the previous comment. We have added several lines in the discussion to compare and discuss both model settings. Unfortunately, comparison of 2D plumes and 3D plumes is not straightforward, but we have tried our best.

Figure 3. The authors mention: 'The purple contour outlines the region of active melting while the orange contour outlines the region of finite melt presence, including where active melt re-freezing occurs'. I don't see these as contours, but rather as surfaces. Perhaps a vertical cross section through the plume would be useful. Why are colors al the side boundaries different from colors at the back face? Please, add orientation axis (easily added in Paraview).

We correct "contours" for "isosurfaces" in the text. As said above, we hope that the new figures help to illustrate our results better.

**List of typos:**

Line 49. Say 'these archipelagos' instead of 'this archipelagos'
Line 114 extra parenthesis )
Line 122 remove cf.[
Line 146 say 'and another' instead of 'an another'
Line 201. ..'the dotted grey line in fig. 6' please, specify which panel.
Lines 254-256. The sentence 'Very likely, these predictions have implications for dynamic
topography and swell geometry' is repeated.
Line 303 replace pyroxenite. by pyroxenite, (comma instead of point)
Line 382, remove the word 'plue' (plume, I guess)
Line 385, use lowercase E in Edge.
Figure 2. Use lowercase k for km (not Km).

We correct the typos and thank again the reviewer for her thorough review.

---

## Author Comment (AC3)

**Reply to comments by Rhodri Davies on "The role of Edge-Driven Convection in the generation of volcanism – part 2: Interactions between Edge-Driven Convection and thermal plumes, application to the Eastern Atlantic"**

*Antonio Manjón-Cabeza Córdoba and Maxim D. Ballmer*

We appreciate the thorough review by D. R. Davies. We agree with most of his formal comments and address them below. A full tracked-changes version will be provided upon submission of the revised manuscript.

Note: When citing our previous work and current manuscript, we remind the reviewer to refer to "Manjón-Cabeza Córdoba and Ballmer" instead of  "Córdoba and Ballmer".

**Main Comments**

1. The interaction between edge-driven convective (EDC) cells and mantle plumes occurs both ways, with plumes likely modifying edge-driven cells and cells potentially influencing plumes. Although the authors quantify how plumes (both the conduit and pancake) are deflected during plume ascent in the vicinity of lithospheric steps, there was very little (if any) quantification about how edge driven cells behave prior to, during, and after, plume interaction. In other words, the study focuses on one aspect of the interaction between plumes and EDC, but does very little to shed light on other aspects. My expectation would be that the dynamics and melting expression of the cell adjacent to the lithospheric step changes quite dramatically upon interaction with a mantle plume, and this would have important manifestations in the geological record. However, the paper did not analyse this which, to me, is a major shortcoming: how can you examine the interaction between edge-driven convection and mantle plumes without quantitatively demonstrating how edge-driven convection is affected? Given that this paper builds squarely on the authors previous work (where 2-D edge-driven cells were examined in isolation), it is very important to quantify how results differ to those of that previous study: only by doing so can a reader really understand the role of a plume in this scenarios simulated in the paper. I'm left wondering: how, exactly, do plumes modify edge-driven cells? How does this interaction change with time? How is this manifest through melting and what are the potential implications of this for volcanic composition and volume? I'd strongly recommend that the authors compute some diagnostics that show, more definitively, how these two important melt-generation processes interact: this would really add value to the paper, as not many studies have examined such interactions.

Considering this the other reviewers' comments we modify both, our figures and the explanation of the model setting. It is clear that we need to do a better job in these areas since it seems many questions reflect doubts concerning the model setting.
We agree that a detailed analysis of the effects of plumes on EDC should strengthen the paper. We added such analysis in the discussion and, in addition, we include an example of a model without a plume in the supplementary material to better distinguish between EDC and plume effects.

As a reminder, however, we analyze our models are in steady state, and therefore are not suited to study plume arrival.

2. I am not convinced by one of the paper's main conclusions, specifically that the ascent of plumes is modified by EDC. I am not doubting the authors results that a plume is deflected during its ascent, generally from beneath the continent towards, and away from, lithospheric steps. My uncertainty comes from what is causing this deflection. Unless I'm missing something (which is entirely possible!), given how the models examined have been set up, there will be a pressure gradient driving flow from beneath the continent (thick lithosphere) towards the oceanic realm (thinner lithosphere), which will be sufficient, in many cases, to deflect plumes in that direction. This is very different from a small-scale, shallow instability at the lithospheric step (i.e. edge-driven convection) inducing this deflection. With this in mind, I am left wondering what is causing plumes to deflect during their ascent in the models shown? Is it the larger-scale pressure gradient, or the shallow flow regime adjacent to the lithospheric step – i.e. EDC? I feel that the authors need to pull these potential mechanisms apart, to provide more support to their conclusions that shallow edge-driven convection is sufficient to deflect a plume. At what stage of its ascent does a plume start to deflect? It seems from the plume stem diagnostics shown and the 3-D snapshots provided that this happens at depth, which (at least to my understanding) fits better with the pressure gradient driving the deflection. The results highlighted on line 289-291 (for thicker continents) are also consistent with the pressure gradient being a key factor. One potential avenue that authors could use to pull these contributions apart would be to run an 'instantaneous flow model' where the thermal and compositional fields in their reference model remain fixed, and flow velocities are computed in response. Is flow driven towards the oceanic realm in this scenario? If so, what are these velocities relative to the ascent velocities of the plume? If these velocities are negligible (as I said, I could very well be wrong) and it turns out that shallow EDC is the main driver of plume deflection, I feel that a careful explanation of why this is the case would really add weight to the paper.

Again, we need to apologize for not explaining properly the setting of the models. The reviewer seems to suggest that there is pressure-drive flow from the right to the left side of the models (in the perspective of the current figures) due to open boundaries. However, our side boundaries at the left and right are closed (free slip). Only the front and back boundaries are open (the front with imposed Couette-like inflow; the back with free outflow). Accordingly, we do not expect any (or very minor) pressure-driven flow in the model setup.

We now better explain the boundary conditions of the model (which were also confusing for the other referees). We also include a case without a plume in the supplementary material (similar to the instantaneous case suggested by the reviewer) which shows nearly-identical results to our previous (2D) work, demonstrating that pressure-driven flow is indeed very minor or absent. Moreover, the vast majority of small scale convection cells (SSC, "Richter Rolls) are parallel to plate movement, which would not happen if the pressure-driven flow as suggested by the reviewer was strong.

3. Given the manuscript title, I was expecting more background to the volcanic record of the Eastern Atlantic, as, ultimately, this is what the models were set up to understand. What is it

about these volcanic provinces that is inconsistent with the mantle plume hypothesis and why? I feel that the manuscript falls short in this regard. If the authors really want to focus on the Eastern Atlantic, more background to regional volcanism should be provided, providing more context for a non-specialist reader. Saying that, the results of this paper are potentially also applicable to other intra-plate volcanic regions such as South America and Australia where interactions between plumes and lithospheric steps have been postulated (e.g. Davies et al. 2015, Rawlinson et al. 2017) - so the paper could potentially be expanded to include such regions, with less of an emphasis on the Eastern Atlantic. Obviously this is the authors decision - but both will be of interest.

We agree that the title was not a very good fit, and therefore modify it to be more specific (it was also overly long): "[…] part – 2: Interaction with Mantle Plumes, application to the Canary Islands. We add a more detailed discussion of volcanism at the Canaries, but also mention other hotspots on Earth.

4. In its current form, I would not be able to go and reproduce the results in this paper: the models are generally too briefly described. Yes, the authors refer back to their previous study, but I'm not a fan of having to dig out another paper to find some key model information. At the very least, the authors should provide more of a summary of how each component of their models are set-up in this paper (with only the in depth information restricted to the previous paper): I found some of this key information lacking (discussed further below).

We expand the methods section and added several explanations for clarity.

5. The limitations of the models and how these may impact results need to be discussed. As with all models, there will be shortcomings and we have to make assumptions, but these should be highlighted to a reader. They can also be used to identify important avenues for future research. The authors are more qualified than I am to identify these limitations, but some aspects that I would recommend covering are: (i) models are 3-D which is great – however, the step geometry is essentially 2-D, extending across the entire length of the 3-D domain. The model therefore misses some 3-D complexity that likely exists on Earth and this should be acknowledged; (ii) melting model – I like the model used and it has some nice features, such as multi-component melting. However, please spell out its limitations for a non-expert (for example, do you consider reactions between pyroxenite melts and adjacent mantle? These are likely to be important.).

We add some discussion about the major limitations of our models. We do not agree that a 'straight' edge is one of them, however. This is rather a simplification, which makes our results more general, than a limitation of the methodology.
The melting model has limitations which are arguably much more important than the reactions between pyroxenite and peridotite mentioned in the comment, which have only been addressed in very few geodynamic works (Ballmer *et al.*, 2013; Jones *et al.*, 2017). We add a few lines underlying these limitations (along with those suggested by the reviewer).

6. Results should really be better placed in the context of existing literature. There are a number of studies that have examined edge-driven convection and shear-driven upwelling.

As you point out, fewer studies have examined the interaction between these processes and mantle plumes. Most of the key studies are cited, although not really discussed, whereas others are not cited or discussed. For example, the study of Duvernay et al. (2021), which you cite, whilst generally agreeing with the 2-D results of your previous study, can, in places predict melt fractions that seem compatible with some of the Eastern Atlantic volcanics quoted in your paper: part of the differences being due to 3-D complexity in lithospheric geometries incorporated in their models. This should be pointed out, so that a reader better understands the uncertainties around the modelling side. I think reviewing some of this literature and showing how your study builds on, complements and improves on earlier work, is important. These are a number of new, important and potentially very exciting findings in your study: for a reader to appreciate these, they need to be placed in the context of existing literature. The studies that that spring to my mind are (Demidjuk et al. 2007, Farrington et al. 2010, Davies & Rawlinson 2014, Afonso et al. 2016, Rawlinson et al. 2017), although I note that other reviewers have suggested some more (some of which I was not familiar with and will be reading myself!)

This comment was raised (to a greater or lesser extent) by all reviewers. We therefore expanded the literature in the introduction section. We do not discuss shear-driven upwelling (SDU) in the introduction, however, since we do not have lateral pressure-driven flow in our models (see above), so we do not feel it is useful to mention it in the introduction. We now do include some lines in the discussion section on SDU, where we now also cite Duvernay *et al.* (2021) and some other relevant studies on.

**Minor points**
Line 24 – it is stated that 'several predictions of plume theory are not fulfilled at many locations worldwide'. What aspects, specifically? Spell them out. I note that a number of studies demonstrate that thermo-chemical plumes can have a complex surface manifestation (e.g. Farnetani & Samuel 2005, Dannberg & Sobolev 2015) (in addition to some of Maxim's own work) whilst plumes simulated in a spherical geometry at realistic Rayleigh number can explain many of the complexities traditionally deemed inconsistent with mantle plume theory (e.g. Davies & Davies 2009). There are obviously other aspects of the volcanic record that seem inconsistent with mantle plumes, even when these complexities are taken in to account, and I agree that they are, but spell them out for a non-expert, so that they, and others in the community, better understand the motivation for the important work that you're doing (allowing them to better see the novelty in your paper).

Line 24: Due to the main focus of our paper, we expanded our description of the Canary Islands, but kept short the discussion about plumes worldwide. We still added a couple of lines and references to other work.

Line 42 – 'in theory, the return upwelling flow would be enough to generate magma to sustain ocean island volcanism'. . . *provided that the overlying lid was sufficiently thin to facilitate decompression melting*. I think the additional qualifier is important, particularly for a non-expert

We agree, we added the clarification as suggested by the reviewer.

Line 46 – 'very' is superfluous here. The Duvernay et al. (2021) study shows that EDC (and SDU) can account for many of Earth's lower volume (and potentially shorter lived) volcanic provinces – saying that magmatism is 'very' restricted could therefore give a false impression. It is markedly less than the magmatism induced by an upwelling plume, admittedly (as demonstrated in the more recent paper that is currently under review at G3: Duvernay et al. (2022)), but melting nonetheless remains significant.

Overall, we disagree that the direct comparison with Duvernay *et al* (2021) is adequate in terms of discussing the volumes of purely EDC-related volcanism. As mentioned above, Duvernay *et al.* (2019) additionally considered the effects of SDU due to pressure-driven flow and additional geometrical complexities. In a less complex setting, we demonstrated in the peer-reviewed and published companion paper (Manjón-Cabeza Córdoba and Ballmer, 2021) that EDC alone is insufficient to sustain major volcanism, and related volcanism is usually very minor. Therefore, we prefer to keep the statement as it is.

4. Line 56: whilst it is true that not many studies have examined plumes interacting with EDC, some studies, by for example Koptev, Burov, Gerya, have carefully examined plume lithosphere interaction: it would be fair to cite these here I think because the dynamical interactions that these studies highlight should be important for controlling magmatism in these settings.

We added these references, although their dynamic, rheological, initial, and melting approximations make these models difficult to directly compare to ours (we would like to remind the reviewer that our models are in steady state).

5. Line 62 – remove comma

We removed it.

6. Methods: as noted in main comments, several details of the modelling approach are lacking. This sections needs to be written more fully. Some key points for me (there are likely others):

We expanded the method section.

• Be specific that you are using the EBA approximation.

We are more specific now. In fact, we use a simplification of the EBA, since our adiabatic gradient is linear with depth.

• Is your mesh spacing uniform in the vertical dimension? Have you run resolution tests to confirm that these plume models are fully-resolved? I note that you mentioned this in your original paper, but these models are more complex and likely demand higher resolution, so wanted to confirm.

Yes, the mesh is uniform in the vertical dimension. We now clarify the vertical resolution and explain the choice of grid spacing in the text.

• Line 75 – you specify a Couette profile at the inflow boundary that is consistent with the viscosity profile – spell out how you do this (from personal experience, it's not particularly straightforward, and requires explanation – unless again I'm missing the obvious!).

We now explain our assumptions for the inflow velocity boundary condition in the text. We calculate the Couette flow by using the upper velocity boundary condition (bottom is equal 0) and assuming constant stress. Under Newtonian conditions the calculation that ensues is straightforward.

• You have 'free-inflow' and an 'unconstrained' outflow boundary – are these fully unconstrained or do they essentially prescribe a hydrostatic pressure? Again, it's important to spell this out as they will drive very different flow regimes.

Sorry for the typo. In fact, the inflow is not free, but rather constrained by the viscosity profile and the upper velocity boundary. The outflow is free/unconstrained except for the condition that all outflow must be perpendicular to the boundary. We better describe this in the text now.

• Line 89 – linearly interpolated transition. What is linearly interpolated along the transition? Age? Temperature? Depth of LAB? There will be subtle (but important!) differences between each.

As described in the previous paper, it is both, temperature and composition of the lithosphere. We better specify this in the text.

• Lower boundary condition – I find this highly unusual and it requires justification – you maintain an (almost) constant buoyancy flux with an open boundary condition by changing the radius over time. Why? Why not inject material at a constant buoyancy flux which will naturally be handled through the outflow boundary condition? There will clearly be a motivation behind your choice – but again, this needs to be explained – essentially you are switching between a zero normal-flow and an inflow boundary condition by changing r, which is unusual in finite element modelling.

For some nodes of the mesh, we are switching between a zero vertical flow and free vertical inflow condition to obtain the preferred plume buoyancy flux. Once this value is reached, however, no such switches occur during the steady state, in which model results are evaluated. During the design of the models, we had to choose between injection, constant radius or constant buoyancy flux. We wanted to avoid injection because it causes artificial dynamic pressures that would influence how plumes interact with EDC. Constant radius is a more common approximation in these cases, but considering how different distances (between the plume and the edge) influence plume flow, the related differences of buoyancy flux between plumes can be significant, therefore making the comparison difficult (especially as far as melting volumes go). Finally, we settled with constant buoyancy flux because this parameter

is better constrained by observations (*e.g.*, dynamic topography) than the radius of the plume (King and Adam, 2014). We add an explanation to the methods section.

• Provide your viscosity relationship and a figure showing viscosity as a function of depth both inside and outside of the plume. Without this relationship, the key material property in your simulations is hard to visualise - and Section 3.4 is more challenging to interpret as a result.

We add a viscosity profile in the supplementary material to complement that of the companion paper (Manjón-Cabeza Córdoba and Ballmer, 2021).

7. The paper examines plumes with an excess temperature of 100-200K. I assume this is the excess temperature at the base of the model? Could the authors comment on how these temperatures change with depth and, specifically, what they are in the melt region for each case? In an EBA model, plume excess temperatures change with depth, so it'd be nice to have this information for comparison with other studies.

The reviewer's assumption is correct. Due to the compsumption of latent heat of fusion, the plume excess temperature is not meaningful in the melting region. However we add in the text the DT of the plume before the melting region (one third of the model, depth=220 km). We would like to add that, as mentioned above, the adiabat is imposed as a constant gradient and the models are incompressible. Under these approximations, DT does not change significantly with depth.

8. Line 118 – it is stated that conclusions from 2-D study hold in 3-D. This is true in this simplified geometry and it is indeed nice: but you are essentially assuming a 2-D step, so it is not overly surprising. As demonstrated in Davies & Rawlinson (2014), Duvernay et al. (2021), complex 3-D lithospheric geometries can lead to coalescing edgedriven cells, and secondary instabilities, which are further complicated by shear-driven upwelling and background mantle flow. These complexities can have important impacts on the flow regime and associated melting in the vicinity of lithospheric steps. This should probably be highlighted somewhere.

We agree that it is not surprising given our setting. We add the relevant references to the text. However, effectively they show that complex edge geometries boost magmatism for right or acute angles. It is less clear that obtuse or reflex angles, such as those present around the African cratons, will behave similarly to Duvernay 2021. An exception of course, may be the Cameroon Volcanic Line (added the text).

9. Line 127: 'this displacement suggests some interaction of plume flow with EDC-related flow' – see main comment 2 above. Likewise line 246 – 'plume deflection, caused by the effects of EDC'. I think you need to more clearly demonstrate cause and effect here.

We add a figure showing a case without a plume in the supplementary material.

10. Line 265: 'Since the vigour of EDC decreases with increasing viscosity' – also fair to cite Davies & Rawlinson (2014) here, in addition to Duvernay et al. (2021), both of which examined this sensitivity (amongst others).

We add citations.

11. Section 3.4 – effects of mantle viscosity: could you add a comparable image to Figure 7 showing the plumes in these cases? It may help a reader try to understand the puzzling results highlighted on lines 260-264.

Also in response to the comments of previous reviewers, we reorganize the figures.

12. Discussion – line 279 – end of paragraph 1: in this study, the buoyancy flux of the plume is one of the most important components controlling plume lithosphere interaction, but I think it's presumptuous to state that it is the main influence on hotpost magmatism. The models examined in this study are idealized. On Earth, the LAB is far more complex, and several studies argue that lithospheric structure is a key control on how plumes and EDC induce magmatism, particularly beneath continents such as Australia and Africa, which host large changes in lithospheric thickness over small length-scales. In addition, work by Burov, Gerya, Koptev (etc. . . ) demonstrates that the rheology of the crust and lithosphere will likely play a huge role on how plumes (and EDC) induce volcanism in these regions. With this in mind, I would suggest re-framing that statement - and acknowledging the other important factors not considered in your study.

We address this comment by adding text in the discussion section where the references suggested by the reviewer are included.

13. Line 297 – final sentence of paragraph – I'm not sure I follow what is meant by this sentence sorry.

Line 297: We apologize; we reword the end of the paragraph.

14. Line 306 – I find this statement interesting. The results of Duvernay et al. (2021) suggest that EDC could be sufficient to generate magmatic fluxes such as those observed in the Canary islands. Part of the reason that Duvernay et al. (2021) got higher melting rates was the addition of 3-D complexity, as noted above. I would therefore suggest toning down the statement that your previous paper 'clearly showed that EDC alone is insufficient to generate such magmatism'. The differences between melting rates in your study and Duvernay et al. (2021) probably need to be carefully examined (and I am not suggesting doing so as part of this paper) – but at this stage, I think your statement is too strong.

We respectfully disagree. First, we believe that the reviewer is underestimating the magmatic fluxes of the Canary Islands. In our previous work, we analyzed only the rates at El Hierro (Carracedo *et al.* 1998), since this value was enough to surpass our melting rates. However, the archipelago has two main shield-building active centers (La Palma and El Hierro).  La Palma alone displays volcanic fluxes of 1 km$^3$ / kyr (Day *et al.* 1999).  For the whole archipelago, independent calculations place the extrusion rate at 1-10 km$^3$ / kyr

depending on whether the Timanfaya eruption at Lanzarote (the largest Atlantic Tholeiitic eruption outside Iceland) is considered an anomalous event (Longpré and Felpeto, 2021). These estimates can easily be doubled when considering underplating and plutonism beneath the islands (Klügel *et al.*, 2005). Taken together, the rates in the Canary Islands are significantly larger than those obtained by Duvernay *et al.* (2021) for EDC models even when considering geometrical complexities that do not exist near the Canary Islands (0.8-0.9 km$^3$/kyr).

Nonetheless, we think the paper of Duvernay *et al.* is relevant for other locations in the Eastern Atlantic (Cameroon Volcanic Line), since their results related to the geometry of cratons are robust, and we expand the discussion regarding this work.

We also added to the methods section our calculations of a consistent thermochemical profile of the oceanic lithosphere via the pre-calculation of a simplified mid-ocean ridge model (an important difference with Duvernay *et al.*, 2021). This parameterization of the oceanic lithosphere in our models was specified in the companion paper, but we decided not to include it here for brevity. We apologize if this made things more complicated and thank again the referee for his thorough review.

**References**

Ballmer, M. D., Ito, G., Wolfe, C. J., and Solomon, S. C., 2013. *Double layering of a thermochemical plume in the upper mantle beneath Hawaii. Earth and Planetary Science Letters 376, 155-164.*

Carracedo, J. C., Day, S., Guillou, H., Rodríguez Badiola, E., Canas, J. A., and Pérez Torrado, F. J., 1998. *Hotspot volcanism close to a passive continental margin: the Canary Islands. Geological Magazine 135 (5), 591-604.*

Davies D. R., and Davies J. H., 2009. *Thermally-driven mantle plumes reconcile multiple hot-spot observations. Earth and Planetary Science Letters 278, 50-54.*

Day, S. J., Carracedo, J. C. , Guillou, H., and Gravestock, P., 1999. *Recent structural evolution of the Cumbre Vieja volcano, La Palma, Canary Islands: volcanic rift zone reconfiguration as a precursor to volcano flank instability? Journal of Volcanology and Geothermal Research, 94, 135-167*

Duvernay, T., Davies, D. R., Mathews, C. R., Gibson, A. H., Kramer, S. C., 2021. *Linking Intraplate Volcanism to Lithospheric Steps and Asthenospheric Flow. Geochemistry, Geophysics, Geosystems 22(8), e2021GC009953.*

Jones, T. D., Davies, D. R., Campbell, I. H., Iaffaldano, G., Yaxley, G., Kramer, S. C., and Wilson, C. R., 2017. *The concurrent emergence and causes of double volcanic hotspot tracks on the Pacific plate. Nature 545, 472-476.*

Klügel, A., Hansteen, T. H., and Galipp, K., 2005. *Magma storage and underplating beneath Cumbre Vieja volcano, La Palma (Canary Islands). Earth and Planetary Science Letters 236, 211-226*

Longpré, M.-A., and Felpeto, A., 2021. *Historical volcanism in the Canary Islands; part 1: A review of precursory and eruptive activity, eruption parameter estimates, and implications for hazard assessment. Journal of Volcanology and Geothermal Research 419, 107363*

Manjón-Cabeza Córdoba A., and Ballmer M., 2021. *The role of edge-driven convection in the generation of volcanism – Part 1: A 2D systematic study. Solid Earth 12 (3), 613-632.*

---

## Author Response (AR1)

**Reply to comments by Russel Pysklywec on "The role of Edge-Driven Convection in the generation of volcanism – part 2: Interactions between Edge-Driven Convection and thermal plumes, application to the Eastern Atlantic"**

*Antonio Manjón-Cabeza Córdoba and Maxim D. Ballmer*

We appreciate the comments by referee Russel Pysklywec.

Please, find detailed changes and line references (to the reviewed file) below. Note that we needed to slightly change the public comments to accommodate the specific changes made.

I appreciate that the authors are upfront about referring to this contribution as a "companion" study to their previous (2021) work. I think it's fine to write a paper like this, although it does reduce the novelty of the work somewhat--e.g., when this research is portrayed as an extension to the previous work even in the abstract. There are a few places where I think the authors rely too much on a citation to the previous study--detailed below--that should be expanded on.

While we chose this format because we believed that the work would be more readable and relevant, all reviewers agree that the dependence of this work on the previous study is excessive and/or the number of references is short. Therefore, we have decided to expand our introduction (see below) and the methods.

There are a number of relevant edge-driven convection studies that have been done, but aren't cited here. For example, similar questions on the role of EDC in the western Atlantic have explored features of topographic elevation, volcanism and elevated heat flow. I suggest that a paragraph be added to the Introduction to discuss some of this work: Vogt et al., Geology, 1991; Shahnas and Pysklywec, GRL, 2004; Conrad et al., Geology, 2004; Ramsay and Pysklywec, J. Geodyn., 2011. This would help fill in the discussion on EDC, but also expand the application and broaden the implications for the authors' work.

In this paper, we tried to keep the introduction as concise as possible, since the companion paper did explain the background of small-scale convection and EDC more thoroughly (for example, Vogt is cited in part 1). Nonetheless, all reviewers suggest to expand the introduction or/and discussion regarding EDC, as well as to add key citations, so we include Vogt *et al.* (line 46), Shahnas and Pysklywec, and Ramsay and Pysklywec (line 47).

Line 60: "We conclude that many of the discrepancies... We also find..." I don't think it's necessary or appropriate to put the conclusions in an Introduction section, and would suggest they be deleted here.

We delete it.

Line 73: Is it necessary to say "Kinematic boundary conditions are similar to those in the companion paper." (Again, the manuscript is already full of call-backs to the 2021 paper.) Suggest to delete.

Line 108: Here the authors refer readers to the 2021 companion paper for the density and viscosity formulation. I think these should be included in the present manuscript,

rather than just referred to. Many readers will be familiar with all the parameters listed in Table 1, but others won't be. The density and viscosity formulations will give these parameters context.

As mentioned above, we expand the methods section, which is now self-contained. See the added equations (lines 135-144) and parameters added to table 1.

Line 130: The authors mention that the plume migrates to a dent that is either created by EDC, or the active action of the plume, but leave this unanswered ("...it remains unclear..."). This seems exactly like the type of question this 3d dynamic models should answer: is there some reason why they didn't investigate the behaviour? e.g., prior to the plume impinging, was there local ocean lithosphere thinning?

We added a supplementary figure of a case without a plume to make more accurate comparisons (appendix A3). We also added a clarification of the lithosphere thinning (lines 171-172).

Line 132: Similarly, the unresolved question on the asymmetry seems like something they could/should answer with the models. The opportunity is there (unless I'm missing something with the modelling approach) it seems odd to leave it unanswered.

This comments were also raised by reviewer 3 (Rodhri Davies) and therefore we addressed it carefully. We now include a short reference to a 3D model without a plume and compare it with the plume cases throughout the text. We rephrased our results section (lines 182-184, 242-246, 290-295,349-356, and figure added in section 3.4). We also tried to be more assertive in the discussion, (lines 372-385). We would like to remind the referee, however, that in any Stokes model, all forcers are in balance and are solved together, and therefore all processes influence each other.

Line 277: Fix to "We ran a wide range..."
Line 288: Fix: "distance of the plume to from the edge."
Line 302: Fix to "Therefore, a subset of our models..."
Line 305: Fix flux units: 4?10^2 km^3 Myr

Corrected.

Figures 3, 5, 7: The cropping and dimensions are a bit confusing. (Is the thin black line the bottom boundary on the front face?; how does this reconcile with the depth of the left and right side wall boundaries?) Some appropriate labelling of the x-y-z coordinates would help guide readers on these.

We include labelled axes for the x-y-z coordinates. In addition, we also included additional panels to Figure 3 from different angles, to help the reader to understand the model setting. We also decreased the distortion (depth perspective) of figures 5 and 7 (and added figure) to help the reader have a clearer concept of our models.

**Reply to comments by Ana M. Negredo on "The role of Edge-Driven Convection in the generation of volcanism – part 2: Interactions between Edge-Driven Convection and thermal plumes, application to the Eastern Atlantic"**

*Antonio Manjón-Cabeza Córdoba and Maxim D. Ballmer*

We thank once again referee Ana M. Negredo for her thorough review and her scientific input. Please, find the full response to your comments below, including specific changes and new line numbers (comments are, therefore, slightly different to the public response).

**Main comments:**

**1.** I think that the explanation of the model setup does not allow understanding fundamental aspects of the modelling as for example the implementation of the plume thermal anomaly and the initial phases of the model evolution.

In general, all reviewers agree that the methods section should be better explained. Therefore, we rewrite it considering the suggestions by this and other authors' comments.

The authors should clearly explain how the 'statistical steady-state' (line 98) is achieved. For example, do the authors activate first EDC and later on force plume upwelling? Or both processes are activated simultaneously instead? How is plume upwelling forced? Is equation 1 a bottom boundary condition that forces the development of a plume? I guess this is the case, otherwise, how can r_plume change every 50 timesteps without artificially perturbing the thermal distribution? I find puzzling that this update with time of r_plume does not perturb the steady-state flow and thermal fields. Similarly, which is the radius of the opening at the bottom of the model mentioned in line 97?

We expand on the statistical steady state (lines 115-122, see also Appendix A1). We also expanded the explanation on the plume development (changes in lines 105-108). We emphasize that the radius of the plume thermal anomaly and of the opening at the bottom are fixed at a specific value once the statistical steady state is reached.

Also the approximations assumed (extended Boussinesq approximation, I guess, as in their former EDC study) are not mentioned. Overall, please clearly state which are the initial and boundary conditions, explain how plume upwelling is forced and the initial evolution previous to the statistical steady-state. Below there are additional specific comments related to model setting definition.

We now explicitly mention the Extended Boussinesq approximation (line 77-78). Finally, we improved figure 2, which we hope now makes things clearer.

**2.** I realize that illustrating the dynamics of 3D models can be very challenging, but I still consider that the quality of this fully-coupled 3D modeling is somehow obscured by the figures shown in the manuscript.

This was also a comment by referee 3. We have therefore re-done several figures to better illustrate both, the model setting and the model results.

Note that results are illustrated in only two type of figures, the style of figures 3, 5, 7 and the style of figures 4, 6, 8, 9. For example the interplay between EDC and plume upwelling could be better illustrated in vertical crossections showing the temperature, melting and velocity fields, at least for the reference case. That would alse be useful to distinguish between SSC and EDC. There are a number of statements (I list them in the comments below) that are not illustrated at all in any figure. For example, the authors mention that figures shown (Figures 3, 5, 7) refer to thermal anomalies, and that the melting areas may be deflected even more than the thermal anomaly' (lines 334-335), but this is not illustrated nor quantified by any means, which makes it difficult the comparison with the Canaries. Similarly, the important sentence: 'plume pancake may not necessarily be parallel to the plate movement' (lines 336-337), which is crucial for the comparison with the Canaries, could be shown for example on a horizontal section at the surface. I suggest adding additional figures showing for example the horizontal geometry of melting anomalies.

We included the horizontal cross-section of the reference case in Figure 3, and added another additional view that better showcases the general plate movement. Overall, we believe that PDI reflects better the deflection of the plume, as the figures can be deceiving: we try to explain it better in the text (see added discussion lines 367-371). We also changed the perspective of the figures so that the distortion (depth perspective) is decreased and the deflection of the plume better appreciated.

**3.** Regarding the comparison with the Canary Archipelago, the authors state that several models predict 'deflection of the plume pancake and the melting zones toward the continental margin, which would explain the shape of the whole archipelago and the geographic distribution of volcanism'. However, this deflection is of only 25-35 km, so I don't see in which sense the interaction between plume and EDC is required to explain the E-W extension of the archipelago. In this sense, perhaps a control test with a flat lithosphere would be helpful to see how the geometry of the plume pancake is affected by the mentioned interaction. This is important to support the last conclusion, which states that for the Canary Islands a plume may be rising at 200 km from the continental margin, being deflected and creating the complex age progression and widespread volcanism. I agree that the lateral deflection may explain the widespread volcanism (although the plume pancake is only deflected 25-35 km), but the age progression is not reproduced provided that this modelled deflection is towards de edge, while the Canarian volcanism becomes younger away from the edge.

We apologize for the misunderstanding. As the referee suggest, we did not want to imply that the deflection of EDC can, by itself, reproduce the temporal evolution of the Canaries, but the current state. As suggested by the referee, plume migration may be needed. In fact, a previous version of the paper included a section about potential plume migration. We have rephrased the current text to be more specific and expanded the discussion regarding this issue (lines 442-443 and 451-454). Note however that the absolute values of the deflection may change with different mantle properties or vigor of EDC (to show this effect, we added figure A5).

**Minor comments.**

Lines 33-34. In the statement 'Besides, a cogenetic relation of these volcanoes with other volcanic fields has been suggested on the basis of geochemistry (Doblas et al., 2007; Duggen et al., 2009)', please, be more specific, which volcanic fields, which relation?

We add examples as per the reviewer suggestion (lines 34-36)

Line 46. Please, mention here the similar results found in the recent 2D study by Negredo et al. (2022; EPSL doi: https://doi.org/10.1016/j.epsl.2022.117506) which is closely related to the present work and was probably published after submission of the manuscript by Manjón-Cabeza Córdoba and Ballmer.

While we were aware of this work, the referee is correct, we couldn't cite it because it was not published (we expected its publication during review). We gladly cite it now (lines 55,69,451)

Line 83-84 please clarify in which sense the models are 'bottom heated': by means of a temperature increase or a heat flow increase…?

We are more specific now. We rephrased it, see changes to lines (97-103)

Line 84 better say 'thermal distribution' (it is a surface) instead of 'thermal profile'

We corrected the statement.

Line 88. Why is the plume located at y=660 km rather than being centered in the box, at y=1980/2=990 km. This would make sense to avoid artefacts related to the different distance to the y-normal boundaries.

We had to decide a place that was far away enough from the inflow boundary (to avoid the aforementioned artifacts), but where we could still control EDC flow (i.e. close enough from the inflow boundary) and allowed the pancake and PET to fully develop. We set for y=660 km, but ran a test with the plume at the middle of the model confirming that results were qualitatively the same. See line 106

Line 93. Is this temperature increase a bottom boundary condition?

That is correct. We now describe the plume boundary conditions better (line 108)

Lines 135-136. This sentence 'Plume Erosion Track (PET) that is observed in all models Ribe and Christensen (1994)' seems ambiguous to me. Do the authors mean all models in this study? All models on plume dynamics?

While the PET should appear in all models of plume dynamics including plate velocity, we now corrected the sentence to a more conservative 'in all our models' (line 178)

Lines 136-137. The authors say 'In the reference case (fig. 3), the PET is mostly parallel to the direction of plate motion', but I cannot see this at all, mainly because the figure shows a snapshot in a steady-state situation, so it is difficult to see the development of a track.

Because the models are in steady state, the PET does not change (except for the models with 'cyclic' behavior, see supplementary video). We now try to explain better what the plume track is and hope that it is clearer. See added contours to figure 3 and lines (178-180)

Line 170. The sentence 'Nonetheless, the base of the lithosphere is eroded more efficiently for large DeltaTplume,' is not illustrated in any figure.

To better illustrate this, we rephrased it (lines 218-219) and added a supplementary figure (Appendix, figure A4).

Lines 220-221. The statement '…another notable phenomenon occurs: vigorous SSC occurs in the plume pancake with dominant transverse rolls (i.e., perpendicular to the edge..' as well as the description of two melting anomalies that separate and merge periodically are not illustrated in any figure.

We agree with the referee that it is difficult to see this for the untrained reader based on snapshots (as we do not explicitly visualize the velocity field), particularly in terms of the periodic behavior. We add a supplementary video of one of these cases in the supplementary material, which should provide a good intuition of the flow dynamics as well as periodic behavior.

Line 273-274. I don't understand this sentence 'We also find that the symmetry of the PET is higher for the cases with lower viscosity than for the case with intermediate and with high viscosity'. Symmetry with respect to what? Can the authors add any figure to illustrate this?

We meant symmetric with respect to an axis through the hotspot and parallel to the plate velocity (or parallel to the edge, same direction). We add a clarification and hope that the new figures illustrate this phenomenon better (line 329-336).

Lines 283-286. The authors affirm 'On the other hand, plume pancake deflection commonly (but not always) occurs towards the edge. This prediction may explain why some hotspot tracks (such as the Canaries) do not strictly align with plate velocity, and volcanism is widespread with more activity far from the continental margin than near to it (e.g., La Palma vs. Gran Canaria)'. I agree, but this would not be consistent with volcanic islands age decreasing away from the edge in the Canaries. Can the authors explain this?

We agree completely with the reviewer and apologize for not being clearer in the text. Obviously, some plume migration (or some other anomalous phenomenon) is required for the Canaries. We now added some lines in the discussion to specify it better in the text (lines 442-457).

Lines 295-298. Here the authors compare model results with previous work about the interaction between mantle plume and lithospheric instabilities. In this context, a comparison with the recent 2D transient modelling by Negredo et al., (EPSL, 2022) is pertinent. The results obtained from both studies are consistent and complementary, although the sense of migration of the plume 'pancake' is opposed, perhaps because of the different timesteps of the simulations chosen for interpretation purposes.

We agree and this links with the previous comment. We have added several lines in the discussion to compare and discuss both model settings (lines 451-454).

Unfortunately, comparison of 2D plumes and 3D plumes is not straightforward, but we have tried our best.

Figure 3. The authors mention: 'The purple contour outlines the region of active melting while the orange contour outlines the region of finite melt presence, including where active melt re-freezing occurs'. I don't see these as contours, but rather as surfaces. Perhaps a vertical cross section through the plume would be useful. Why are colors al the side boundaries different from colors at the back face? Please, add orientation axis (easily added in Paraview).

We correct "contours" for "isosurfaces" in the text. We added a plume scheme to the appendix (A4) and added axes to the initial settings and reference case. We changed the illumination angle for all the 3D figures so that the colormap is less confusing. Finally, we also reduced the depth-distortion of the figures.

**List of typos:**

Line 49. Say 'these archipelagos' instead of 'this archipelagos'
Line 114 extra parenthesis )
Line 122 remove cf.[
Line 146 say 'and another' instead of 'an another'
Line 201. ..'the dotted grey line in fig. 6' please, specify which panel.
Lines 254-256. The sentence 'Very likely, these predictions have implications for dynamic
topography and swell geometry' is repeated.
Line 303 replace pyroxenite. by pyroxenite, (comma instead of point)
Line 382, remove the word 'plue' (plume, I guess)
Line 385, use lowercase E in Edge.
Figure 2. Use lowercase k for km (not Km).

We correct the typos and thank again the reviewer for her thorough review.

**Reply to comments by Rhodri Davies on "The role of Edge-Driven Convection in the generation of volcanism – part 2: Interactions between Edge-Driven Convection and thermal plumes, application to the Eastern Atlantic"**

*Antonio Manjón-Cabeza Córdoba and Maxim D. Ballmer*

We again appreciate the thorough review by D. R. Davies. Note that the public comments have been modified to include specific changes and related lines.

**Main Comments**

1. The interaction between edge-driven convective (EDC) cells and mantle plumes occurs both ways, with plumes likely modifying edge-driven cells and cells potentially influencing plumes. Although the authors quantify how plumes (both the conduit and pancake) are deflected during plume ascent in the vicinity of lithospheric steps, there was very little (if any) quantification about how edge driven cells behave prior to, during, and after, plume interaction. In other words, the study focuses on one aspect of the interaction between plumes and EDC, but does very little to shed light on other aspects. My expectation would be that the dynamics and melting expression of the cell adjacent to the lithospheric step changes quite dramatically upon interaction with a mantle plume, and this would have important manifestations in the geological record. However, the paper did not analyse this which, to me, is a major shortcoming: how can you examine the interaction between edge-driven convection and mantle plumes without quantitatively demonstrating how edge-driven convection is affected? Given that this paper builds squarely on the authors previous work (where 2-D edge-driven cells were examined in isolation), it is very important to quantify how results differ to those of that previous study: only by doing so can a reader really understand the role of a plume in this scenarios simulated in the paper.

We agree that a more detailed analysis of the effects of plumes on EDC should strengthen the paper. We added such analysis in the discussion and in addition, we include an example of a model without a plume in the appendix to better distinguish between EDC and plume effects.

I'm left wondering: how, exactly, do plumes modify edge-driven cells?
We added some paragraphs in the results section (lines 182-184, 242-246, 290-295, 349-356) and in the discussion (lines 372-380). Given our initial and boundary conditions, the effect of plumes in EDC vigor may be too model-dependent, but we added a couple of lines in the discussion (line 381-385).

How does this interaction change with time?
Our models are designed as (quasi) steady state models, and therefore are not suited to study plume arrival, or the time-dependency of plume EDC interaction (see Appendix, Figure A1).

How is this manifest through melting and what are the potential implications of this for volcanic composition and volume? I'd strongly recommend that the authors compute some diagnostics that show, more definitively, how these two important melt-generation processes interact: this would really add value to the paper, as not many studies have examined such interactions.

There is no melting in our models that is directly related to EDC. Melting exclusively happens at the hotspot (where the plume conduit impinges the base of the lithosphere). We now clarify this in lines 159-161, now clearly pointing out that with the parameters used in our study, EDC alone is insufficient to support magmatism (Appendix A3 shows a model without a plume).

We also now clarify that the composition of melting due to plume+EDC is generally less enriched than for melting due to EDC-only, unless plume temperature and buoyancy flux are very small (lines 399).

2. I am not convinced by one of the paper's main conclusions, specifically that the ascent of plumes is modified by EDC. I am not doubting the authors results that a plume is deflected during its ascent, generally from beneath the continent towards, and away from, lithospheric steps. My uncertainty comes from what is causing this deflection. Unless I'm missing something (which is entirely possible!), given how the models examined have been set up, there will be a pressure gradient driving flow from beneath the continent (thick lithosphere) towards the oceanic realm (thinner lithosphere), which will be sufficient, in many cases, to deflect plumes in that direction. This is very different from a small-scale, shallow instability at the lithospheric step (i.e. edge-driven convection) inducing this deflection. With this in mind, I am left wondering what is causing plumes to deflect during their ascent in the models shown? Is it the larger-scale pressure gradient, or the shallow flow regime adjacent to the lithospheric step – i.e. EDC? I feel that the authors need to pull these potential mechanisms apart, to provide more support to their conclusions that shallow edge-driven convection is sufficient to deflect a plume.

Again, we need to apologize for not explaining properly the setting of the models. The reviewer seems to suggest that there is pressure-driven flow from the right to the left side of the models (in the perspective of the current figures) due to open boundaries. However, our side boundaries at the left and right are closed (free slip). Only the front and back boundaries are open (the front with imposed Couette-like inflow; the back with free outflow). Accordingly, we do not expect any pressure-driven flow in the model setup.

We now better explain the boundary conditions of the model. We also include a case without a plume in the appendix (Figure A3 similar to the instantaneous case suggested by the reviewer) which shows nearly-identical results to our previous (2D) work, demonstrating that pressure-driven flow is indeed very minor or absent. Moreover, the vast majority of small-scale convection cells (SSC, "Richter Rolls) are parallel to plate movement, which would not happen if pressure-driven flow as suggested by the reviewer was strong.

At what stage of its ascent does a plume start to deflect? It seems from the plume stem diagnostics shown and the 3-D snapshots provided that this happens at depth, which (at

least to my understanding) fits better with the pressure gradient driving the deflection. The results highlighted on line 289-291 (for thicker continents) are also consistent with the pressure gradient being a key factor.

One potential avenue that authors could use to pull these contributions apart would be to run an 'instantaneous flow model' where the thermal and compositional fields in their reference model remain fixed, and flow velocities are computed in response. Is flow driven towards the oceanic realm in this scenario? If so, what are these velocities relative to the ascent velocities of the plume? If these velocities are negligible (as I said, I could very well be wrong) and it turns out that shallow EDC is the main driver of plume deflection, I feel that a careful explanation of why this is the case would really add weight to the paper.

We show now a figure of a model without a mantle plume (Appendix, Figure A3) which shows 3 vertical cross-sections at different $y$ showing little-to-no pressure driven flow from the continent side to the ocean side (as expected given our previous work).

In summary, we addressed this comment by expanding the methods section (line 98-99, 93-96) and improving Figures 2 and 3, and adding Figure A3.

3. Given the manuscript title, I was expecting more background to the volcanic record of the Eastern Atlantic, as, ultimately, this is what the models were set up to understand. What is it about these volcanic provinces that is inconsistent with the mantle plume hypothesis and why? I feel that the manuscript falls short in this regard. If the authors really want to focus on the Eastern Atlantic, more background to regional volcanism should be provided, providing more context for a non-specialist reader. Saying that, the results of this paper are potentially also applicable to other intra-plate volcanic regions such as South America and Australia where interactions between plumes and lithospheric steps have been postulated (e.g. Davies et al. 2015, Rawlinson et al. 2017) - so the paper could potentially be expanded to include such regions, with less of an emphasis on the Eastern Atlantic. Obviously this is the authors decision - but both will be of interest.

We agree that the title was not a very good fit, and therefore modify it to be more specific (it was also overly long): "[…] part – 2: Interaction with Mantle Plumes, application to the Canary Islands".

We added a more detailed discussion of volcanism at the Canaries, but also mention other hotspots on Earth. The Introduction has also been changed and expanded (lines 45-60). We now include some references to Australia (line 48) and North-America (line 49). We do not include explicitly references to South America, but check our companion paper (Manjón-Cabeza Córdoba and Ballmer, 2021)

4. In its current form, I would not be able to go and reproduce the results in this paper: the models are generally too briefly described. Yes, the authors refer back to their previous study, but I'm not a fan of having to dig out another paper to find some key model information. At the very least, the authors should provide more of a summary of how each component of their models are set-up in this paper (with only the in depth information restricted to the previous paper): I found some of this key information lacking (discussed further below).

We improved our description of the boundary conditions (changes in lines 87-116) along with initial profiles (lines 131-132), as well as Figure 2. We also added our viscosity and density approximations (previously only in the companion paper, now in lines 135-144)

5. The limitations of the models and how these may impact results need to be discussed. As with all models, there will be shortcomings and we have to make assumptions, but these should be highlighted to a reader. They can also be used to identify important avenues for future research. The authors are more qualified than I am to identify these limitations, but some aspects that I would recommend covering are: (i) models are 3-D which is great – however, the step geometry is essentially 2-D, extending across the entire length of the 3-D domain. The model therefore misses some 3-D complexity that likely exists on Earth and this should be acknowledged; (ii) melting model – I like the model used and it has some nice features, such as multi-component melting. However, please spell out its limitations for a non-expert (for example, do you consider reactions between pyroxenite melts and adjacent mantle? These are likely to be important.).

We add some discussion about the major limitations of our models. We do not agree that a 'straight' edge is one of them, however. This is rather a simplification, which makes our results more general, than a limitation of the methodology.

Nevertheless, a discussion of the potential effects of geometrical complexities are added (lines 571-476, 490-495). We further added a discussion of model limitations concerning our rheology and melting approximation, and providing context for future work (lines 477- 495).

6. Results should really be better placed in the context of existing literature. There are a number of studies that have examined edge-driven convection and shear-driven upwelling. As you point out, fewer studies have examined the interaction between these processes and mantle plumes. Most of the key studies are cited, although not really discussed, whereas others are not cited or discussed.

This comment was raised (to a greater or lesser extent) by all reviewers. We therefore expanded the literature in the introduction section. We do not discuss shear-driven upwelling (SDU) in the introduction, however, since we do not have lateral pressure-driven flow in our models (see above), so we do not feel it is useful to mention it in the introduction.

Note: we refer here to SDU with the same meaning as Duvernay *et al.* (2021), but please be aware that the original concept of Shear-Driven Upwelling refers to a different process (Conrad *et al.*, 2010), which was the meaning we originally referred to in the manuscript.

For example, the study of Duvernay et al. (2021), which you cite, whilst generally agreeing with the 2-D results of your previous study, can, in places predict melt fractions that seem compatible with some of the Eastern Atlantic volcanics quoted in your paper: part of the differences being due to 3-D complexity in lithospheric geometries incorporated in their models. This should be pointed out, so that a reader better understands the uncertainties around the modelling side.

Duvernay et al (2021) is now discussed in line 55 (and also 474, 492)

I think reviewing some of this literature and showing how your study builds on, complements and improves on earlier work, is important. These are a number of new, important and potentially very exciting findings in your study: for a reader to appreciate these, they need to be placed in the context of existing literature. The studies that that spring to my mind are (Demidjuk et al. 2007, Farrington et al. 2010, Davies & Rawlinson 2014, Afonso et al. 2016, Rawlinson et al. 2017), although I note that other reviewers have suggested some more (some of which I was not familiar with and will be reading myself!)

We included several papers mentioned by the reviewer (lines 47-50) and mentioned by other reviewers (lines 46-47, 55).

**Minor points**
Line 24 – it is stated that 'several predictions of plume theory are not fulfilled at many locations worldwide'. What aspects, specifically? Spell them out. I note that a number of studies demonstrate that thermo-chemical plumes can have a complex surface manifestation (e.g. Farnetani & Samuel 2005, Dannberg & Sobolev 2015) (in addition to some of Maxim's own work) whilst plumes simulated in a spherical geometry at realistic Rayleigh number can explain many of the complexities traditionally deemed inconsistent with mantle plume theory (e.g. Davies & Davies 2009). There are obviously other aspects of the volcanic record that seem inconsistent with mantle plumes, even when these complexities are taken in to account, and I agree that they are, but spell them out for a non-expert, so that they, and others in the community, better understand the motivation for the important work that you're doing (allowing them to better see the novelty in your paper).

Line 24: Due to the main focus of our paper, we expanded our description of the Canary Islands in lines (34-36 and 56-60, see also 405-411). We did not expand the descriptions of other hotspots because, as the reviewer says, they could be explained by other combinations of processes.

Line 42 – 'in theory, the return upwelling flow would be enough to generate magma to sustain ocean island volcanism'. . . *provided that the overlying lid was sufficiently thin to facilitate decompression melting*. I think the additional qualifier is important, particularly for a non-expert

We agree, we added the clarification as suggested by the reviewer (Lines 44-45).

Line 46 – 'very' is superfluous here. The Duvernay et al. (2021) study shows that EDC (and SDU) can account for many of Earth's lower volume (and potentially shorter lived) volcanic provinces – saying that magmatism is 'very' restricted could therefore give a false impression. It is markedly less than the magmatism induced by an upwelling plume, admittedly (as demonstrated in the more recent paper that is currently under review at G3: Duvernay et al. (2022)), but melting nonetheless remains significant.

Overall, we disagree that the direct comparison with Duvernay et al (2021) is adequate in terms of discussing the volumes of purely EDC-related volcanism. As mentioned above, Duvernay et al. (2021) additionally considered the effects of SDU due to pressure-driven flow and additional geometrical complexities. In a less complex setting, we demonstrated in the peer-reviewed and published companion paper (Manjón-Cabeza Córdoba and Ballmer, 2021) that EDC alone is insufficient to sustain major volcanism, and related volcanism is usually very minor. Therefore, we prefer to keep the statement as it is.

See also lines (128-132, 405-411)

*4. Line 56: whilst it is true that not many studies have examined plumes interacting with EDC, some studies, by for example Koptev, Burov, Gerya, have carefully examined plume lithosphere interaction: it would be fair to cite these here I think because the dynamical interactions that these studies highlight should be important for controlling magmatism in these settings.*

We added these references (lines 386 and 394), although their dynamic, rheological, initial, and melting approximations make these models difficult to directly compare to ours (we would like to remind the reviewer that our models are examined in statistical steady state).

*5. Line 62 – remove comma*

We removed it.

*6. Methods: as noted in main comments, several details of the modelling approach are lacking. This sections needs to be written more fully. Some key points for me (there are likely others):*

We expanded the method section.

> *• Be specific that you are using the EBA approximation.*

> We are more specific now (line 77). In fact, we use a simplification of the EBA, since our adiabatic gradient is linear with depth.

> *• Is your mesh spacing uniform in the vertical dimension? Have you run resolution tests to confirm that these plume models are fully-resolved? I note that you mentioned this in your original paper, but these models are more complex and likely demand higher resolution, so wanted to confirm.*

> Yes, the mesh is uniform in the vertical dimension. We now clarify the vertical resolution and explain the choice of grid spacing in the text (Lines 83-86).

> *• Line 75 – you specify a Couette profile at the inflow boundary that is consistent with the viscosity profile – spell out how you do this (from personal experience, it's*

not particularly straightforward, and requires explanation – unless again I'm missing the obvious!).

We now explain our assumptions for the inflow velocity boundary condition in the text. We calculate the Couette flow by using the upper velocity boundary condition (bottom is equal 0) and assuming constant stress. Under Newtonian conditions the calculation that ensues is straightforward. See changes in lines 87-96

• You have 'free-inflow' and an 'unconstrained' outflow boundary – are these fully unconstrained or do they essentially prescribe a hydrostatic pressure? Again, it's important to spell this out as they will drive very different flow regimes.

Sorry for the typo. In fact, the inflow is not free, but rather constrained by the viscosity profile and the upper velocity boundary. The outflow is free/unconstrained except for the condition that all outflow must be perpendicular to the boundary. We better describe this in the text now. See changes in lines 87-96 and changes to Figure 2.

• Line 89 – linearly interpolated transition. What is linearly interpolated along the transition? Age? Temperature? Depth of LAB? There will be subtle (but important!) differences between each.

As described in the previous paper, it is both, temperature and composition of the lithosphere. We better specify this in lines 104-105

• Lower boundary condition – I find this highly unusual and it requires justification – you maintain an (almost) constant buoyancy flux with an open boundary condition by changing the radius over time. Why? Why not inject material at a constant buoyancy flux which will naturally be handled through the outflow boundary condition? There will clearly be a motivation behind your choice – but again, this needs to be explained – essentially you are switching between a zero normal-flow and an inflow boundary condition by changing r, which is unusual in finite element modelling.

For some nodes of the mesh, we are switching between a zero vertical flow and free vertical inflow condition to obtain the preferred plume buoyancy flux. Once this value is reached, however, no such switches occur during the statistical steady state stage, in which model results are evaluated. During the design of the models, we had to choose between injection, constant radius or constant buoyancy flux. We wanted to avoid injection because it causes artificial dynamic pressures that would influence how plumes interact with EDC. Constant radius is a more common approximation in these cases, but considering how different distances (between the plume and the edge) influence plume flow, the related differences of buoyancy flux between plumes can be significant, therefore making the comparison difficult (especially as far as melting volumes go). Finally, we settled with constant buoyancy flux because this parameter is better constrained by observations (e.g., dynamic topography) than

the radius of the plume (King and Adam, 2014). We add an explanation to the methods section. Lines 110-115 and supplementary video.

• Provide your viscosity relationship and a figure showing viscosity as a function of depth both inside and outside of the plume. Without this relationship, the key material property in your simulations is hard to 8isualize – and Section 3.4 is more challenging to interpret as a result.

We add a viscosity profile in the supplementary material to complement that of the companion paper (Manjón-Cabeza Córdoba and Ballmer, 2021). See Figure A2 (Appendix). Note that because plumes are deflected, the viscosity profile inside the plume is only orientative.

7. The paper examines plumes with an excess temperature of 100-200K. I assume this is the excess temperature at the base of the model? Could the authors comment on how these temperatures change with depth and, specifically, what they are in the melt region for each case? In an EBA model, plume excess temperatures change with depth, so it'd be nice to have this information for comparison with other studies.

The reviewer's assumption is correct. Due to the consumption of latent heat of fusion, the plume excess temperature is not meaningful in the melting region. Nevertheless, we now add in the text the DT of the plume just below the melting region (one third of the model, depth=220 km). We would like to add that, as mentioned above, the adiabat is imposed as a constant gradient and the models are incompressible. Under these approximations, DT does not change significantly with depth. See changes in lines 167-171.

8. Line 118 – it is stated that conclusions from 2-D study hold in 3-D. This is true in this simplified geometry and it is indeed nice: but you are essentially assuming a 2-D step, so it is not overly surprising. As demonstrated in Davies & Rawlinson (2014), Duvernay et al. (2021), complex 3-D lithospheric geometries can lead to coalescing edgedriven cells, and secondary instabilities, which are further complicated by shear-driven upwelling and background mantle flow. These complexities can have important impacts on the flow regime and associated melting in the vicinity of lithospheric steps. This should probably be highlighted somewhere.

We agree that it is not surprising given our setting. We add the relevant references to the text (lines 48, 55, 474, 493). However, effectively they show that complex edge geometries boost magmatism for right or acute angles. It is less clear that obtuse or reflex angles, such as those present around the African cratons, will behave similarly to Duvernay 2021. An exception of course, may be the Cameroon Volcanic Line (added in lines 472-477).

9. Line 127: 'this displacement suggests some interaction of plume flow with EDC-related flow' – see main comment 2 above. Likewise line 246 – 'plume deflection, caused by the effects of EDC'. I think you need to more clearly demonstrate cause and effect here.

We add a figure showing a case without a plume in the appendix *(*Figure A3). In addition, we improved the discussion (lines 369-395)

10. Line 265: 'Since the vigour of EDC decreases with increasing viscosity' – also fair to cite Davies & Rawlinson (2014) here, in addition to Duvernay et al. (2021), both of which examined this sensitivity (amongst others).

We add citations in line 314.

11. Section 3.4 – effects of mantle viscosity: could you add a comparable image to Figure 7 showing the plumes in these cases? It may help a reader try to understand the puzzling results highlighted on lines 260-264.

Also in response to the comments of previous reviewers, we reorganize the figures. We add a figure with examples for high Ra numbers (Figure 10) and expanded the results section for high Ra number (lines 320-328).

12. Discussion – line 279 – end of paragraph 1: in this study, the buoyancy flux of the plume is one of the most important components controlling plume lithosphere interaction, but I think it's presumptuous to state that it is the main influence on hotpost magmatism. The models examined in this study are idealized. On Earth, the LAB is far more complex, and several studies argue that lithospheric structure is a key control on how plumes and EDC induce magmatism, particularly beneath continents such as Australia and Africa, which host large changes in lithospheric thickness over small length-scales. In addition, work by Burov, Gerya, Koptev (etc. . . ) demonstrates that the rheology of the crust and lithosphere will likely play a huge role on how plumes (and EDC) induce volcanism in these regions. With this in mind, I would suggest re-framing that statement - and acknowledging the other important factors not considered in your study.

We address this comment by adding text in the discussion section where the references suggested by the reviewer are included. Lines 387-394

13. Line 297 – final sentence of paragraph – I'm not sure I follow what is meant by this sentence sorry.

We apologize; we reworded the end of the paragraph.

14. Line 306 – I find this statement interesting. The results of Duvernay et al. (2021) suggest that EDC could be sufficient to generate magmatic fluxes such as those observed in the Canary islands. Part of the reason that Duvernay et al. (2021) got higher melting rates was the addition of 3-D complexity, as noted above. I would therefore suggest toning down the statement that your previous paper 'clearly showed that EDC alone is insufficient to generate such magmatism'. The differences between melting rates in your study and Duvernay et al. (2021) probably need to be carefully examined (and I am not suggesting doing so as part of this paper) – but at this stage, I think your statement is too strong.

We respectfully disagree. First, we believe that the reviewer is underestimating the magmatic fluxes of the Canary Islands. In our previous work, we analyzed only the rates at El Hierro (Carracedo et al. 1998), since this value alone was enough to surpass our predicted melting rates. However, the archipelago has two main shield-building active centers (La Palma and El Hierro).  La Palma alone displays volcanic fluxes of 1 km$^3$ / kyr (Day et al. 1999). For the whole archipelago, independent calculations place the extrusion rate at 1-10 km$^3$ / kyr depending on whether the Timanfaya eruption at Lanzarote (the largest Atlantic Tholeiitic eruption outside Iceland) is considered an anomalous event (Longpré and Felpeto, 2021). These estimates can easily be doubled when considering underplating and plutonism beneath the islands (Klügel et al., 2005). Taken together, the rates in the Canary Islands are significantly larger than those obtained by Duvernay et al. (2021) for EDC models even when considering geometrical complexities that do not exist near the Canary Islands (0.8-0.9 km$^3$/kyr).

Nonetheless, we think the paper of Duvernay et al. is relevant for other locations in the Eastern Atlantic (Cameroon Volcanic Line), since their results related to the geometry of cratons are robust, and we expand the discussion regarding this work.

To a large extent, the differences in melting rates between our study (including the companion paper) and Duvernay et al. can be explained by the initial condition. We carefully calculate the initial compositional and thermal profile at the base of the oceanic lithosphere (now explained in lines 128-132). Such a self-consistent thermochemical profile of the oceanic lithosphere is computed from the pre-calculation of a simplified mid-ocean ridge model. This parameterization of the oceanic lithosphere in our models was specified in the companion paper, but we now also include it here. We apologize if this made things more complicated in the first place, and thank the referee again for his thorough review.

This explanation is summarized in lines 404-412. Duvernay et al.'s potential suitability for explaining the Cameroon Volcanic line is also included (line 475)

**References**

Ballmer, M. D., Ito, G., Wolfe, C. J., and Solomon, S. C., 2013. Double layering of a thermochemical plume in the upper mantle beneath Hawaii. Earth and Planetary Science Letters 376, 155-164.

Carracedo, J. C., Day, S., Guillou, H., Rodríguez Badiola, E., Canas, J. A., and Pérez Torrado, F. J., 1998. Hotspot volcanism close to a passive continental margin: the Canary Islands. Geological Magazine 135 (5), 591-604.

Davies D. R., and Davies J. H., 2009. Thermally-driven mantle plumes reconcile multiple hot-spot observations. Earth and Planetary Science Letters 278, 50-54.

Day, S. J., Carracedo, J. C. , Guillou, H., and Gravestock, P., 1999. Recent structural evolution of the Cumbre Vieja volcano, La Palma, Canary Islands: volcanic rift zone reconfiguration as a precursor to volcano flank instability? Journal of Volcanology and Geothermal Research, 94, 135-167

Duvernay, T., Davies, D. R., Mathews, C. R., Gibson, A. H., Kramer, S. C., 2021. Linking Intraplate Volcanism to Lithospheric Steps and Asthenospheric Flow. Geochemistry, Geophysics, Geosystems 22(8), e2021GC009953.

Jones, T. D., Davies, D. R., Campbell, I. H., Iaffaldano, G., Yaxley, G., Kramer, S. C., and Wilson, C. R., 2017. The concurrent emergence and causes of double volcanic hotspot tracks on the Pacific plate. Nature 545, 472-476.

Klügel, A., Hansteen, T. H., and Galipp, K., 2005. Magma storage and underplating beneath Cumbre Vieja volcano, La Palma (Canary Islands). Earth and Planetary Science Letters 236, 211-226

Longpré, M.-A., and Felpeto, A., 2021. Historical volcanism in the Canary Islands; part 1: A review of precursory and eruptive activity, eruption parameter estimates, and implications for hazard assessment. Journal of Volcanology and Geothermal Research 419, 107363

Manjón-Cabeza Córdoba A., and Ballmer M., 2021. The role of edge-driven convection in the generation of volcanism – Part 1: A 2D systematic study. Solid Earth 12 (3), 613-632.